# Mannose ameliorates experimental colitis by protecting intestinal barrier integrity

Lijun Dong[1,2,3], Jingwen Xie[2,3], Youyi Wang[2,3], Honglian Jiang[3], Kai Chen[3], Dantong Li[3], Jing Wang[4], Yunzhi Liu[2,3], Jia He[5], Jia Zhou [3], Liyun Zhang[3], Xiao Lu[3], Xiaoming Zou[1], Xiang-Yang Wang [6], Qingqing Wang [5] ✉, Zhengliang Chen [3] ✉ & Daming Zuo [1,2,7] ✉

Metabolite alteration has been associated with the pathogenesis of inflammatory bowel disease (IBD), including colitis. Mannose, a natural bioactive monosaccharide that is involved in metabolism and synthesis of glycoproteins, exhibits anti-inflammatory and anti-oxidative activities. We show here that the circulating level of mannose is increased in patients with IBD and mice with experimental colitis. Mannose treatment attenuates intestinal barrier damage in two mouse colitis models, dextran sodium sulfate (DSS)-induced colitis and spontaneous colitis in *IL-10*-deficient mice. We demonstrate that mannose treatment enhanced lysosomal integrity and limited the release of cathepsin B, preventing mitochondrial dysfunction and myosin light chain kinase (MLCK)-induced tight junction disruption in the context of intestinal epithelial damage. Mannose exerts a synergistic therapeutic effect with mesalamine on mouse colitis. Cumulatively, the results indicate that mannose supplementation may be an optional approach to the treatment of colitis and other diseases associated with intestinal barrier dysfunction.

Inflammatory bowel disease (IBD), including Crohn's disease (CD) and ulcerative colitis (UC), is characterized by chronic and relapsing inflammation in the gastrointestinal (GI) tract. The main features of IBD include dysregulated immune responses, abnormal cytokine production, imbalance of the intestinal flora, and barrier damage[1]. The intestinal barrier bridges the gap between intestinal microbes and the intestinal immune system and thus helps maintain mucosal homeostasis[2]. Impaired intestinal barrier function is often associated with a variety of GI disorders, including IBD[3]. The integrity of the intestinal barrier is mainly supported by epithelial cell tight junctions, which effectively prevent harmful substances (e.g., pathogens and endotoxin) from entering the blood through the intestinal mucosa[4,5]. The downregulation of tight junction protein expression can lead to increased permeability of the intestine in IBD[4]. Notably, the stability of the intestinal epithelium and the maintenance of tight junctions are key cellular processes that are dependent on properly functioning mitochondria[6]. The intestinal mucosa of patients with IBD appears to be in a state of energy deficiency, as indicated by low adenosine triphosphate (ATP) levels and low energy charge potential[6,7]. Increased mucosal mitochondrial oxidative phosphorylation (OXPHOS) activity and elevated ATP levels protect mice from colitis[8]. A recent study reported that metabolic syndrome accompanied by hyperglycemia is associated with an impaired intestinal barrier and an increased risk of GI infection[9]. It is therefore of interest

[1]The Fifth Affiliated Hospital, Southern Medical University, Guangzhou, Guangdong 510900, China. [2]Department of Medical Laboratory, School of Laboratory Medicine and Biotechnology, Southern Medical University, Guangzhou, Guangdong 510515, China. [3]Guangdong Provincial Key Laboratory of Proteomics, Department of Immunology, School of Basic Medical Sciences, Southern Medical University, Guangzhou, Guangdong 510515, China. [4]Department of Gastroenterology, Nanfang Hospital, Southern Medical University, Guangzhou, Guangdong 510515, China. [5]Institute of Immunology, Zhejiang University School of Medicine, Hangzhou, Zhejiang 310058, China. [6]Department of Human and Molecular Genetics, Virginia Commonwealth University, Richmond, VA 23298, USA. [7]Microbiome Medicine Center, Zhujiang Hospital, Southern Medical University, Guangzhou, Guangdong 510282, China. ✉e-mail: wqq@zju.edu.cn; zhlchen@smu.edu.cn; zdaming@smu.edu.cn

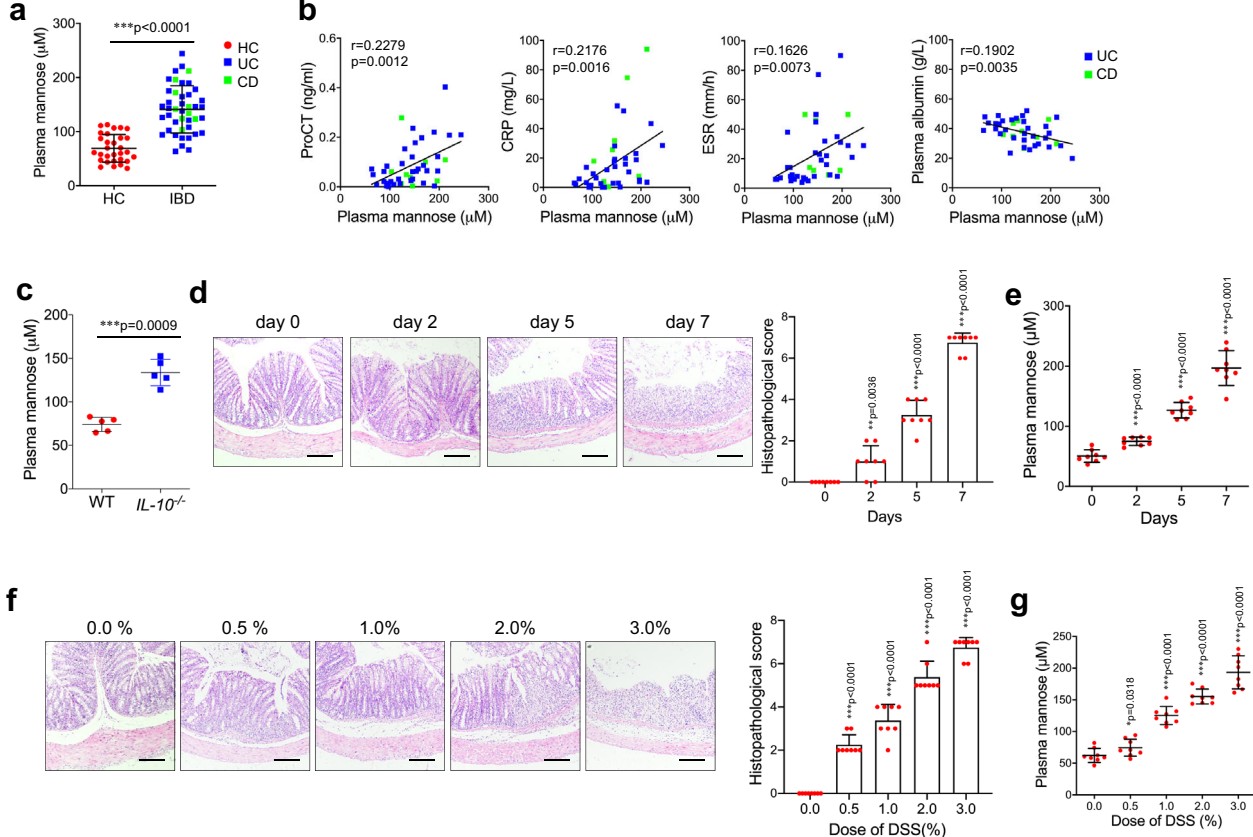

**Fig. 1 | Mannose levels are increased in IBD patients and colitis model mice.**
**a** The plasma mannose concentration in HCs ($n = 30$) and patients with IBD ($n = 44$), including CD ($n = 9$) and UC ($n = 35$), was determined by LC–MS/MS. Error bars represent 95% confidence intervals. **b** The relationship of plasma mannose concentration with ProCT, CRP, ESR, and plasma albumin levels in IBD patients. **c** The mannose level in serum samples from 19-week-old WT mice and *IL-10$^{-/-}$* spontaneous colitis model mice ($n = 5$) were assessed by LC–MS/MS. **d, e** C57BL/6 J mice ($n = 8$) received 3.0% DSS or tap water for the indicated days. **d** The colonic pathological changes were analyzed by H&E staining, and the histological scores of the DSS-induced colitis were blindly scored. **e** The serum mannose concentrations were evaluated at the indicated time points. **f, g** Mice

($n = 8$) were treated with the indicated dilution of DSS or tap water. On day 7, the colon tissue and blood were collected. The histological scores and serum mannose concentrations were compared with the group treated with tap water for day 0 (**d, e**) or tap water alone (**f, g**). **f** The colonic pathological changes were evaluated by H&E staining and scored. **g** The serum mannose concentrations were evaluated by LC–MS/MS analysis. Scale bar = 100 μm. *$p < 0.05$, **$p < 0.01$, ***$p < 0.001$. Data from one representative experiment of three independent experiments are presented. Data were represented as mean ± SD. Statistical significance was assessed by one-way ANOVA (**a**), two-sided Spearman's correlation (**b**), and two-sided Student's *t*-test (**c–g**). Source data are provided as a Source Data file.

to broaden our understanding of the potential link between metabolic regulation and GI disorders.

Mannose, a natural bioactive monosaccharide, is detectable in mammal plasma. A previous study showed that plasma concentrations of mannose positively correlate with insulin resistance independent of body mass index (BMI)[10]. Indeed, the elevation of plasma mannose levels is an indicator of the risk of several metabolic diseases, e.g., type 2 diabetes (T2D) and cardiovascular disease[11]. However, mannose also exhibits beneficial effects for the treatment of multiple diseases. Mannose can be used as a prophylactic agent to prevent recurrent urinary tract infections (UTIs) caused by *Escherichia*[12]. Mannose supplementation is an effective therapy for carbohydrate-deficient glycoprotein syndrome, an inherited metabolic disorder[13,14]. The oral administration of mannose in early life prevents diet-induced obesity in mice, which is attributed to the changes in gut microbiota composition and energy metabolism[15]. Additionally, mannose can suppress immunopathology in models of experimental type 1 diabetes (T1D) and lung airway inflammation[16]. It is noteworthy that metabolic disorders, such as obesity and T2D, are also associated with intestinal barrier dysfunction[9,17]. However, it remains unknown whether mannose may impact the intestinal barrier and the pathogenesis of colitis.

In this study, we report a significant increase in circulating mannose levels in both human subjects with IBD and colitis model mice. We further show that mannose can maintain the integrity of intestinal tight junctions in mice with dextran sodium sulfate (DSS)-induced colitis and *IL-10*-deficient (*IL-10$^{-/-}$*) mice with spontaneous colitis. Moreover, mannose treatment suppresses mitochondrial dysfunction both in vitro and in vivo in the context of intestinal epithelial damage. The beneficial effects of mannose may rely on its protection against lysosomal rupture and cathepsin B release in damaged intestinal epithelial cells. Our data suggest the potential use of mannose supplementation to treat colitis and other diseases related to tight junction dysfunction and lysosomal disorders.

## Results

### Increased circulating mannose levels correlate with disease progression in patients with IBD and colitis model mice

We first assessed the plasma levels of mannose in patients with IBD and age-matched healthy controls (HCs; Supplementary Tab. 1) using LC–MS/MS determination (Supplementary Fig. 1a). A significantly increased plasma mannose level was observed in patients with IBD compared with HCs (Fig. 1a). We next analyzed the disease severity based on several laboratory markers for IBD, such as procalcitonin

(ProCT), C-reactive protein (CRP), erythrocyte sedimentation rate (ESR), and plasma albumin[18]. The soluble mannose level was found to positively correlate with ProCT, CRP, and ESR levels (Fig. 1b). In addition, we found that the levels of plasma mannose negatively correlated with the plasma concentrations of albumin, a prognostic marker for IBD[18] (Fig. 1b). These data indicate an association of the plasma mannose level with IBD.

*IL-10*[-/-] mice, maintained under specific pathogen-free (SPF) conditions spontaneously develop colitis that is phenotypically similar to chronic IBD in humans[19]. Serum mannose levels were significantly higher in *IL-10*[-/-] mice than in wild-type (WT) controls (Fig. 1c and Supplementary Fig. 1b). Additionally, we made a similar observation in mice with DSS-induced colitis, a widely used chemically-induced experimental colitis model[20], which showed a substantial increase in the serum levels of mannose during disease progression (Fig. 1d, e). The serum level of mannose induced by the DSS challenge was dose-dependent (Fig. 1f, g). These results suggest that serum mannose levels correlate with colonic pathology in colitis model mice.

## Mannose ameliorates the severity of chemically-induced colitis and spontaneous colitis by strengthening tight junctions

To determine whether mannose affects colitis, we treated mice with 3.0% DSS to induce acute colitis with or without mannose supplementation. Mice simultaneously treated with DSS and mannose exhibited significantly less body weight loss than those treated with DSS alone (Fig. 2a), and this protective effect of mannose was dose-dependent (Supplementary Fig. 2a). Consistently, mannose supplementation also substantially attenuated the shortening of the colon length in DSS-challenged mice (Fig. 2b and Supplementary Fig. 2b). By contrast, glucose failed to protect against DSS-induced colitis when used at the same concentrations as mannose (Fig. 2a, b). Histological analyses showed reduced intestinal epithelial destruction, limited inflammatory cell infiltration, reduced goblet cell loss, and increased numbers of Ki67-positive cells in mannose-treated mice compared to control mice (Fig. 2c and Supplementary Fig. 2c). These results indicate that mannose treatment attenuates colonic inflammation and prevents the alteration of the colonic mucosal structure by DSS stimulation.

Because intestinal barrier dysfunction is a significant feature in the pathogenesis of IBD[4,5], we hypothesized that the protective effect of mannose in DSS-induced colitis is facilitated by maintaining intestinal barrier function. Mannose-treated colitis mice exhibited higher levels of serum albumin (Fig. 2d) and lower levels of fecal α1-antitrypsin (Fig. 2e) than the colitis mice. Moreover, mannose-treated colitis mice exhibited reduced plasma levels of fluorescein isothiocyanate-dextran, average mol wt 3000–5000 (FD4) compared with the control colitis mice after oral gavage of FD4 (Fig. 2f), indicating that mannose supplementation significantly alleviated intestinal permeability in DSS-treated mice. Of note, the protective effect of mannose on intestinal permeability was dose-dependent (Supplementary Fig. 2d). Notably, there is no gender difference in mannose-mediated attenuation of colonic inflammation in DSS-induced colitis mice (Supplementary Fig 3).

The expression levels and localization of zonula occludens-1 (ZO-1), occludin, and claudin-1 were examined by immunofluorescence staining. All three tight junction proteins were present along the inner lining of the columnar epithelium of the colon tissue, and the expression levels of these tight junction proteins were significantly increased in the colitis model mice concomitantly treated with mannose (Fig. 2g). The upregulation of tight junction proteins, including ZO-1, occludin, and claudins (claudin-1, −2, −4), was confirmed by immunoblotting analysis (Fig. 2h). We also found that mannose supplementation significantly inhibited the phosphorylation of myosin light chain 2 (MLC2), which is widely known to facilitate the disruption of the tight junction barrier[21] (Fig. 2i). In contrast, glucose did not affect the expression of tight junction proteins and the phosphorylation of

MLC2 in DSS-induced colitis mice when used at the same concentrations as mannose (Supplementary Fig. 4a, b). Mannose treatment significantly increased the transepithelial electrical resistance (TER) of DSS-stimulated NCM460 monolayers (Supplementary Fig. 5b). Consistently, mannose treatment restored the mRNA and protein expression of tight junction proteins in the DSS-treated colonic epithelial cells in vitro (Supplementary Fig. 5c–e). Notably, mannose also exhibited an attenuative effect on epithelial tight junction disruption when administered after DSS treatment (Supplementary Fig. 5f). However, mannose had minimal effect on tight junction proteins of cells when added before DSS stimulation (Supplementary Fig. 5g). Besides, after mannose treatment, an increase in the TER of DSS-stimulated primary colonic epithelial cells of mice was observed (Supplementary Fig. 5h). Immunoblotting analysis showed that mannose restored the expression of tight junction proteins in the DSS-treated primary colonic epithelial cells (Supplementary Fig. 5i). These results suggest that mannose could rescue the DSS-induced disruption of epithelial tight junctions.

Furthermore, we assessed the potential therapeutic effect of mannose treatment on *IL-10*[-/-] spontaneous colitis model mice. As expected, mannose treatment attenuated body weight loss and colonic shortening in *IL-10*[-/-] mice (Fig. 3a, b). Additionally, the *IL-10*[-/-] mice showed a significant reduction in colonic inflammation and inflammatory cell infiltration after mannose administration, as determined by histologic analyses (Fig. 3c). Moreover, the intestinal permeability was decreased in mannose-treated *IL-10*[-/-] mice compared to control mice (Fig. 3d, e). The immunofluorescence analysis showed that the fluorescence intensity and distribution of the tight junction proteins in the colonic tissue of *IL-10*[-/-] mice were significantly improved by mannose treatment (Fig. 3f). The results of the immunoblotting analysis revealed that mannose treatment enhanced the colonic expression of tight junction proteins in *IL-10*[-/-] mice (Fig. 3g). Additionally, mannose treatment inhibited MLC2 phosphorylation by inactivating MLCK in the colonic tissue of *IL-10*[-/-] mice (Fig. 3h). By contrast, glucose did not affect tight junction expression and MLC2 phosphorylation in the aged *IL-10*[-/-] mice when used at the same concentrations as mannose (Supplementary Fig. 4c, d). These data indicate that mannose treatment attenuates colonic inflammation and restores the tight junction damage in spontaneous colitis model mice.

## Epithelial tight junction function is associated with mitochondrial energy metabolism and the oxidative stress response

Mitochondrial oxidative energy metabolism plays a vital role in maintaining the gut epithelial barrier during inflammation[6,22,23]. We investigated whether mannose affects colitis-related mitochondrial dysfunction in colonic epithelial cells. Compared to DSS treatment alone, treatment with DSS combined with mannose resulted in markedly reduced mitochondrial dysfunction, as indicated by large mitochondrial mass, reduced levels of mitochondrial oxidants, and increased mitochondrial membrane potential (Fig. 4a). We next addressed whether mannose administration influences mitochondrial respiration in DSS-treated cells. We found that upon coculture with mannose, DSS-treated cells displayed significantly increased basal and maximum mitochondrial respiration (Fig. 4b). The ATP-producing capacity of DSS-stimulated cells in the presence of mannose was also increased compared with that in control cells (Fig. 4b). We further found that mannose treatment substantially restored the expression of respiratory complexes I, II, III, IV, and V (i.e., NADH dehydrogenase, succinate dehydrogenase, ubiquinol dehydrogenase, cytochrome c oxidase, and ATP synthase or FoF1-ATPase) in DSS-treated cells (Fig. 4c). Mitochondrial pyruvate dehydrogenase kinase 1 (PDHK1) is a key regulatory enzyme involved in the reduction of mitochondrial respiration via the phosphorylation and inactivation of pyruvate dehydrogenase (PDH), a gate-keeping mitochondrial enzyme[24]. We observed a reduction in PDHK1 expression that coincided with

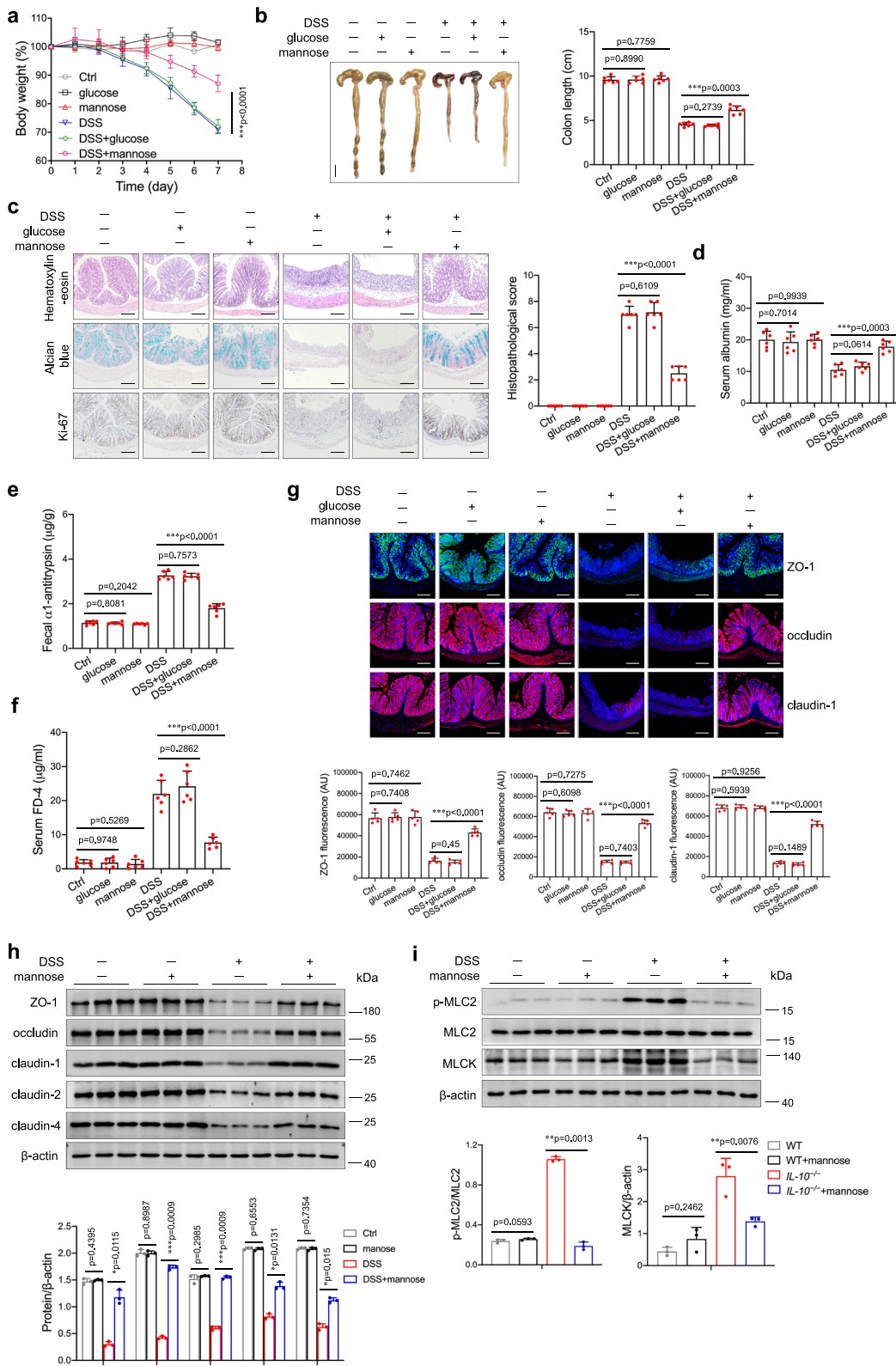

increased PDH expression in colonic epithelial cells stimulated with DSS plus mannose compared with those stimulated with DSS alone (Fig. 4c). Consistently, mannose treatment restored the levels of respiratory complexes and PDH in the colon tissues from DSS-treated mice and *IL-10⁻/⁻* mice, while glucose treatment did not show the effect (Supplementary Fig. 6a–d). The data also showed that mannose rescued the expression of respiratory complexes and PDH in the colonic epithelial cells from DSS-challenged mice (Supplementary Fig. 6e). To

further examine the role of mitochondrial respiration in the mannose-mediated regulation of tight junctions, colonic epithelial cells were pretreated with the mitochondrial respiratory inhibitor rotenone and the mitochondria-targeted antioxidant mito-tempo, respectively, before DSS and mannose administration. Rotenone pretreatment prevented the mannose-mediated restoration of tight junction protein expression in DSS-stimulated cells. In addition, the expression levels of tight junction proteins were similar in DSS only or DSS plus mannose-

**Fig. 2 | Mannose administration ameliorates DSS-induced colitis by protecting against intestinal barrier damage and tight junction disruption.** Mice (male, $n = 6$ per group) were treated with 3.0% DSS in the presence or absence of mannose (500 μg/g/d) for 7 consecutive days. Glucose (500 μg/g/d) served as a control sugar. **a** The body weight changes during the experiments were monitored. Mean ± SD from three independent experiments. **b, c** Colon tissues were isolated on the last day of the experiment. **b** A representative photograph of colon tissue from each group is provided, and the colon length was recorded. Scale bar = 1 cm. **c** The histological analysis of mouse colon tissue was performed by H&E, alcian blue, and Ki67 staining. Scale bar = 100 μm. Histological scores of the DSS-induced colitis were evaluated (9 slides/sample). **d–f** Intestinal permeability was determined by the albumin level of serum (**d**), the fecal α1-antitrypsin level (**e**), as well as serum FD4 concentration (**f**) on day 7 after the DSS challenge. **g** The localization of ZO-1, occludin, and claudin-1 in the mouse colon was determined on the last day of the experiment by immunofluorescence staining. Scale bar = 100 μm. **h** The level of ZO-1, occludin, claudin-1, claudin-2, and claudin-4 in the mouse colon was assessed by western blot analysis. **i** The levels of MLC2, phospho-MLC2, and MLCK protein in the colons from DSS-treated mice with or without mannose were analyzed using immunoblotting analysis. *$p < 0.05$, **$p < 0.01$, ***$p < 0.001$. Data were analyzed by two-side, unpaired Student's $t$-test (**a–i**) and shown as means ± SD. Data were representative of at least three independent experiments. Source data are provided as a Source Data file.

treated cells in the presence of rotenone (Fig. 4d). Similar to mannose treatment, mito-tempo pretreatment significantly reversed the decreased expression of tight junction proteins (Fig. 4e). Of note, the levels of tight junction proteins were similar in cells treated with DSS alone or DSS plus mannose in the presence of mito-tempo (Fig. 4e). Moreover, mito-tempo treatment relieved the DSS-induced colitis (Supplementary Fig. 7a–c) and protected the intestinal barrier disruption by DSS (Supplementary Fig. 7d–f). Collectively, these data show that mannose treatment of the damaged colonic epithelium facilitates mitochondrial functioning, which contributes to the energy production required for the expression of tight junction proteins.

## Mannose maintains lysosomal function in damaged colonic epithelial cells

Considering the structure of mannose[25], there's no binding site available for DSS, in other words, the direct binding of mannose to DSS is rarely possible. It was previously reported that mannose enters epithelial cells via the $Na^+$-dependent transport system[26]. We used ouabain to inhibit the $Na^+/K^+$-ATPase membrane pump, which abolished the mannose-mediated upregulation of tight junction proteins in DSS-treated cells (Fig. 5a). Additionally, the high levels of basal and maximal respiration, as well as the increase in ATP production induced by mannose, were significantly diminished by ouabain pretreatment (Fig. 5b). We next examined the intracellular distribution of mannose in colonic epithelial cells. Mannose (mannose-1-$^{13}$C) was used to trace the distribution of mannose as previously described[26]. LC–MS/MS analysis showed that intracellular residual mannose could be located in lysosomes in NCM460 cells (Fig. 5c). By contrast, glucose was dispersed in the cytoplasm and not specifically localized in the lysosome of epithelial cells (Fig. 5d). Several studies determined mannose can be used for further metabolism and glycoprotein biosynthesis[27,28]. We, therefore, measured the mannose metabolism in colonic epithelial cells over a long time. Besides the presence of mannose in the lysosome, a quantity of mannose was phosphorylated into mannose-6-phosphate or used to synthesize glycoproteins. Several other molecules, such as lactate, pyruvate, and GDP-mannose, were also observed in mannose-treated cells. (Fig. 5e–g). It has been reported that DSS-initiated inflammasome activation in macrophages is dependent on lysosomal maturation[29]. Indeed, DSS treatment induced a loss of red fluorescence in colonic epithelial cells stained with acridine orange, indicating lysosomal damage. Strikingly, mannose treatment rescued the loss of lysosomal integrity in NCM460 cells induced by DSS (Fig. 5h). A rapid loss of punctate lysotracker staining was observed in NCM460 cells exposed to DSS, which was reversed by mannose treatment (Fig. 5h). PDMPO staining showed that mannose treatment restored the increase of lysosome pH in NCM460 cells induced by DSS (Fig. 5i). To monitor the lysosomal membrane permeabilization, cells were labeled with FITC-dextran and LAMP2 after DSS combined with mannose stimulation. Fluorescent images revealed that mannose treatment increased the localization of FITC-dextran in lysosomes in DSS-treated cells (Fig. 5j).

The lysosomal-associated membrane protein1, 2 (LAMP-1 and LAMP2) are responsible in part for maintaining lysosomal integrity[30].

The western blotting analysis revealed increased levels of LAMP-1 and LAMP2 in DSS-treated NCM460 cells or primary colonic epithelial cells cocultured with mannose compared with those cultured without mannose (Supplementary Fig. 8). The inhibition of mannose transporter activity blocked mannose localization to lysosomes (Supplementary Fig. 9a) and abolished the increase in LAMP expression induced by mannose treatment in DSS-challenged cells (Supplementary Fig. 9b). Consistently, ouabain pretreatment prevented the mannose-mediated restoration of respiratory complex and PDH expression in DSS-stimulated cells (Supplementary Fig. 9c). To further validate the role of lysosomes in mannose-mediated protection against DSS-induced inflammation, NCM460 cells were treated with a lysosomal inhibitor, bafilomycin A1, before DSS and mannose treatment, and the tight junction proteins and mitochondrial respiration were analyzed. We firstly evaluated whether bafilomycin A1 treatment affected cell viability. The data showed that bafilomycin A1 had no effect on cell viability (Supplementary Fig. 10). The levels of tight junction proteins were comparable in cells treated with DSS alone or with DSS combined with mannose in the presence of bafilomycin A1 (Fig. 5k). Upon bafilomycin A1 pretreatment, the mitochondrial respiration in the DSS-treated cells with or without mannose coculture showed no significant difference, as indicated by seahorse analysis (Fig. 5l). Additionally, bafilomycin A1 treatment abrogated the mannose-mediated colonic protection in DSS-administrated mice (Supplementary Fig. 11). These data suggest that mannose improves tight junction by partially strengthening lysosomal function.

## Reduced release of the lysosomal enzyme cathepsin B is associated with the mannose-mediated regulation of tight junctions

Lysosomes include various enzymes, such as peptidases, phosphatases, and proteases[31]. Among these enzymes, cathepsins constitute a significant class of lysosomal proteases, which consist of aspartic, cysteine, and serine cathepsins[32]. Cathepsins are known to regulate an exquisite range of biological functions, and the dysregulation of cathepsin activity contributes to the development of inflammatory and metabolic diseases in humans[32,33]. Therefore, we sought to examine the effect of pharmacological inhibition of lysosomal cathepsins on the mannose-mediated upregulation of tight junction proteins. Among the cathepsin inhibitors, aloxistatin[34,35], pepstatin A[36], and nafamostat mesylate[37] have inhibitory effects on cysteine proteases (e.g., cathepsins B, K, and L), aspartic proteases (e.g., cathepsins D and E), and serine proteases (e.g., cathepsins A and G), respectively. Interestingly, only aloxistatin treatment cooperated with the mannose-mediated upregulation of tight junction protein expression in DSS-treated cells, whereas pepstatin A and nafamostat mesylate treatment did not (Supplementary Fig. 12a–d). It is worth mentioning that aloxistatin did not influence the cell viability of NCM460 cells at the concentration we used (Supplementary Fig. 12e, f). Of the cysteine proteases, cathepsin B has been implicated in a variety of cellular processes, such as cell death and inflammation[38]. The level of activated cathepsin B was increased significantly in the colon tissues from the DSS-induced colitis model mice, as indicated by

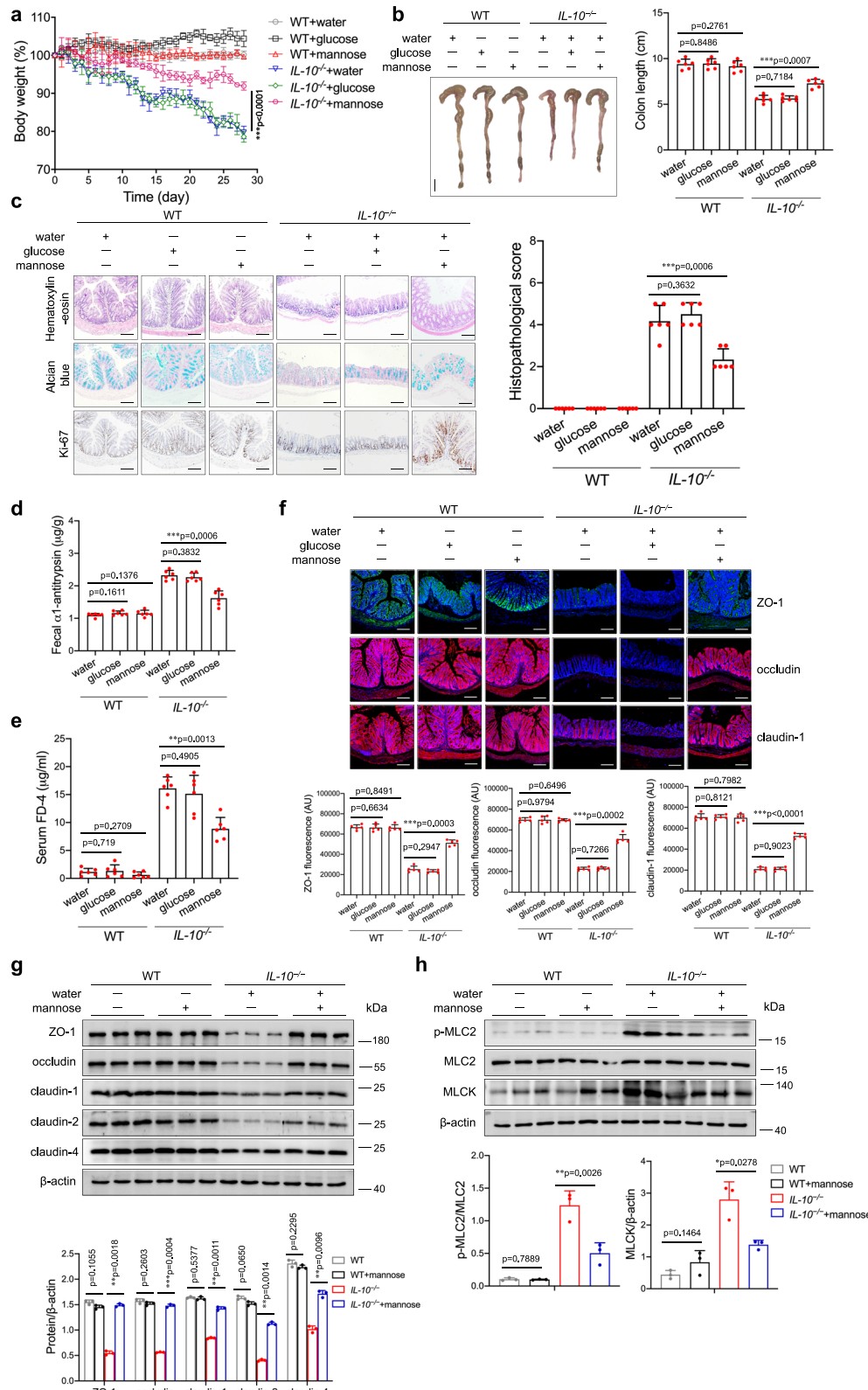

immunoblotting (Supplementary Fig. 13a) and immunohisto-chemical staining (Supplementary Fig. 13b). As expected, mannose treatment sharply suppressed the activation of cathepsin B in DSS-treated mice (Supplementary Fig. 13a, b). Similar observations were made when the activity of cathepsin B was measured in the colonic tissues from DSS-treated mice (Supplementary Fig. 13c). Of note, in vivo studies determined that the expression and activity of cytoplasma cathepsin B were reduced in the colonic epithelial cells

from mannose-treated colitis mice compared to those from control mice (Supplementary Fig. 13d, e). Notably, the mannose-treated *IL-10⁻/⁻* mice also exhibited less cathepsin B expression than untreated *IL-10⁻/⁻* mice (Supplementary Fig. 14a, b). In contrast, glucose treatment did not influence the cathepsin B activation in DSS-induced colitis model mice (Supplementary Fig. 14c) and *IL-10⁻/⁻* colitis mice (Supplementary Fig. 14d). We also observed that cathepsin B release is mainly from the colonic epithelial cells in the

**Fig. 3 | Mannose treatment attenuates chronic colitis and intestinal barrier dysfunction in *IL-10*[-/-] mice.** The *IL-10*[-/-] mice (15 weeks old, *n* = 6 per group) were fed with 1.0% mannose or 1.0% glucose for 4 weeks. **a** The body weight changes during the experiments were monitored. **b, c** Colon tissues were isolated on the last day of the experiment. **b** A representative photograph of the colon tissue from each group is provided, and the colon length was recorded. Scale bar = 1 cm. **c** Histological analysis of mouse colon tissue was performed by H&E, alcian blue, and Ki67 staining. Scale bar = 100 μm. Histological scores of the colitis damage were evaluated (9 slides/sample). **d, e** Intestinal permeability was determined by the fecal α1-antitrypsin concentration (**d**) and serum FD4 level (**e**) on day 28 after mannose or glucose treatment. **f** The localization of ZO-1, occludin, and claudin-1 in the mouse colon was determined on the last day of the experiment by immuno-fluorescence staining. Scale bar = 100 μm. **g** The level of of ZO-1, occludin, claudin-1, claudin-2, and claudin-4 in the mouse colon was assessed by western blot analysis. **h** The levels of MLC2, phospho-MLC2, and MLCK protein in the colons from *IL-10*[-/-] mice with or without mannose treatment were analyzed using an immunoblotting assay. **P* < 0.05, ***P* < 0.01, and ****P* < 0.001. Data from one representative experiment of three independent experiments are presented. Data were means ± SD. Two-side, unpaired *t*-test for (**a**–**i**). Source data are provided as a Source Data file.

colon from DSS-challenged mice (Supplementary Fig. 14e). Additionally, mannose treatment also slightly suppressed the increased expression of cathepsin D and cathepsin L in the colon tissue from DSS-challenged mice (Supplementary Fig. 14f). Furthermore, we examined the effect of mannose on DSS-induced cathepsin B activation in colonic epithelial cells in vitro. The results showed that mannose treatment strongly inhibited the increase in cathepsin B in DSS-treated cells (Fig. 6a, b) as well as the activity of cathepsin B (Fig. 6c). It is worth noting that mannose treatment inhibited the level and activity of cytoplasm cathepsin B in colonic epithelial cells treated with DSS in vitro (Fig. 6d, e). Remarkably, the efficiency of a mannose-mediated inhibitory effect of cathepsin B release from the lysosome to the cytoplasm was associated with the level of mannose in the lysosome (Supplementary Fig. 15).

To further validate the role of cathepsin B in mannose-mediated protection against mitochondrial dysfunction and tight junction disruption, colonic epithelial cells were pretreated with a specific cathepsin B inhibitor, CA-074, prior to DSS and mannose treatment. Similar to mannose treatment, CA-074 pretreatment significantly reversed the decreased expression of tight junction proteins (Fig. 6f). The levels of tight junction proteins were similar in cells treated with DSS alone or DSS plus mannose in the presence of CA-074 (Fig. 6f). CA-074 treatment also reversed the DSS-induced mitochondrial damage, as indicated by increased mitochondrial mass, reduced mitochondrial oxidant levels, and increased mitochondrial membrane potential (Fig. 6g, h). Additionally, cathepsin B inhibition substantially restored the expression of mitochondrial PDH upon DSS administration (Fig. 6i). The mitochondrial status and PDH expression levels were comparable in cells treated with DSS alone or DSS plus mannose (Fig. 6i). Notably, the absence of PDH diminished the protective effect of mannose on tight junctions but did not affect the activation of cathepsin B (Fig. 6j). In addition, cathepsin B deficiency diminished mannose-mediated upregulation of tight junctions in colonic epithelial cells (Fig. 6k). These data suggest that mannose prevents DSS-induced mitochondrial dysfunction and tight junction disruption by limiting cathepsin B release from lysosomes.

**Mannose exerts a potential therapeutic effect on mouse colitis**

To further determine the potential therapeutic effects of mannose on DSS-induced colitis, mice were challenged with a high dose of DSS for 3 days and then treated with mesalazine, the first-line treatment for IBD, and mannose alone or in combination from day 4 until the end of the experiment (Fig. 7a). Similar to mesalazine treatment, mannose treatment robustly reversed the body weight loss (Fig. 7b) and colon length shortening (Fig. 7c) in DSS-treated mice. Surprisingly, the combination of mesalazine and mannose exhibited an efficient therapeutic effect, and this finding was supported by pathological analyses (Fig. 7d). Mesalazine and mannose administered either alone or in combination also restored the tight junction disruption in DSS-treated mice (Fig. 7e). The evaluation of lysosomal and mitochondrial function showed that supplementation with mannose and mesalazine significantly suppressed lysosomal cathepsin B release (Fig. 7f) and maintained mitochondrial OXPHOS function (Fig. 7g) in DSS-treated

mice. Taken together, these results verify the therapeutic activity of mannose in the treatment of acute colitis.

## Discussion

Mannose is a simple sugar that is metabolized following its absorption through the intestine[39]. Here, we report that excessive circulating mannose levels in human specimens are indicative of an increased risk of IBD. It is possible that disruption of intestinal epithelial cells results in the dysfunction of mannose metabolism, which might be the main reason for the elevation of circulating mannose. We also identified a beneficial effect of mannose in preventing IBD by modulating the tight junctions. Molecular and cellular studies further revealed that mannose may facilitate mitochondrial functioning in colonic epithelial cells by inhibiting lysosomal cathepsin B activity as well as the assembly of lysosomes (Fig. 8).

The tight junction complex plays a crucial role in the formation and maintenance of the intestinal epithelial barrier. Disordered tight junctions and epithelial permeability have been implicated in the development of IBD[20,40]. Our studies show that mannose can protect against the increased colon permeability and loss of tight junction integrity induced by DSS administration. Indeed, metabolic syndromes often accompany obesity or hyperglycemia and have been linked to the disruption of the intestinal barrier as well as an increased risk of systemic infection[9]. Indeed, the oral administration of mannose in drinking water prevents the development of hyperglycemia in both nonobese diabetic (NOD) mice and high-fat diet-fed mice[15,16]. Previous studies determined that MLCK-dependent MLC phosphorylation is an essential intermediate event in the regulation of tight junctions[21,41]. Our data show that the upregulation of tight junction protein expression in damaged intestinal epithelial cells by mannose involves the MLCK-MLC pathway. Since epithelial transport of glucose by the Na⁺-glucose cotransporter is associated with MLCK expression and MLC phosphorylation[42], the colitis-preventing effect of mannose may also be due to changes in glucose metabolism.

Mitochondria are membrane-bound organelles that maintain cellular energy production through oxidative phosphorylation. Given that most cellular functions, as well as the epithelial barrier maintenance, are energy-dependent[6,8], it is conceivable that mitochondrial dysfunction plays a key role in both the onset and recurrence of IBD. Indeed, the intestinal mucosa of IBD patients has been shown to be in a state of energy deficiency, characterized by low ATP levels and low membrane potential[6]. OXPHOS, a fundamental function of mitochondria, combines electron transport with cellular respiration and ATP synthesis[43]. In epithelial cells, impaired OXPHOS results in the enhanced conversion of guanosine triphosphate (GTP) to cyclic guanosine monophosphate (cGMP), thereby increasing MLCK expression and MLC phosphorylation[44]. A quantity of mannose was phosphorylated into mannose-6-phosphate or used to synthesize glycoproteins after entering the cell, which also produced other molecules, such as lactate, pyruvate, and GDP-mannose. Indeed, the great majority of lactate and pyruvate formed from mannose might be secreted into culture supernatants. Thus, the evaluation of the mannose-associated molecules in the culture supernatants and the production of $CO_2$ will help us understand the fate of mannose, which still needs further

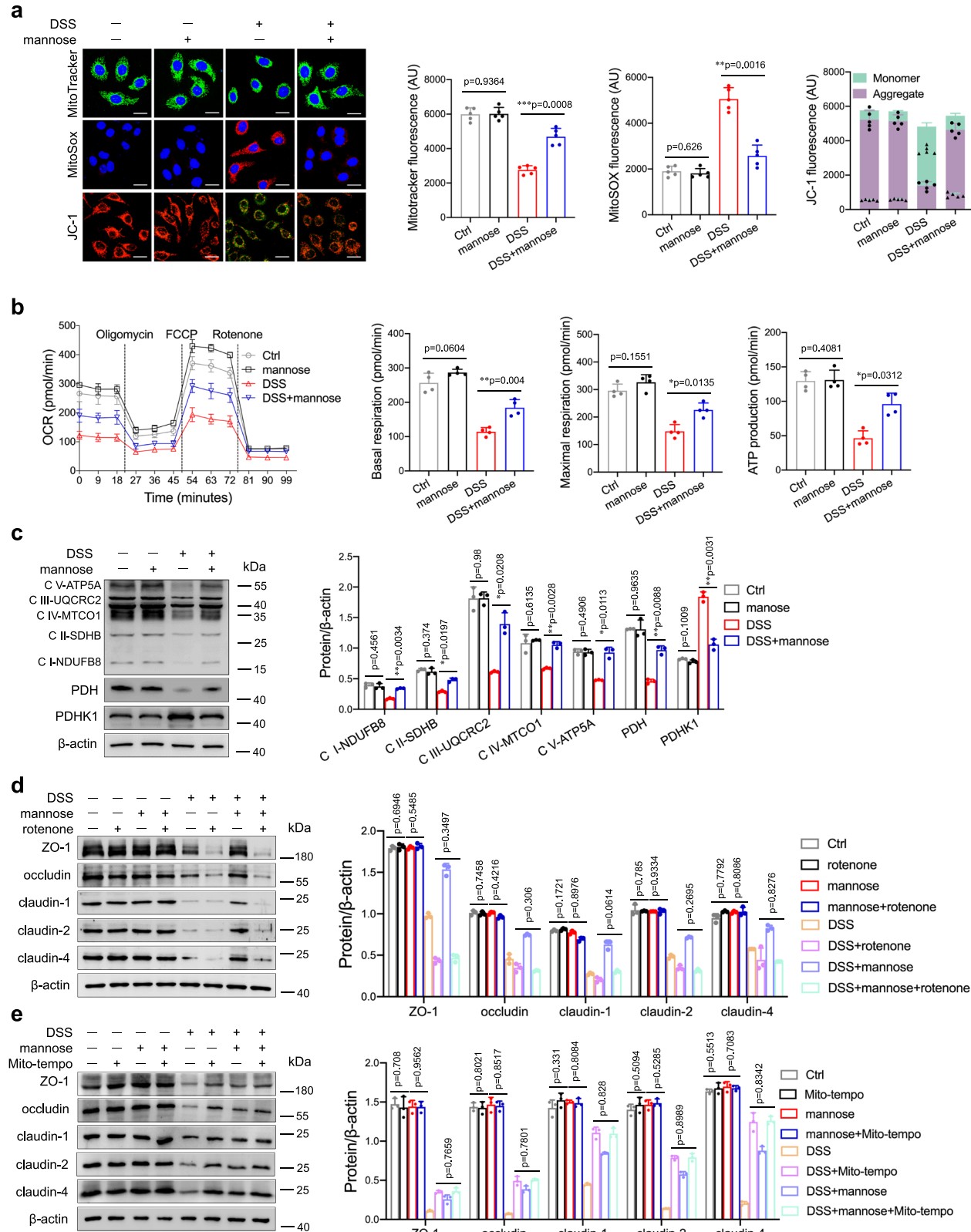

investigation. Mannose-mediated prevention of DSS-induced tight junction disruption is associated with mitochondrial respiration, determined by the in vitro experiments in the presence of rotenone treatment. Also, mito-tempo pretreatment relieved the DSS-induced mice colitis model. This finding is consistent with that of a previous study showing that mannose supplementation of high-fat diet-fed mice increased endurance and maximal $O_2$ consumption[15].

The physical and functional crosstalk between mitochondria and lysosomes may play a significant role in metabolic regulation[45]. Lysosome inhibitor Baf A1 pretreatment significantly reversed the mannose-mediated protection of mitochondria dysfunction, implying the critical role of lysosome in the mannose-induced benefit against intestinal barrier damage. Free polymannose-type oligosaccharides incubated with liver cells have previously been shown to

**Fig. 4 | Mannose protects against DSS-induced epithelial tight junction disruption through its effects on mitochondrial function. a–c** Normal colonic epithelial NCM460 cells were treated with 2.0% DSS in combination with or without mannose (25 mM) for 24 h. **a** MitoTracker was employed to mark mitochondria. Mitochondrial ROS and the mitochondrial membrane potential were measured by MitoSOX and JC-1 probes, respectively. Scale bar = 20 μm. **b** The cellular OCR was evaluated using a Seahorse Bioscience XF24 Extracellular Flux Analyzer. The quantification data of basal OCR, maximal OCR, and ATP production are presented. **c** The expression of OXPHOS proteins in the mitochondrial fractions was determined by western blot analysis. The levels of PDHK1 and PDH were examined by immunoblotting analysis. **d** NCM460 cells were pretreated with the mitochondrial electron transport chain complex I inhibitor rotenone (5 μM), followed by DSS and mannose treatment. The expression of ZO-1, occludin, and claudin-1, 2, and 4 in the cells was determined by western blot analysis. **e** NCM460 cells were cultured with DSS and mannose for 24 h in the presence or absence of mito-tempo (2 μM). The expression of ZO-1, occludin, and claudin-1, 2, and 4 in the cells was determined by western blot analysis. $*P < 0.05$, $**P < 0.01$, and $***P < 0.001$. Data were analyzed by an unpaired Student's $t$-test (**a**–**e**) and shown as means ± SD. Data from one representative experiment of three independent experiments are presented. Source data are provided as a Source Data file.

predominantly accumulate in lysosomes[46]. To quantify mannose uptake, radiolabeled probes have been used effectively[26,47]. We demonstrated that unmetabolized mannose is mainly located in the lysosome of colonic epithelial cells. Indeed, the mannose can be phosphorylated into mannose-6-phosphate and used for glycosylation purposes[48]. Gonzalez et al.[49], reported that mannose accounted for the mannose-induced hexose-6-phosphate accumulation. Torretta et al.[50], revealed that mannose was less efficiently metabolized through glycolysis but accumulated intracellularly as mannose-6-phosphate in macrophages. Our study demonstrated that the majority of intracellular mannose was phosphorylated into mannose-6-phosphate or used to synthesize glycoproteins in colonic epithelial cells. It should be mentioned that mannose-6-phosphate is crucial for the trafficking of lysosomal acid hydrolases into the lysosome[51]. Improper trafficking or lack of certain hydrolases leads to a toxic accumulation of their substrates in the lysosomes. We speculate that both mannose and mannose-related metabolites might work together to influence the activities of lysosomes, which warrants further investigations. Lysosomal storage diseases (LSDs) are rare inherited metabolic diseases that are often fatal. Additionally, lysosomal dysfunction is a potentially causative element in IBD pathogenesis[52]. Recently, several reports have indicated the crucial role played by mitochondrial dysfunction in the pathophysiology of some LSDs, suggesting that lysosomal dysfunction directly impacts mitochondria[53]. Therefore, targeting the lysosomal pathway may help restore mitochondrial function and improve cell health in individuals with metabolic diseases. However, the exact mechanisms governing the functional crosstalk between mitochondria and lysosomes remain to be elucidated.

Several lysosomal proteases, such as cathepsins, are released into the cytosol of target cells as a result of lysosomal disruption[32,33]. Among all cathepsins, cathepsin B is one of the most abundant cysteine proteases in lysosomes and has been associated with mitochondrial damage[54]. Consistently, our current work shows that lysosomal cathepsin B is responsible for mitochondrial damage in colonic epithelial cells. Moreover, the mannose-mediated protection against mitochondrial dysfunction involves its inhibition of cathepsin B release. It is worthy to mention that the release of cathepsin B from the lysosome to the cytosolic was associated with the maintenance of mannose in the lysosome. We found that the absence of cathepsin B abrogated the mannose-mediated protection of mitochondria and upregulation of PDH expression. Intriguingly, PDH (mitochondrial oxidative phosphorylation key enzyme) disruption eliminated the mannose-mediated protection against mitochondrial damage without impacting the release of lysosomal cathepsin B. It remains to be determined whether lysosomal cathepsin B regulates mitochondrial functions by controlling mitochondrial PDH.

In summary, our study demonstrates that oral administration of mannose attenuates both chemically-induced and spontaneous colitis mouse models by protecting intestinal epithelial barrier function. The sustained release allows delivery of a specific drug at a programmed rate, controlling the drug release for a prolonged period. Therefore, sustained-release mannose may be used safely and effectively in patients, which warrants further investigation. Mechanistic studies revealed that mannose maintains lysosomal function, thereby promoting mitochondrial metabolism and the assembly of epithelial tight junctions. Most importantly, we have provided experimental evidence showing that mannose combined with mesalazine exhibits a synergistic therapeutic effect on colitis in mice, suggesting that mannose may be useful as a supplement in patients with IBD. However, the direct mechanism of action of mannose on ulcerative colitis remains unclear and requires future work to elucidate.

## Methods

### Participants

All patients with IBD were recruited from the Department of Gastroenterology at Nanfang Hospital, affiliated with Southern Medical University (Guangzhou, China). The diagnosis of CD or UC was based on clinical, radiologic, and endoscopic examinations and histologic findings. The baseline characteristics are described in Supplemental Table 1. All patients provided informed consent, and the study was approved by the Ethics Committee of Nanfang Hospital, Southern Medical University.

### Mice and treatments

Male IL-10-deficient ($IL\text{-}10^{-/-}$) C57BL/6 J mice (12 weeks old, 22–25 g) were obtained from the Shanghai Research Center for Model Organisms (Shanghai, China). Male C57BL/6 J mice (8 weeks old, 22–25 g) were purchased from the Animal Institute of Southern Medical University (Guangzhou, China). Mice were maintained under SPF conditions. All animal experiments in this study were approved by the Welfare and Ethical Committee for Experimental Animal Care of Southern Medical University.

**DSS-induced colitis.** Chemically-induced colitis was established by providing the mice with 3.0% (wt/vol) DSS (MP Biomedicals, Santa Ana, CA, USA, molecular weight of 36,000–50,000) dissolved in drinking water for 7 days. Starting from the first day of the DSS challenge, mannose (Sigma-Aldrich) or glucose (Sigma-Aldrich) was administered orally at the indicated dose (5, 50, 500, and 5000 μg/g/d) per day. Furthermore, 8 a.m. and 8 p.m. were chosen as a set time for feeding, with a duration of 7 days.

**Mito-tempo treatment.** Mice were treated with mito-tempo (Merck, HY-112879) at a dose of 2 mg/kg by i.p injection every day. Mice were randomly divided into eight groups: control group (mice received regular drinking water), mito-tempo group (mice received regular drinking water together with i.p injection of 2 mg/kg of mito-tempo), mannose group (mice received regular water together with administration by gavage of 500 μg/g/d mannose), mito-tempo + mannose group (mice received regular drinking water together with i.p injection of 2 mg/kg of mito-tempo and administration by gavage of 500 μg/g/d mannose), DSS model group (mice received 3% DSS in drinking water), DSS + mito-tempo group (mice received 3% DSS in drinking water together with i.p injection of 2 mg/kg of mito-tempo), DSS + mannose group (mice received 3% DSS in drinking water together with administration by gavage of 500 μg/g/d mannose), DSS + mito-tempo +mannose group (mice received 3% DSS in drinking water together

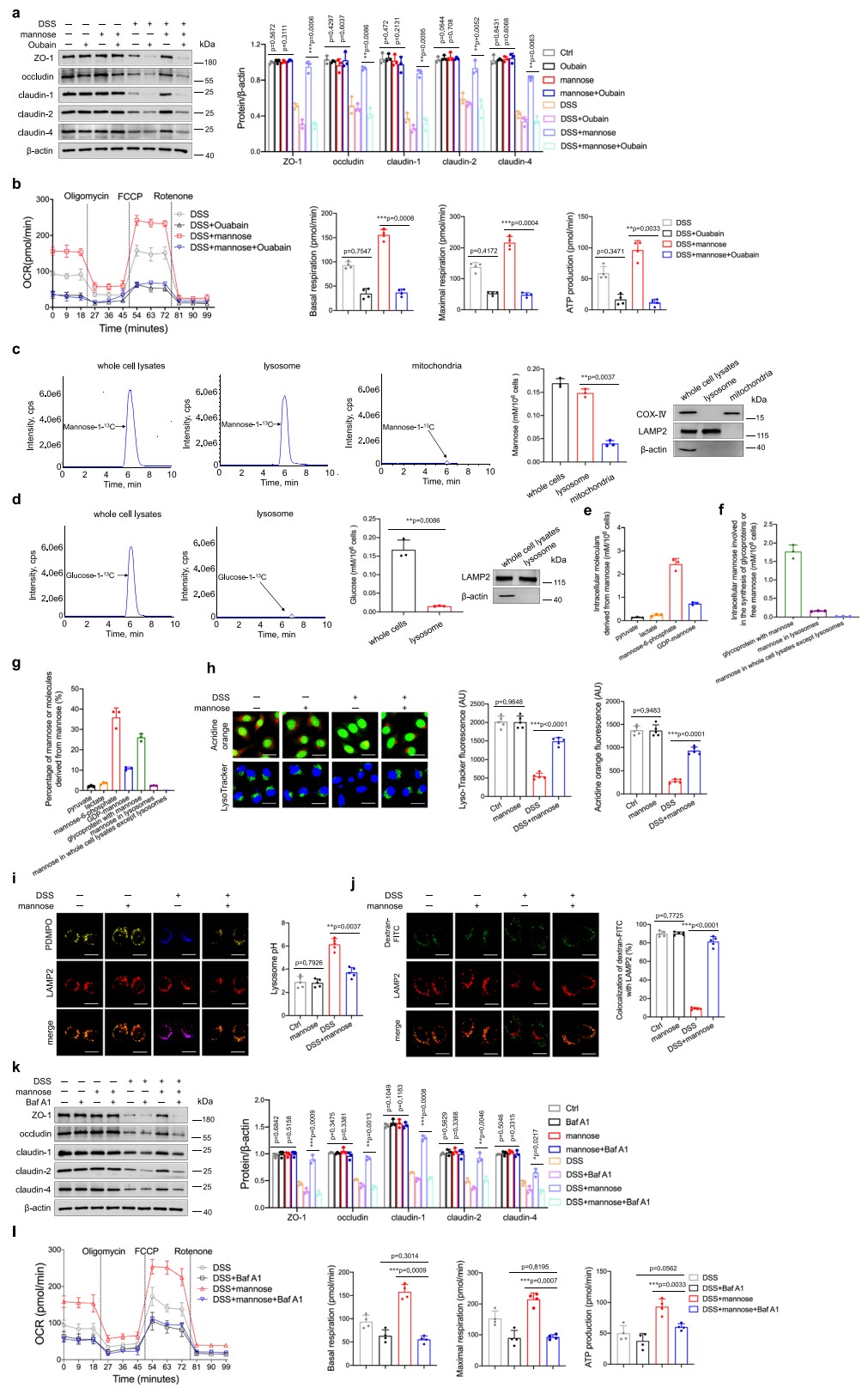

with i.p injection of 2 mg/kg of mito-tempo and administration by gavage of 500 µg/g/d mannose).

**Bafilomycin A1 treatment.** Mice were treated with bafilomycin A1(Baf A1) (Selleck, S1413) at a dose of 2 mg/kg by i.p injection every day. Mice were randomly divided into eight groups: control group (mice received regular drinking water), Baf A1 group (mice received regular

drinking water together with i.p injection of 2 mg/kg of Baf A1), mannose group (mice received regular water together with administration by gavage of 500 µg/g/d mannose), Baf A1 + mannose group (mice received regular drinking water together with i.p injection of 2 mg/kg of Baf A1 and administration by gavage of 500 µg/g/d mannose), DSS model group (mice received 3% DSS in drinking water), DSS + Baf A1 group (mice received 3% DSS in drinking water together with i.p

**Fig. 5 | Mannose rescues the alterations in epithelial cell lysosomal integrity induced by DSS. a, b** NCM460 cells were pretreated with ouabain (1 μM) for 4 h, followed by DSS and mannose administration. **a** The protein levels of the tight junction were evaluated by immunoblotting analysis. **b** Seahorse analysis of OCR in the cells is presented. The OCR of basal and maximal mitochondrial respiration, as well as ATP production, were analyzed. **c, d** NCM460 cells were incubated with 25 mM mannose-1-[13]C or glucose-1-[13]C for 24 h, respectively. **c** The amount of mannose in the lysosomal, mitochondrial fraction, and the whole cell lysates were determined by LC−MS/MS analysis. **d** The amount of glucose in the lysosomal fraction and the whole cell lysates were determined. The markers of isolated lysosomal fraction and mitochondrial fraction were evaluated by immunoblotting analysis. **e–g** NCM460 cells were incubated with 25 mM mannose-1-[13]C for 24 h. **e** The amount of mannose-1-[13]C derived molecules was detected by LC−MS analysis. **f** The amount of mannose-1-[13]C involved in glycoproteins and free mannose was detected. **g** The percentage of mannose-1-[13]C derived molecules, mannose involved in glycoproteins, and free mannose was calculated. **h, i** NCM460 cells were stimulated with DSS with or without mannose for 24 h, followed by acridine orange (Scale bar = 20 μm) (**h**). Lysosomal acidification was assessed by PDMPO staining. Scale bar = 20 μm (**i**). Fluorescence analysis of lysosomal integrity using FITC-conjugated dextran. Scale bar = 20 μm (**j**). Fluorescence intensity was analyzed by ImageJ software. **k, l** NCM460 cells were cultured with DSS and mannose for 24 h with or without bafilomycin A1 (1 μM), the expression of the tight junction was determined by immunoblotting analysis (**k**). Seahorse analysis of OCR in the cells is presented. The basal and maximal mitochondrial respiration, as well as ATP production were analyzed (**l**). **p < 0.01, ***p < 0.001. Data from one representative experiment of three independent experiments are presented. Two-side, unpaired t-test for (**a–d**, **h–l**). Source data are provided as a Source Data file.

injection of 2 mg/kg of Baf A1), DSS + mannose group (mice received 3% DSS in drinking water together with administration by gavage of 500 μg/g/d mannose), DSS + Baf A1 + mannose group (mice received 3% DSS in drinking water together with i.p injection of 2 mg/kg of Baf A1 and administration by gavage of 500 μg/g/d mannose).

**Mesalazine treatment.** To evaluate the therapeutic effect of mannose on colitis, mice were given a high dose of DSS (5.0%, wt/vol) for 3 days, followed by mannose (5 mg/kg per day) or glucose (5 mg/kg per day) and mesalazine (Adisha, H20143164, 1 mg/kg per day) alone or in combination for another 4 days. Mice were randomly divided into six groups: DSS model group (mice received 5.0% DSS in drinking water for three days), DSS + glucose group (mice received intragastric (i.g) administration of 500 μg/g/d of glucose after receiving 5.0% DSS in drinking water for three days), DSS + mannose group (mice received intragastric (i.g) administration of 500 μg/g/d of mannose after receiving 5.0% DSS in drinking water for 3 days), DSS + mesalazine group (mice received intragastric (i.g) administration of 1 mg/kg/d of mesalazine after receiving 5.0% DSS in drinking water for 3 days), DSS + glucose+mesalazine group (mice received intragastric (i.g) administration of 500 μg/g/d of glucose plus 1 mg/kg/d of mesalazine after receiving 5.0% DSS in drinking water for three days), DSS + mannose+mesalazine group (mice received intragastric (i.g) administration of 500 μg/g/d of mannose plus 1 mg/kg/d of mesalazine after receiving 5.0% DSS in drinking water for three days).

**Spontaneous chronic colitis.** To determine the effect of mannose on spontaneous chronic colitis, 15-week-old IL-10[−/−] mice in the treatment group were treated with glucose (1.0%, wt/vol) or mannose (1.0%, wt/vol) for 4 weeks.

**Cells and stimulation conditions**
Human colonic epithelial NCM460 cells were cultured in RPMI 1640 medium with no glucose (Invitrogen, Carlsbad, CA, USA, 11879020) supplemented with 100 U/ml penicillin, 100 μg/ml streptomycin, 1 mM glucose, and 10% fetal bovine serum (FBS). The cells were treated with 2.0% DSS (MP Biomedicals, Santa Ana, CA, USA, molecular weight of 36,000–50,000) in the presence or absence of mannose (25 mM) for 24 h. To eliminate the effect of the mannose transporter, cells were pretreated with ouabain (1 μM, Selleck, S4016) for 4 h before incubation. Rotenone (5 μM, Merck, HY-B1756) was added simultaneously with DSS and mannose to inhibit mitochondrial respiratory activity. In some cases, protease inhibitors, such as the cathepsin B inhibitor CA-074 (10 μM, Selleck, HY-103350), cysteine protease inhibitor aloxistatin (1 μM, Selleck, S7393), serine protease inhibitor nafamostat mesylate (5 μM, Selleck, S1386), and aspartic protease inhibitor pepstatin A (10 μM, Selleck, S7381), were added simultaneously with DSS and mannose. Bafilomycin A1 (1 μM, Selleck, S1413) was added simultaneously with DSS and mannose for 24 h. mito-tempo (2 μM, Merck, HY-112879), a mitochondrial-targeted antioxidant, was used to stimulate cells along with DSS plus mannose stimulation.

**Quantitative RT-PCR**
Total RNA was extracted from mouse colon tissue or NCM460 cells using TRIzol reagent (TransGene Biotech, Beijing, China) and transcribed into cDNA using TranScript All-in-One First-Strand cDNA Synthesis SuperMix (TransGene Biotech). Real-time PCR analysis using TransStart Tip Green qPCR SuperMix (TransGene Biotech) was performed on a 7900HT fast real-time PCR system (Applied Biosystems, San Francisco, CA, USA). The target gene expression levels were normalized to the expression of β-actin in the same samples.

**Immunoprecipitation and immunoblotting**
Protein samples were loaded on SDS-polyacrylamide gels, separated by electrophoresis, and then transferred to polyvinylidene fluoride (PVDF) membranes (Millipore, Billerica, MA, USA). After blocking with bovine serum albumin (BSA, 5%) for 1 h at room temperature, the membranes were incubated overnight at 4 °C with primary antibodies. Subsequently, the membranes were incubated with the corresponding horseradish peroxidase-conjugated secondary antibody for 1 h at room temperature. For immunoprecipitation, whole cell lysates were incubated with 1 μg of antibody and protein A/G agarose (Santa Cruz Biotechnology, Santa Cruz, CA, USA) at 4 °C overnight. The eluted immunoprecipitates were resolved via SDS-PAGE, and the associations between proteins of interest were examined using specific antibodies.

**Immunohistochemical and immunofluorescence staining**
For immunohistochemical staining, antigen retrieval was performed in citrate buffer (pH 6.0) (Sigma-Aldrich, C2488) at 120 °C for 10 min, and endogenous peroxidase activity was blocked by exposure to 3.0% $H_2O_2$ for 15 min. Sections were then incubated with primary antibodies at 4 °C overnight. Immunoreactivity was detected using the corresponding HRP-conjugated secondary antibody and visualized using a Pierce™ DAB Substrate kit (Thermo Fisher, Carlsbad, CA, USA).

For immunofluorescence staining, cells were grown in confocal dishes, fixed in 4.0% formaldehyde for 15 min at room temperature, and permeabilized with 0.25% Triton X-100 for 10 min at room temperature. After blocking with 5.0% FBS for 1 h, cells were incubated with primary antibodies overnight at 4 °C, rinsed, and incubated with fluorescently labeled secondary antibodies for 1 h in the dark. Finally, cells were counterstained with Hoechst 33342 (Cell Signaling Technology, 4082). The quantitative colocalization was calculated using the JACoP plugin in single Z-stack sections of deconvolved images. The nonspecific signal correction was performed using ImageJ with the subtract background plugin. All images were subjected to identical post-acquisition processing.

**Histopathological assessment**
For pathological assessment, the H&E stained sections were evaluated by a blinded pathologist following the criteria of histological score previously reported[55]. The colitis score (maximum = 8) was the sum of

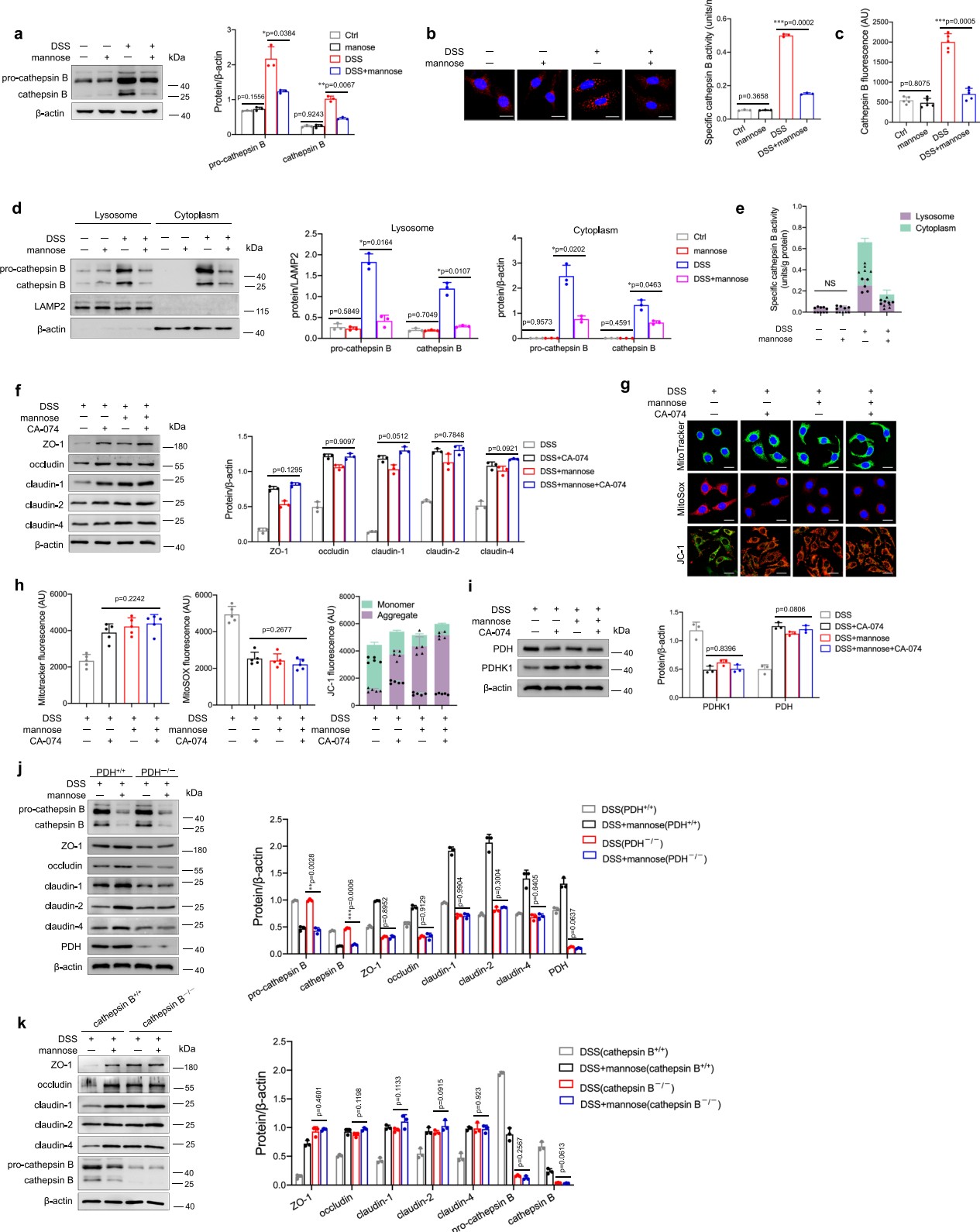

the following two features: Epithelium (0: Normal morphology; 1: Loss of goblet cells, 2: Loss of goblet cells in large areas; 3: Loss of crypts; 4: Loss of crypts in large areas), and infiltration (0: No infiltrate; 1: Infiltrate around crypt basis; 2: Infiltrate reaching to lamina muscularis mucosae; 3: Extensive infiltration reaching the lamina muscularis mucosae and thickening of the mucosa with abundant edema; 4: Infiltration of the lamina submucosa).

## Measurement of serum mouse albumin

Serum albumin levels were measured by a mouse-specific albumin ELISA kit (Abcam, ab207620) according to the manufacturer's instructions. In brief, the sample and the antibody cocktail were added to each well and incubated for 1 h at room temperature After washing the wells three times with washing buffer, 100 µl of TMB development solution was added and incubated for 10 min in the dark. Then, 100 µl

**Fig. 6 | Mannose protects mitochondrial function by reducing the abnormal release of cathepsin B from lysosomes. a**–**e** NCM460 cells were stimulated with 2.0% DSS with or without mannose for 24 h, and the protein level of cathepsin B in the cells was determined by western blotting (**a**) and immunofluorescence staining (Scale bar = 20 μm) (**b**). The activity of cathepsin B in the culture medium was also evaluated (**c**). The expression levels (**d**) and activities (**e**) of cathepsin B in the lysosomal fraction and other remaining cytoplasm of NCM460 cells were assessed. **f**–**i** NCM460 cells were pretreated with CA-074 (10 μM), a selective inhibitor of cathepsin B in live cells, before DSS and mannose stimulation. **f** The expression of tight junction proteins in the cells was determined by western blot analysis. **g**, **h** Mitochondrial morphology, mitochondrial ROS production, and mitochondrial membrane potential were detected by MitoTracker, MitoSOX, and JC-1

probes, respectively. Scale bar = 20 μm. ImageJ software was used to analyze the images. **i** The levels of mitochondrial PDHK1 and PDH were evaluated by immunoblotting analysis. **j** PDH⁺ᐟ⁺ and PDH⁻ᐟ⁻ NCM460 cells were stimulated with 2.0% DSS with or without mannose for 24 h, then the expression of activated cathepsin B and tight junction proteins was determined by western blotting analysis. **k** Cathepsin B⁺ᐟ⁺ and Cathepsin B⁻ᐟ⁻ NCM460 cells were stimulated with 2.0% DSS with or without mannose for 24 h, then the expression of tight junction proteins was determined by western blotting analysis. *$p < 0.05$, **$p < 0.01$, ***$p < 0.001$. Data were analyzed by an unpaired Student's $t$-test (**a**–**k**) and shown as means ± SD. Data from one representative experiment of three independent experiments are presented. Source data are provided as a Source Data file.

of stop solution was added to each well and the OD value was recorded at 450 nm.

## Measurement of fecal α1-antitrypsin

Fecal samples were dissolved in PBS to a concentration of ~500–1000 ng/ml. Alpha-1-antitrypsin levels were quantified using a mouse-specific α1-antitrypsin SimpleStep ELISA Kit (Abcam, ab267809) according to the manufacturer's instructions. Briefly, the sample and the antibody cocktail were added to each well and incubated for 1 h at room temperature. After washing three times with the washing buffer, 100 μl of TMB development solution was added to each well and incubated for 10 min in the dark. Then, 100 μl of stop solution was added and the OD value was recorded at 450 nm.

## Measurement of mannose

**Measurement of circulating mannose.** An aliquot (50 μl) of standard, quality control (QC), a human plasma sample, or mouse serum sample was first mixed with 200 μl methanol-acetonitrile-water (v:v:v, 2:2:1) mixed solvent containing 5 μg/ml D-mannose-1-¹³C (Sigma-Aldrich) as an internal standard. The mixture was then vortex mixed for 30 s and centrifuged for 30 min at 20,000 × g at room temperature. After centrifugation, an aliquot (10 μl) of the supernatant was taken for LC–MS/MS analysis. The chromatographic separation was achieved by a Shimadzu Nexera UHPLC LC-30A equipped with an AB SCIEX Triple Quad™ 4500, crosslinked HPLC column (3.0mml.D.×100 mm, 1.7 μm, waters) with a flow rate of 0.35 ml/min. The LC–MS/MS system was controlled, and data was acquired by Analyst software version 1.6.

**Measurement of mannose in cells.** After incubated with D-mannose-1-¹³C (Sigma-Aldrich, 25 mM) for 12 and 24 h, the lysosomal fraction, mitochondrial fraction, and the whole cell lysates of NCM460 cells were isolated. The fraction was extracted with 1 ml ethanol: water (1:1). The mixture was then centrifuged for 30 min at 20,000 × g at room temperature. After centrifugation, an aliquot (10 μl) of the supernatant was taken for LC–MS/MS analysis. The same method was used for the determination of glucose in cells.

## Metabolic extraction of intracellular metabolites

For ¹³C-mannose-tracing assays, NCM460 cells were treated in glucose-free RPMI 1640 medium (11879, Thermo Fisher Scientific) supplemented with 100 U/ml penicillin, 100 μg/ml streptomycin, 1 mM glucose, and 10% FBS and 25 mM ¹³C-labeled-mannose. Cells were rapidly washed with ice-cold PBS three times and extracted with 500 μl extraction solvent (50% methanol, 30% acetonitrile, and 20% water) after 24 h. Cells were then centrifuged at 16,000 × g for 30 min at 4 °C, and the supernatants were assessed by LC–MS analysis. Exactive Orbitrap mass spectrometer (Thermo Fisher Scientific) was used together with a Thermo Fisher Scientific Accela HPLC system. The HPLC setup consisted of a ZIC-pHILIC column (SeQuant, 150 mm × 2.1 mm, 5 μm, Merck KGaA) with a ZIC-pHILIC guard column (SeQuant, 20 mm × 2.1 mm) and an initial mobile phase of 20% 20 mM

ammonium carbonate, pH 9.4 and 80% acetonitrile. Cell extracts (5 μl) were injected, and the metabolites were separated over a 15-min mobile phase gradient. All metabolites were detected across a mass range of 75–1000 m/z using the Exactive mass spectrometer at a resolution of 25,000 (at 200 m/z), with electrospray ionization and polarity switching to enable both positive and negative ions to be determined in the same run over a total analysis time of 23 min. Lock masses were used, and the mass accuracy obtained for all metabolites was below 5 p.p.m. Data were acquired with Thermo LCquan 2.7 (Thermo Fisher Scientific) software. To examine whether mannose is involved in glycoprotein synthesis, protein lysate of NCM460 cells were deglycosylated by Protein Deglycosylation Mix 2 (New England Bio-labs), according to the non-denaturing reaction protocol distributed by the manufacturer. Briefly, 100 μg of protein lysate of NCM460 cells, 5 μl of 10 × deglycosylation mix buffer, and 5 μl of protein deglycosylation mix 2 were mixed to a total volume of 50 μl and incubated at 25 °C for 30 min, followed by overnight incubation at 37 °C. The supernatant was hydrolyzed by 2 M trifluoroacetic acid at 100 °C for 16 h. An aliquot (10 μl) of the supernatant was taken for LC–MS/MS analysis.

## FITC-dextran permeability assay

Intestinal permeability was assessed by oral administration of fluorescein isothiocyanate-dextran with a molecular weight of 3000–5000 Da (FD4, Sigma-Aldrich). At the beginning of the experiment, food and water were forbidden for 4 h, and mice were subsequently gavage fed with FITC-dextran solution at 500 μg/g body weight. Serum was collected 4 h post-feeding, and FITC-dextran measurements were performed in duplicate by fluorometry (excitation, 490 nm; emission, 530 nm; Cytofluor 2300, Millipore).

## TER measurement

Confluent monolayers of NCM460 cells or isolated primary colonic epithelial cells were grown in 24-well Transwell chambers (polycarbonate membrane, filter pore size 0.4 μm, area 0.33 cm²; Costar) for 24 h, and then TER was measured at 37 °C using an Epithelial Volt Ohm Meter (Millipore, Billerica, MA, USA). TER values were calculated by subtracting the blank filter and by multiplying the surface area of the filter. All the measurements were performed in triplicate.

## Isolation of primary colonic epithelial cells

The primary colonic epithelial cells of mice were purified as previously described[34]. Briefly, the colonic tissues were cut into pieces and trained at 37 °C in Dulbecco's modified Eagle medium (DMEM) containing 5% FBS and 1 mM dithiothreitol (DTT) for 30 min. The remaining tissue was induced in 30 ml phosphate-buffered saline (PBS) containing 1.5 mM EDTA for an additional 10 min The supernatants were filtered, centrifuged for 5 min at 400 × g, and the cell pellet was resuspended in DMEM containing 5% FBS. Finally, the primary colonic epithelial cell suspension was purified by centrifugation through a 25%/40% discontinuous percoll gradient at 600 × g for 30 min.

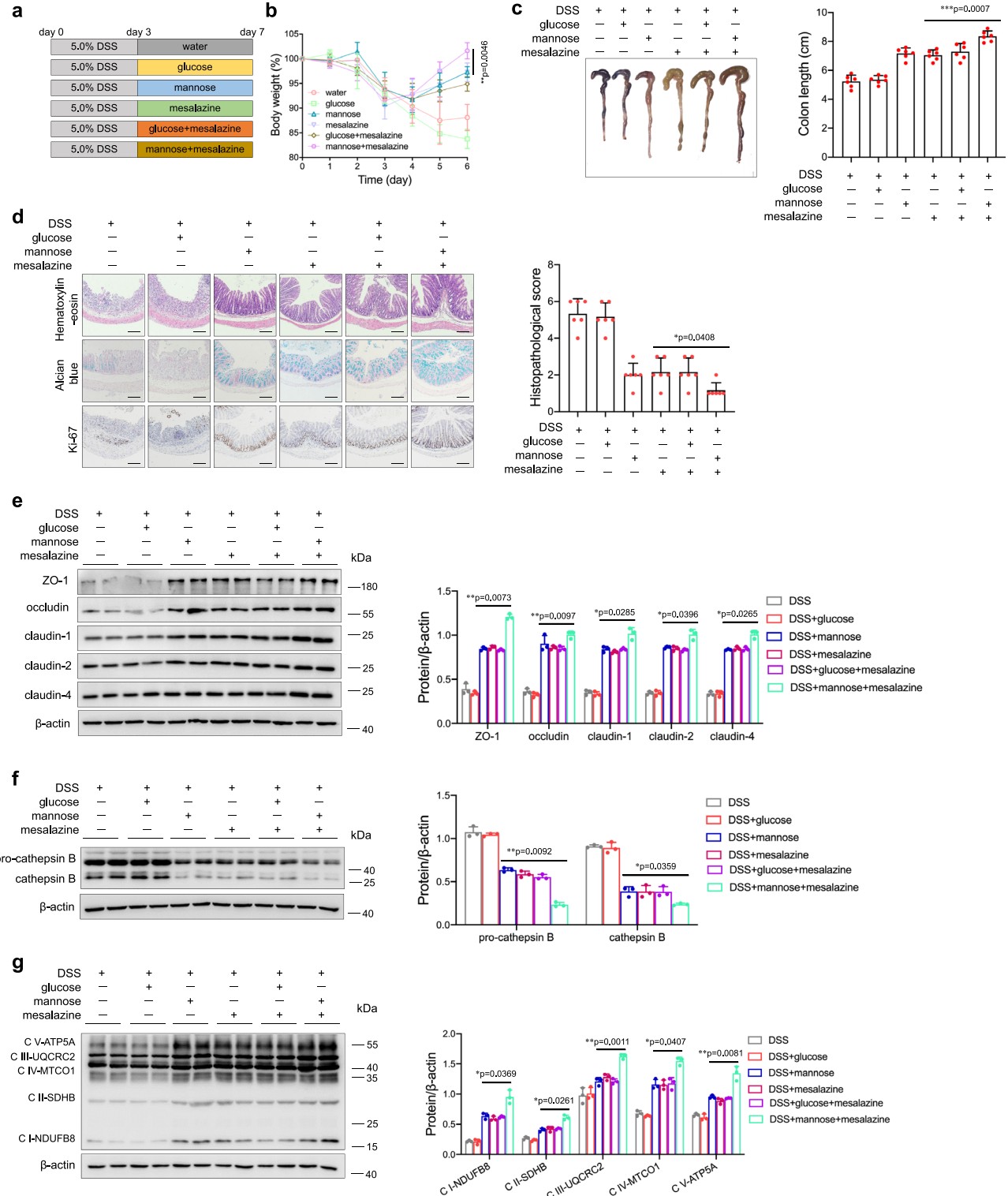

**Fig. 7 | Mannose exerts a therapeutic effect against DSS-induced colitis.**
**a** Schematic overview of the experimental design. **b**–**g** Mice (male, $n = 6$ per group) were administered 5.0% DSS from day 0 to day 3, followed by treatment with mannose and mesalazine either alone or in combination from day 4 to day 7. **b** Changes in body weights during the experiments were recorded. **c** Colon tissues were isolated on the last day of the experiment. A representative photograph of the colon tissue from each group is presented, and the colon length was measured. Scale bar = 1 cm. **d** Histological analysis of mouse colon tissue taken on the last day of the experiment was performed by H&E, alcian blue, and Ki67 staining. Scale

bar = 100 μm. Histological scores of the DSS-induced colitis were evaluated. **e** The expression of the indicated tight junction proteins in the mouse colon was examined on the last day of the experiment by immunoblotting analysis. **f** The protein level of activated cathepsin B in the colon tissues was assessed by western blotting analysis. **g** The expression of mitochondrial OXPHOS proteins in colon tissues was determined by western blot analysis compared to the group treated with DSS only. $*p < 0.05$, $**p < 0.01$. Data were analyzed by an unpaired Student's $t$-test and shown as means ± SD. Data from one representative experiment of three independent experiments are presented. Source data are provided as a Source Data file.

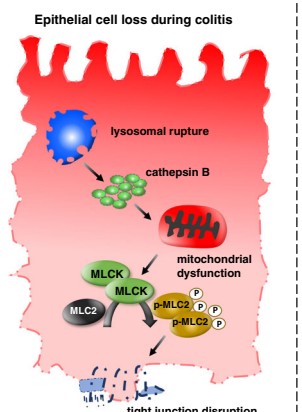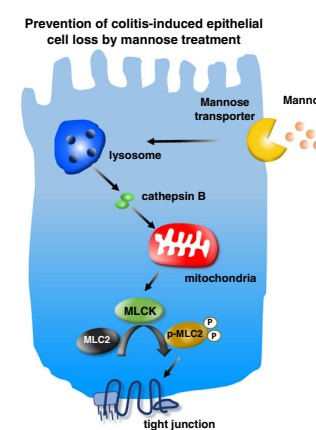

Epithelial cell loss during colitis

Prevention of colitis-induced epithelial cell loss by mannose treatment

**Fig. 8 | A model of the proposed mechanism by which mannose protects against colitis.** The proposed scheme shows that mannose protects lysosomal integrity and subsequently limits the release of cathepsin B, leading to the prevention of mitochondrial dysfunction and MLCK-induced tight junction disruption in response to intestinal epithelial cell damage during colitis.

## Cell viability assay

For apoptosis quantification by annexin V, NCM460 cells were harvested and stained with annexin V-FITC and propidium iodide according to the manufacturer's instructions (Annexin V-APC/7-AAD apoptosis kit, Multi Sciences, China). In brief, the cells were washed with PBS and subsequently incubated for 5 min at room temperature in the dark in 500 μl of 1× binding buffer containing 5 μl of Annexin V-PE and 10 μl of 7-AAD. The cells were acquired and analyzed in the Dakewe EXFLOW-206.

## Mitochondrial respiration measurements

Oxygen consumption rate (OCR) was analyzed using the Seahorse Bioscience Extracellular Flux Analyzer (XF96) (Seahorse Biosciences, MA, USA). After exposure to several inhibitors, the media of NCM460 cells was removed and replaced by XF basic media 30 min prior to the measurement of oxygen consumption rate (OCR). Several drugs were added in sequence, including oligomycin (1 μg/mL), FCCP (1 μM), and rotenone (1 μM) + antimycin A (10 μM). Basal OCR was determined before any following operations, while non-mitochondrial OCR was defined as the value after the injection of the last drug. The rate of respiration-driven ATP synthesis and proton leak-driven respiration was measured after the addition of oligomycin (1 μg/ml). The difference between this rate and basal OCR was defined as ATP-linked OCR, and similarly, Proton leak OCR was defined as the gap between this rate and non-mitochondrial OCR. After two measurement cycles, 1 μM of the FCCP was added, and the rate was determined. The difference between which and non-mitochondrial OCR was defined as maximal OCR in the same way. After a further two measurement cycles, 1 μM rotenone was added to block complex I in addition to 10 μM antimycin A to inhibit complex III, and thus non-mitochondria was determined. Each value of OCR was normalized by the amount of cellular protein content in each well.

## Mitochondrial function assays

The cells were washed twice with PBS and labeled at 37 °C for 30 min with the following fluorescent probes: 400 nM Mito-Tracker Green (Ex490 nm/Em516 nm), 2.5 μM MitoSOX Red (specifically detects mitochondrial $O_2$), and 10 μM JC-1 (a sensitive marker of mitochondrial membrane potential). Fluorescence was detected on a Nikon A1R scanning laser confocal microscope (Nikon Corporation, Tokyo, Japan). Quantification was performed in images after appropriate thresholding using the ImageJ software (NIH Image).

## Lysosomal function assays

**LysoTracker staining.** NCM460 cells were incubated with LysoTracker Green DND-26 (1 μM) in culture media at 37 °C for 30 min. Hoechst 33342 (Cell Signaling Technology) was used to stain the cells for another 20 min.

**Acridine orange staining.** Cells were washed three times with PBS and stained with 2.5 μg/ml acridine orange (Sigma-Aldrich) for 15 min.

**PDMPO staining.** After being washed by PBS three times, NCM460 cells were incubated with PDMPO (1 μM, AAT Bioquest, 21204) in culture media at 37 °C for 30 min and then fixed in 4.0% formaldehyde for 15 min at room temperature and permeabilized with 0.25% Triton X-100 for 10 min at room temperature. After blocking with 5.0% FBS for 1 h, cells were incubated with anti-LAMP2 antibody overnight at 4 °C, rinsed, and incubated with PE-labeled secondary antibodies for 1 h in the dark.

**FITC-dextran measurement.** After being washed by PBS three times, NCM460 cells were incubated with fluorescein isothiocyanate-dextran (70 kDa, Merck, 46945) for 4 h and then fixed in 4.0% formaldehyde for 15 min at room temperature and permeabilized with 0.25% Triton X-100 for 10 min at room temperature. After blocking with 5.0% FBS for 1 h, cells were incubated with anti-LAMP2 antibody overnight at 4 °C, rinsed, and incubated with PE-labeled secondary antibodies for 1 h in the dark. For immunofluorescence and direct fluorescence quantifications of colocalization, numbers of puncta per cell and the ratio were performed blinded. Quantification was performed in images after appropriate thresholding using the ImageJ software. The quantitative colocalization was calculated using the JACoP plugin in single Z-stack sections of deconvolved images. The nonspecific signal correction was performed using ImageJ software with the subtract background plugin. All images were subjected to identical post-acquisition processing.

## Lysosome and mitochondria isolation

Lysosomes were isolated by a lysosome isolation kit (Abcam, ab234047) according to the manufacturer's protocol. In brief, the cells were isolated in ice-cold lysosome isolation buffer for 2 min and homogenized. The supernatant was collected by centrifugation (500 × g, 10 min) at 4 °C and layered onto a discontinuous density gradient. The lysosomes were further isolated using an ultracentrifuge for 2 h at 145,000 × g at 4 °C.

Mitochondria were isolated by a mitochondria isolation kit for the cells (Abcam, ab110170) according to the manufacturer's instruction. Briefly, the cells were collected and suspended in Reagent A. Then, the cells were homogenized and centrifuged at 1000 × g at 4 °C for 10 minutes. The supernatant was saved as SN1. The pellet was resuspended with Reagent B. After homogenization and centrifugation, the supernatant was collected as SN2. SN1 and SN2 were combined and centrifuged at 12,000 × g at 4 °C for 15 min. Subsequently, the pellet was resuspended in Reagent C supplemented with protease inhibitors and used for further analysis.

## Measurement of cathepsin B activity

Cathepsin B activity was measured in the culture medium or in tissue homogenate using the fluorogenic substrate N-Suc-Leu-Leu-Val-Tyr7-AMC. Excitation at 380 nm and emission at 440 nm were used for activity determinations.

## Generation of gene knockout cells with CRISPR/Cas9

NCM460 cells were transfected with 2 μg of PDH CRISPR/Cas9 KO plasmid or cathepsin B CRISPR/Cas9 KO plasmid (Santa Cruz Biotechnology) using UltraCruz® transfection reagent (Santa Cruz Biotechnology) according to the manufacturer's instructions. Twenty-

four hours after transfection, the expression levels of PDH and cathepsin B in the cells were assessed by immunoblotting analysis.

### Statistical analysis

All results were expressed as mean ± SD. Statistical significance between two groups was evaluated using the Student's $t$-test, while comparisons of multiple groups were assessed by two-way analysis of variance (ANOVA), followed by Student-Newman–Keul's test. $p < 0.05$ was considered significant.

### Reporting summary

Further information on research design is available in the Nature Research Reporting Summary linked to this article.

## Data availability

All data supporting the findings described in the manuscript are available in the article and in the supplementary information and from the corresponding author upon reasonable request. Source data are provided with this paper.

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

## Acknowledgements

This work was supported in part by the National Natural Science Foundation of China (grant nos.: 82071781, 81873872, 81771771, and 81930041), Science and Technology Planning Project of Guangzhou (grant no.: 202002030160 and 202102020111), Innovation team of chronic kidney disease with integrated traditional Chinese and Western Medicine (grant no.: 2019KCXTD014).

## Author contributions

D.Z., Z.C., Q.W., and LD designed research. L.D., J.X., Y.W., H.J., K.C., D.L., J.W., Y.L., J.H., L.Z., X.L., and X.Z. performed experiments. L.D., J.Z., X.-Y.W., Q.W., and D.Z. analyzed the data. D.Z., Z.C., X.-Y.W., and L.D. wrote the manuscript.

## Competing interests

The authors declare no competing interests.
