## [Peer Review File · Nature Communications]

Mannose ameliorates experimental colitis by protecting intestinal barrier integrityReviewers' comments:

Reviewer #1 (Remarks to the Author):

In this study, Dong et al demonstrated that mannose regulates intestinal homeostasis through promoting epithelial barrier function. They showed nicely that mannose protected colitis in DSS model and IL-10ko model, which was mediated by prevention of mitochondrial dysfunction and tight junction disruption. The study was well designed and performed with comprehensive approaches. The conclusion was supported by major results. However, some concerns need be addressed to further improve the quality of the manuscript.

- 1) Fig 1G and I, please show the pathological scores. Fig 1J, although the DSS dose response is interesting for increase of mannose in plasma, what drives such increase? did DSS in low doses cause colitis or disruption of epithelial barrier function?
- 2) Fig 2A, need show pathological scores. 2G and H, need show bar chart of OD value for western blots. 2E, also need check plasma albumin
- 3) Fig 3, need show pathological scores. Did IL-10 KO mice show intestinal inflammation when the treatment started?
- 4) Fig 7, need show pathological scores.
- 5) Fig 8, need show pathological scores.

Reviewer #2 (Remarks to the Author):

The authors present data showing that providing small amount of mannose to mice who develop IBD following DSS insult or as a result of IL10 deficiency ameliorates the effects on the quantitative measures (symptoms) of colitis. The effects appear to be quite dramatic and the authors propose that mannose influences lysosomal integrity through promotion of an AMPK activation that leads to improved tight junction function. Mannose in combination with mesalazine treatment provides a synergistic decrease of colitis severity. They propose that this combination might be effective way to treat colitis and other lysosomal disorders.

As presented the data look impressive, but some of the methods provided or mentioned without substantiation or reference seriously undermine the conclusions.

In figure 1-mannose is measured by ELISA. Mannose is a monosaccharide and cannot be measured by ELISA. Moreover, that method is not referenced or validated. The values presented for the control patients are within expected range, but those for mice are much lower than previously seen. The lack of any referenced, validated or recognized method is major problem.

In figure 4 the amount of mannose used to establish the in vitro effects (25mM) are in stark contrast to the small amounts of mannose given to the mice developing colitis. Is this added in addition to glucose where the concentration is not given? Presumably glucose could be 25mM, as in many media. The consequences of 30-50mM monosaccharide make the results without meaningful physiological interpretation.

In various places there is insufficient description of how the colonic tissue was isolated. Were the relevant colonic cells enriched by scraping or was the entire organ homogenized. It was not clear. In Figure 5, mannose was added to cells, but the concentration was not stated. 50mM? if so, that is not acceptable. Also the use of coumarin-labeled mannose as a physiological tracer is without precedent, reference or justification. There is no evidence that such a compound can give any meaningful measurement. It would require extensive justification. Also bafilomycin is not a specific lysosomal inhibitor as claimed in the figure and text.

Overall the results look striking, but serious doubt remains because very important details of the methods are not acceptable.

This renders the proposed mechanism inadequate.

Reviewer #3 (Remarks to the Author):

This is an interesting piece of work from Dong et al that contains a considerable amount of experimentation using two murine models of colitis, and an epithelial cell line that is prefixed by an observation of serum molecules in patients with IBD. The topic is interesting and timely, but there are numerous vagaries and over-statements that detract from the paper.

Concerns:

It seems paradoxical if mannose is anti-inflammatory that the levels would be increased with disease severity. One might argue that increased circulating mannose should be consistent with reduced disease. This is difficult to argue one way or the other, but some commentary is needed on this point.

The table of IBD patients is not satisfactory. IBD needs to be split into Crohn's disease and ulcerative colitis (this should also be clear on figure 1 using different symbols to designate the groups), and the site of inflammation (i.e. colon, ileum), medications, smoking status, etc. documented. IBD is heterogeneous and using a single group may miss important information. For example looking at Fig1A, could all CD patients be clustered at the top of the graph?

I will give the authors the benefit of the doubt and suggest that they have used the journal format stipulations to present their figures, unfortunately this make the individual images very difficult to see and the validity of some data impossible to judge. This is particularly true for immunostains of gut tissue.

The methods are overly brief and insufficient. Doses and route of drug administration should be rationalized and supported by literature, if possible. Similarly, for immunoblotting antibodies and concentrations need to be added. Unless detailed methods are provided in additional supplementary material that I am unaware of, the methods as presented would not allow replication of the experiments.

There is a lot of emphasis placed on barrier in this paper, and so fluxes of albumin should be complemented with analyses of other markers of gut permeability (e.g. 70kDa FITC-dextran). Also, in all of the epithelial in vitro studies, barrier is never assessed. This is simple, just grow the cells on porous filters until electrically confluent and measure TER and fluxes.

An issue with the point above may be that the epithelial cells are not suitable for barrier studies and if so this is a poor choice of model. I am unaware of the NCM460 cell line and so this line should be clearly characterized and reference to prior publication provided. As will all cell line work, analysis of one or two key readouts with isolated enterocytes (e.g. organoids) would be beneficial.

How was the dosing regimen for oral mannose decided upon – a single does given at what point in the data to the animals.

Is there any potential that mannose binds to DSS in culture and there blocks it's affect, this seems unlikely given the data with the mannose uptake inhibitor but maybe worth a comment.

Is the mannose added before, with or after the DSS; the timing and kinetics of treatments were not specified.

Very little rational is provided for many of the readouts – why assess ZO1, occludin and claudin1 and not claudins 2 and 4 that have also been implicated in regulation of tight junction permeability.

Examining Fig. S2B, the occludin in this cell line is not actually restricted to the tight junction, and

similarly the pattern of claudin one is much more diffuse than one would expect (contrast with the chicken-wire pattern of ZO-1 in the same figure). This is a concern!

The scoring system for histopathology (Fig 2D) needs to be provided and the data assessed with the proper statistics. The manuscript only alludes to statistics for parametric data, the histopathology measure of colitis is non-parametric. I would also comment, that in my experience seldom do we see such tight data relating to colitis in mice.

The authors need to state if their n value in mice (i.e. n=6) is all from a single experiment, as I suspect is mostly the case, or combined from multiple experiments.

There is no quantification of histopathology in figure 3 and the images in 3E cannot be discerned (i.e. the IL-10^{-/-} model). This model can suffer from variable penetrance of disease in different housing facilities. The authors should describe the % of IL-10^{-/-} that got colitis, and if they performed any procedure to synchronize colitis in the mice.

The authors mention "mitochondrial mass" but this is never actually measured. Similarly with ROS and other mitochondrial functions an image is provided without any quantification!

Is DSS at 2% cytotoxic to the epithelium used in these studies? Direct in vitro cytotoxic effects of DSS on epithelium has been described before.

The image in support of reduced PDHK phosphorylation (4E) is unconvincing. There is no PDHK as a loading control.

In figure 4F, rotenone only is missing as a control. How was total ATP measured?

What was the isolation procedure for lysosome and how was contamination controlled for?

For figure 8, panels D and E are impossible to evaluate and I see no additive or synergistic effect of mannose and mesalazine on tight junction or mitochondrial ETC protein expression on the western blots provided.

Figure S3, why examine whole tissue extracts and not isolated epithelia to match the main body of the paper.

In Figure S5, Aloxistatin does not interfere with the effect of mannose on the cells.

Figure 7 does not show how the compound C accelerated the release of cathepsin B. The figure alludes to amount of enzyme but does not assess time of release.

Line 246, it is not possible to validate the effect of lysosomes in a model of colitis by conducting work on a cell line. The latter may hint at an in vivo mechanism, but it does not test one.

To more unequivocally link the pieces in this study, experiments with a cathepsin B KO are needed (mouse or cell line), and not just reliance on a single drug at a single dose.

The functional link between the lysosome and the tight junction is not addressed.

The statement on the discussion that mannose blocks IBD by maintaining intestinal permeability is at best correlative and I would soften this conclusion.

While I like the inclusion of, and focus on mitochondria, it is not entirely clear how they relate to the colitis and could in fact be a parallel event. Use of mitochondrial targeted antioxidants (e.g. mitoTEMPO) could be used to experimentally implicate (or refute) the mitochondria in a more tangible way.

The finding that mesalazine and mannose together result in less wasting to DSS and longer colons is nice from a therapeutic perspective, but adds little to the main thrust of this paper. I would move figure 8 to the supplementary material and remove the statement on the combined effects from the abstract as it oversells the mechanism, which, as noted above, is not apparent in the blots provided.

Reviewer #4 (Remarks to the Author):

Dong et al describe the effects of treatment with the sugar mannose in two mouse models of IBD: IL10 KO and DSS treatment, and in DSS treated colonic epithelial cells. They postulate that IBD leads to lysosomal disruption, release of CTSE and consequent mitochondrial damage and epithelial tight junction disruption and that mannose can function at the level of lysosomes to attenuate these phenotypes. They also investigate the role of AMPK in mediation of mannose effects.

The topic is very interesting and the proposed mechanism novel. However, the manuscript suffers from lack of rigor in experimental design, including lack of appropriate controls, lack of image and western blot quantification, and inappropriate or not specified statistical approaches. Additionally, the methods used for lysosomal function evaluation are not adequate to ascertain the proposed mechanism involving loss of lysosomal integrity.

Specific comments:

- Fig 2A: What statistical tests were used for animal experiments? Statistical approaches have to be specified for each experiment, not just in M&M. One-way ANOVA is mentioned in M&M but this is not appropriate here since there are 2 variables (DSS and mannose) and measurements are taken over time. This applies to all in vivo experiments. There is also no discussion of power analysis for animal numbers.
- Fig 2A: What was the control for mannose treatment? Neither no treatment (as in Fig 2) or water (as in Fig 3) are appropriate controls - another sugar (such as sucrose or glucose) should be included. This applies to all in vivo experiments.
- Fig 2F: All imaging data need to be quantified. This applies for all imaging data in vivo and in vitro.
- Fig 2G-H: Western blot data needs to be quantified. Data from more than one representative animal per group should be shown (ideally at least 3). Please specify # of animals per group for all experiments. This applies to all western blot experiments.
- Fig 3A: Wt controls are missing from all IL10 KO experiments.
- Fig 4F: Since the effects of rotenone are overall much stronger than the effects of DSS, it is difficult to tell anything from this experiment beside that mannose can't suppress the effects of rotenone. Untreated/vehicle, rotenone only and rotenone + mannose controls are missing. This applies to many in vitro experiments in the manuscript – untreated and single treatment controls should always be included.
- Fig 5D: Images are too low quality to appreciate co-localization. Please provide higher magnification images. All images have to be quantified, including co-localization. Do all sugars localize to lysosomes or is this specific to mannose? A control with another sugar that does not have protective effects should be included.
- Fig 5E: LysoTracker and acridine orange staining is not sufficient to ascertain loss of lysosomal integrity as it can be also caused by inhibition of acidification. Loss of integrity would be expected to lead to leakage of lysosomal contents, which presence and/or activity in the cytosol can be assessed by fractionation or imaging. Leakage of dextran species from lysosomes can also be visualized.
- Fig 5F: Decrease in LAMP 1/2 is not sufficient to ascertain loss of lysosomal integrity.
- Fig 5G: Untreated, bafA only and bafA + mannose controls are missing. Therefore, it is not possible to discern what is due to interaction of bafA with DSS and what to baf A alone. bafA and lysosomal inhibition can be toxic depending on cell type; viability should be assessed to make sure lack of response to mannose is not due to bafA toxicity.
- Lines 256-257 “mannose maintains tight junctions by preserving lysosomal integrity”: this is overinterpretation - data indicate that mannose improves tight junctions and some parameters of lysosomal function but does not show loss of lysosomal integrity.

- Fig S5: What controls were used to demonstrate that inhibitors were effectively inhibiting target proteases? The doses required are cell type specific and appropriate controls must be included. Additionally, lysosomal inhibition per se, can be toxic. How long were the cells treated and was viability checked? None of the inhibitors are fully specific, so the only way to prove specificity of CTSB involvement would be to KO/KD it down. Did you check effect of alloxistatin on caspase activity - they are cysteine proteases as well. Aloxistatin seems to work better than mannose, so it's hard to tell whether it eliminated or just masked effects of mannose - a dose response would be helpful.
- Fig 6A-B: What is the cell type accumulating CTSB in vivo? How about other lysosomal enzymes? Are they affected as well - ie. general expansion of the lysosomal compartment, or is this unique to CTSB?
- Fig 6C: I am not sure if increased CTSB activity really goes with the proposed mechanism being loss of lysosomal integrity. Loss of lysosomal integrity would be expected to decrease activity since activity is dependent on low pH. pH increases when lysosomal membranes are compromised. CTSB processing would also be affected since mature form is dependent on low pH. What are the sizes of the bands on western blot? Where is the mature form? Fractionation to check CTSB activity in lysosomes vs. cytosol could address this. Additionally, if lysosomal integrity is compromised, other enzymes in addition to CTSB should be affected.
- Fig 7A: AMPK is usually activated in response to low energy - for example glucose starvation. Why is it activated in response to addition of mannose (sugar)?
- Fig S7: compC seems more effective than mannose as far as increasing LAMP 1/2 expression. Also, mannose fails to increase LAMP 1/2 expression in this experiment - inconsistent with Fig 5 and with hypothesis.

Detailed responses to the reviewer' comments (NCOMMS-19-34782)

We are grateful for the reviewers' positive comments, and appreciate the valuable suggestions of the expert reviewers. We have conducted additional experiments to strengthen our manuscript and presented new data in the revised manuscript. Here we address the concerns of the reviewers, point by point.

Reviewer #1 (Remarks to the Author):

In this study, Dong et al demonstrated that mannose regulates intestinal homeostasis through promoting epithelial barrier function. They showed nicely that mannose protected colitis in DSS model and IL-10ko model, which was mediated by prevention of mitochondrial dysfunction and tight junction disruption. The study was well designed and performed with comprehensive approaches. The conclusion was supported by major results. However, some concerns need be addressed to further improve the quality of the manuscript.

1. Fig 1G and I, please show the pathological scores. Fig 1J, although the DSS dose response is interesting for increase of mannose in plasma, what drives such increase? did DSS in low doses cause colitis or disruption of epithelial barrier function?

Response: We thank the reviewer for this suggestion. The colitis severity was assessed blinded by a pathologist following the criteria of histological score previously reported ¹. The colitis score (maximum = 8) was the sum of the following two features: Epithelium (0: Normal morphology; 1: Loss of goblet cells, 2: Loss of goblet cells in large areas; 3: Loss of crypts; 4: Loss of crypts in large areas), and infiltration (0: No infiltrate; 1: Infiltrate around crypt basis; 2: Infiltrate reaching to lamina muscularis mucosae; 3: Extensive infiltration reaching the lamina muscularis mucosae and thickening of the mucosa with abundant edema; 4: Infiltration of the lamina submucosa). We have included the detailed information in the method section (**line 555-562**). The pathological scores were calculated and showed in Figure 1 (**Fig. 1d and 1f**). The results indicated that the extent of intestinal pathological severity was associated with the time duration and concentration of DSS administration in the colitis model mice.

Mannose is a simple sugar, which absorbed through the intestine and metabolized ². We assume that DSS damaged the intestinal epithelial cells, leading to the dysfunction of mannose metabolism. Additionally, DSS induces a loss of intestinal barrier function in mice. Overall, the intestinal damage might be the main reason of the elevation of plasma mannose. We include a short discussion about this in the revised manuscript (**line 349-351**).

Oral administration of the sulfated polysaccharide DSS to mice via drinking water induces severe colitis characterized by weight loss, bloody diarrhea, ulcer formation, and loss of epithelial cells, resembling some features of flares in human UC. Our data indicates that DSS in low dose can cause a mild intestinal barrier damage (**Fig. 1f**). Additionally, we demonstrated that DSS induced the disruption of intestinal barrier function at a dose-dependent manner, determined by FITC-dextran permeability assay (**Supplementary Fig. 1d**).

2. Fig 2A, need show pathological scores. 2G and H, need show bar chart of OD value for western blots. 2E, also need check plasma albumin

Response: As suggested, the pathological scores were obtained from blinded assessment by a pathologist (**Fig. 2c**).

In Figure 2, we have quantified the densitometry of western blot bands, and statistically analyzed the difference among the groups. The data has been included in the figures in the revised manuscript (**Figure 2h and 2i**).

We conducted ELISA analysis to determine plasma albumin levels in the DSS-administrated mice with or without mannose treatment. The result showed that mannose treatment significantly increased the plasma level of albumin in DSS-challenged mice (**Fig. 2d**). Additionally, fluorescein isothiocyanate-dextran (FD4) permeability assay was employed to confirm the gut leakiness in vivo in this revised manuscript (**Fig. 2f**).

3. Fig 3, need show pathological scores. Did IL-10 KO mice show intestinal inflammation when the treatment started?

Response: As suggested, we have included the pathological scores in **Fig. 3c**.

The IL-10^{-/-} mice consistently developed colitis at 10 weeks of age when maintained in the SPF environment as previously reported^{3, 4}. The clinical manifestations and intestinal inflammation delineated an early phase of colitis (10-24 weeks), characterized by a progressive increase in disease severity, followed by a late phase (>25 weeks), in which chronic inflammation persisted indefinitely. In our study, the 15-week-old IL-10^{-/-} mice were used for the experiment. It is worth noting that loose stools, occult fecal blood, and rectal prolapse were observed in the 15-week-old IL-10^{-/-} mice before the treatment started, which indicated that the elder IL-10^{-/-} mice exhibited intestinal inflammation.

4. Fig 7, need show pathological scores.

Response: As suggested by the reviewer, we have included the pathological scores in **Fig. 7g**.

5. Fig 8, need show pathological scores.

Response: As suggested by the reviewer, we have included the pathological scores in **Supplementary Fig. 14d**.

Reviewer #2 (Remarks to the Author):

The authors present data showing that providing small amount of mannose to mice who develop IBD following DSS insult or as a result of IL10 deficiency ameliorates the effects on the quantitative measures (symptoms) of colitis. The effects appear to be quite dramatic and the authors propose that mannose influences lysosomal integrity through promotion of an AMPK activation that leads to improved tight junction function. Mannose in combination with mesalazine treatment provides a synergistic decrease of colitis severity. They propose that this combination might be effective way to treat colitis and other lysosomal disorders.

As presented the data look impressive, but some of the methods provided or mentioned without substantiation or reference seriously undermine the conclusions.

1. In figure 1-mannose is measured by ELISA. Mannose is a monosaccharide and cannot be measured by ELISA. Moreover, that method is not referenced or validated. The values presented for the control patients are within expected range, but those for mice are much lower than previously seen. The lack of any referenced, validated or recognized method is major problem.

Response: We would like to thank the reviewer for this comment and apologize for our previous mistake. The serum mannose levels were evaluated by a competitive enzyme immunoassay (OKEH02595, Aviva Systems Biology, San Diego, CA, USA). We now have provided this essential information in the revised manuscript (**line 573-583**). The microtiter well-plate in this kit has been pre-coated with an anti-mannose antibody. Sample or standards are added to the wells along with a fixed quantity of biotinylated mannose and incubated. The mannose found in the sample or standards competes with the biotinylated mannose for limited binding sites on the immobilized anti-mannose antibody. Excess unbound biotinylated mannose and sample or standard mannose is washed from the plate. Avidin-HRP is used to detect the immobilized biotinylated mannose, followed by colorimetric assays.

We deeply apologize for the result of mice serum mannose levels. After careful checking and repeating the experiments, we observed that the wrong calculation was occurred in serum mannose concentration of mice. We recalculated the serum level of mannose in mice. The corrected data is provided in Figure 1 in the revised manuscript (**Fig. 1**).

2. In figure 4 the amount of mannose used to establish the in vitro effects (25mM) are in stark contrast to the small amounts of mannose given to the mice developing colitis. Is this added in addition to glucose where the concentration is not given? Presumably glucose could be 25mM, as in many media. The consequences of 30-50mM monosaccharide make the results without meaningful physiological interpretation.

Response: Thanks to the reviewer for this comment. We would like to explain that mouse primary colonic epithelial cells and human colon epithelial cell line NCM460 were used for the in vitro study. For the mouse primary colonic epithelial cells, 5 mM mannose is the optimal concentration to protect DSS-induced intestinal barrier injury. For the NCM460 cells, 25 mM mannose was used. We would like to point out that we used the modified RPMI 1640 medium (RPMI 1640 medium, no glucose

(Invitrogen, Carlsbad, CA, USA, 11879020) for the NCM 460 cell culture. In our experiment, we used RPMI 1640 medium with 1 mM glucose, where the concentration of monosaccharide did not exceed 30mM when mannose added. The detailed information about the cell culture was provided in the method section (**line 507-509**).

3. In various places there is insufficient description of how the colonic tissue was isolated. Were the relevant colonic cells enriched by scraping or was the entire organ homogenized. It was not clear.

Response: We apologize for the confusion made. We evaluated the protein expression in the samples from either isolated colonic epithelial cells or colon tissue homogenization. In the revised manuscript, we provided the detail information about the sample in the figure legend section. Also, the detail method about the colonic epithelial cell isolation is included in the method section (**line 600-607**).

4. In Figure 5, mannose was added to cells, but the concentration was not stated. 50mM? if so, that is not acceptable. Also the use of coumarin-labeled mannose as a physiological tracer is without precedent, reference or justification. There is no evidence that such a compound can give any meaningful measurement. It would require extensive justification. Also bafilomycin is not a specific lysosomal inhibitor as claimed in the figure and text.

Response: We apologize for the previous mistake. In Figure 5, 1 mM mannose was added to cells. The compound coumarin-labeled-mannose was synthesized by Ruixi Biological Technology Co., Ltd (Xi'an, China) according to the published paper⁵. In brief, coumarin was conjugated to the terminal amino group of D-mannosamine hydrochloride in a 1: 1 mixture of acetonitrile and sodium borate buffer. After stirred at 4 °C for 2 hours in the dark, the reaction mixture was placed in room temperature storage for 1 hour. The products were isolated and purified by semipreparative, reverse-phase high-performance liquid chromatography (HPLC). The detailed information was included in the method section (**line 661-668**).

Our data suggest that mannose prevents DSS-induced tight junction disruption by limiting cathepsin B release from lysosomes. In the revised manuscript, we employed the CRISPR-Cas9 system to generate cathepsin B-deficient cell. Indeed, cathepsin B deficiency diminished mannose-mediated upregulation of tight junctions in colonic epithelial cells (**Fig. 6l**).

Reviewer #3 (Remarks to the Author):

This is an interesting piece of work from Dong et al that contains a considerable amount of experimentation using two murine models of colitis, and an epithelial cell line that is prefixed by an observation of serum molecules in patients with IBD. The topic is interesting and timely, but there are numerous vagaries and over-statements that detract from the paper.

Concerns:

1. It seems paradoxical if mannose is anti-inflammatory that the levels would be increased with disease severity. One might argue that increased circulating mannose should be consistent with reduced disease. This is difficult to argue one way or the other, but some commentary is needed on this point.

Response: We appreciate and agree with this comment. In our study, we observed a significant increased circulating mannose level in IBD patients and mice model of colitis. Mannose is a simple sugar, which absorbed through the intestine and metabolized ². We assume that DSS damaged the intestinal epithelial cells, leading to the dysfunction of mannose metabolism. Additionally, DSS induces a loss of intestinal barrier function in mice. Therefore, we speculated that the increased level of mannose is a response to the intestinal damage. A short discussion about this was included in the revised manuscript (**line 349-351**).

2. The table of IBD patients is not satisfactory. IBD needs to be split into Crohn's disease and ulcerative colitis (this should also be clear on figure 1 using different symbols to designate the groups), and the site of inflammation (i.e. colon, ileum), medications, smoking status, etc. documented. IBD is heterogeneous and using a single group may miss important information. For example looking at Fig1A, could all CD patients be clustered at the top of the graph?

Response: We greatly appreciate this valuable suggestion. In the revised manuscript, we have included more information about the IBD patients as suggested (**Supplementary Tab. 1**). Additionally, we used different symbols to designate patients with Crohn's disease and patients with ulcerative colitis in Figure 1 (**Fig. 1a, b**).

3. I will give the authors the benefit of the doubt and suggest that they have used the journal format stipulations to present their figures, unfortunately this make the individual images very difficult to see and the validity of some data impossible to judge. This is particularly true for immunostains of gut tissue.

Response: We sincerely apologize for the inconvenience. We have provided much more clear figures, especially the histopathological photomicrographs, in this revised manuscript.

4. The methods are overly brief and insufficient. Doses and route of drug administration should be rationalized and supported by literature, if possible. Similarly, for immunoblotting antibodies and concentrations need to be added. Unless detailed methods are provided in additional supplementary

material that I am unaware of, the methods as presented would not allow replication of the experiments.

Response: We greatly thank for this advice. As suggested, we now provided much more detailed description about the methods in the revised manuscript, including the drug administration and the concentration of immunoblotting antibodies used in the study.

5. There is a lot of emphasis placed on barrier in this paper, and so fluxes of albumin should be complemented with analyses of other markers of gut permeability (e.g. 70kDa FITC-dextran). Also, in all of the epithelial in vitro studies, barrier is never assessed. This is simple, just grow the cells on porous filters until electrically confluent and measure TER and fluxes.

Response: We appreciate this suggestion. As suggested, the FITC-Dextran permeability experiment was employed to confirm gut leakiness *in vivo* in the revised manuscript. The result showed that mannose treatment suppressed the FD4 leakage from the gut to the blood compartment (**Fig. 2f**). Besides, we measure the transepithelial electrical resistance (TER) of intestinal epithelial cell monolayers *in vitro*. The data showed that the TER increased significantly in DSS-stimulated colonic epithelial cells after incubation with mannose (**Supplementary Fig. 3b and h**).

6. An issue with the point above may be that the epithelial cells are not suitable for barrier studies and if so this is a poor choice of model. I am unaware of the NCM460 cell line and so this line should be clearly characterized and reference to prior publication provided. As will all cell line work, analysis of one or two key readouts with isolated enterocytes (e.g. organoids) would be beneficial.

Response: We appreciated for this suggestion. The NCM460 (normal colon mucosa 460) cell line is an epithelial cell line derived from the normal colon of a 68-year-old Hispanic male⁶. The cells were positive for the epithelial markers villin. Usually, NCM460 cells serve as an *in vitro* model for the study of intestinal epithelial function.

As suggested, we isolated mouse primary colonic epithelial cells to evaluate the protective effect of mannose on DSS-induced tight junction disruption. After mannose treatment, an increase in the TER of DSS-stimulated primary colonic epithelial cells was observed (**Supplementary Fig. 3h**). Immunoblotting analysis showed that mannose restored the expression of tight junction proteins in the DSS-treated primary colonic epithelial cells (**Supplementary Fig. 3i**). Also, mannose rescued the expression of respiratory complexes and PDH in the colonic epithelial cells from DSS-challenged mice (**Supplementary Fig. 4e**). Moreover, the expression of lysosomal associated membrane protein in primary colonic epithelial cells was also evaluated (**Supplementary Fig. 7b**). Together, consistent with the results of NCM460 cells, mannose can protect the barrier damage of primary colonic epithelial cells caused by DSS.

7. How was the dosing regimen for oral mannose decided upon – a single does given at what point in the data to the animals.

Response: As described in a previous study⁷, 5 gradients of concentration were set for oral treatment in rodent models. Furthermore, 8 a.m. in the morning was chosen as a set time for feeding,

with a duration of 7 days. We provided the information in the method section in the revised manuscript (**line 444-447**).

8. Is there any potential that mannose binds to DSS in culture and there blocks it's affect, this seems unlikely given the data with the mannose uptake inhibitor but maybe worth a comment.

Response: We would like to thank the reviewer for this comment. Considering the structure of mannose, there's no binding site available for DSS, which in other words, the direct binding of mannose and DSS is rarely possible. We added a short discription about this in the revised manuscript (**line 211-212**).

9. Is the mannose added before, with or after the DSS; the timing and kinetics of treatments were not specified.

Response: We thank the reviewer for this comment. In our previous study, mannose was added simultaneously with DSS. We included the information in the revised manuscript. Additionally, we conducted experiments to evaluate the protective capacity of mannose to DSS-induced epithelial cell dysfunction depended on whether mannose added before, with or after DSS. The result showed that mannose has little beneficial effect on tight junction disruption when mannose added before DSS (**Supplementary Fig. 3g**). Of note, maximum protective effect occurred when mannose when mannose and DSS were added simultaneously (**Supplementary Fig. 3e**).

10. Very little rational is provided for many of the readouts – why assess ZO1, occludin and claudin1 and not claudins 2 and 4 that have also been implicated in regulation of tight junction permeability.

Response: We appreciate this comment. Claudins are a family of proteins, which consists of at least 27 transmembrane proteins, are the important components of the tight junctions. To validate the function of mannose in regulation of tight junction permeability, we conduct several experiments to evaluate the expression of claudin 2 and claudin 4. The result showed that the expression of claudin 2 and claudin 4 in colonic tissue of colitis mice were significantly improved by mannose treatment.

11. Examining Fig. S2B, the occludin in this cell line is not actually restricted to the tight junction, and similarly the pattern of claudin one is much more diffuse that one would expect (contrast with the chicken-wire pattern of ZO-1 in the same figure). This is a concern!

Response: We apologize for the error in the original figure and thank the reviewer for the comment. After cautious inspect, we found that the unacceptable non-specific binding of antibodies was owing to our inappropriate reagent choice during the immunofluorescence process. The new antibody was applied and the results were in line with those in ZO-1 and occludin, which is exactly the structure of chicken-wire pattern (**Supplementary Fig. 3d**).

12. The scoring system for histopathology (Fig 2D) needs to be provided and the data assessed with the proper statistics. The manuscript only alludes to statistics for parametric data, the histopathology measure of colitis in non-parametric. I would also comment, that in my experience seldom do we see

such tight data relating to colitis in mice.

Response: We thank and agree with the reviewer for this suggestion. The colitis severity was assessed blinded by a pathologist following the criteria of histological score previously reported ¹. Briefly, the colitis score (maximum=8) was the sum of the following two features: Epithelium (0: Normal morphology; 1: Loss of goblet cells, 2: Loss of goblet cells in large areas; 3: Loss of crypts; 4: Loss of crypts in large areas), and infiltration (0: No infiltrate; 1: Infiltrate around crypt basis; 2: Infiltrate reaching to lamina muscularis mucosae; 3: Extensive infiltration reaching the lamina muscularis mucosae and thickening of the mucosa with abundant edema; 4: Infiltration of the lamina submucosa). We have included the detailed information in the method section (**line 555-562**). The pathological scores were calculated and showed. The results indicated that the extent of intestinal pathological severity was associated with the time duration and concentration of DSS administration in the colitis model mice. Also, we provided the information of the statistics assay in the method section (**line 693-696**).

13. The authors need to state if there n value in mice (i.e. n=6) is all from a single experiment, as I suspect is mostly the case, or combined form multiple experiments.

Response: In our study, three independent experiments were performed, and 5 to 8 mice per group were used per experiment. We have included the details in the method section and corresponding figure legends in the revised manuscript.

14. There is no quantification of histopathology in figure 3 and the images in 3E cannot be discerned (i.e. the IL-10^{-/-} model). This model can suffer from variable penetrence of disease in different housing facilities. The authors should describe the % of IL-10^{-/-} that got colitis, and if they performed any procedure to synchronize colitis in the mice.

Response: We would like to thank the reviewer for this suggestion. As suggested, the quantification of histopathology in figure 3 was included in the revised manuscript. Additionally, we have provided much more clear immunofluorescence figures (**Fig. 3f**).

Usually, the IL-10^{-/-} mice consistently developed colitis at 10 weeks of age when maintained in the SPF environment as previously reported ^{3,4}. In our study, the 15-week-old IL-10^{-/-} mice were used for the experiment. It is worth noting that loose stools, occult fecal blood, and rectal prolapse were observed in the 15-week-old IL-10^{-/-} mice before the treatment started, which help us to recruit the IL-10^{-/-} mice with colitis in our study.

15. The authors mention “mitochondrial mass” but this is never actually measured. Similarly with ROS and other mitochondrial functions an image is provided without any quantification!

Response: We greatly thank the reviewer for this advice. In the revised manuscript, all the mitochondrial images, as well as the MitoSOX and JC-1 fluorescence, were analyzed using ImageJ analysis software.

16. Is DSS at 2% cytotoxic to the epithelium used in these studies? Direct *in vitro* cytotoxic effects of DSS on epithelium has been described before.

Response: We agree with the reviewer for this comment. The direct *in vitro* cytotoxic effect of DSS on intestinal epithelial cells has been reported previously^{8, 9}. Notably, the cytotoxicity of DSS towards intestinal epithelial cells seems to have concentration, time, and cell type dependency with increasing concentrations and time causing increased cytotoxicity⁸. Araki et al., observed that DSS induced cytotoxicity on Caco-2 cells at 5%⁹. In the revised manuscript, we assessed the effect of different concentrations of DSS on the apoptosis of NCM460 cells. FACS analysis showed that DSS has little cytotoxicity to NCM460 cells at 2%. The data was presented in the **Supplementary Fig. 3a**.

17. The image in support of reduced PDHK phosphorylation (4E) is unconvincing. There is no PDHK as a loading control.

Response: We deeply apologize for this error in our original manuscript. We examined the expression of PDHK1, not its phosphorylation, in our study. We observed a reduction in PDHK1 level that coincided with increased PDH expression in colonic epithelial cells stimulated with DSS plus mannose compared with those stimulated with DSS alone (**Fig. 4c**). The mistake is corrected in the revised manuscript (**line 192**).

18. In figure 4F, rotenone only is missing as a control. How was total ATP measured?

Response: We thank the reviewer for this suggestion. The rotenone only group was added as a control in the revised manuscript (**Fig. 4d**). In the Seahorse assay, oligomycin is added and the resulting OCR is used to derive ATP-linked respiration (by subtracting the oligomycin rate from baseline cellular OCR). In addition, methods of calculation of the basal respiration value and the maximum respiration of cells have been provided in the method section in the revised manuscript (**line 617-632**).

19. What was the isolation procedure for lysosome and how was contamination controlled for?

Response: The lysosome was isolated using the lysosome enrichment kit for tissue and cultured cells, which was purchased from Abcam. We have provided the detail information in the revised manuscript (**line 654-659**). To evaluate the level of target protein in isolated lysosome by western blotting, LAMP-1 serves as a loading control.

20. For figure 8, panels D and E are impossible to evaluate and I see no additive or synergistic effect of mannose and mesalazine on tight junction or mitochondrial ETC protein expression on the western blots provided.

Response: We would like to thank the reviewer for this comment. Combination of mesalazine and mannose have beneficial effect on DSS-induced colitis, which indicated a therapeutic perspective. We agree with the reviewer that this result adds little to the main thrust of the manuscript. Therefore,

we move the statement of synergistic effect and remove this figure to the supplemental material (**Supplementary Fig. 14**).

21. *Figure S3, why examine whole tissue extracts and not isolated epithelia to match the main body of the paper.*

Response: We would like to thank for this comment. In the previous manuscript, we determined the expression levels of OXPHOS in the whole colonic tissue extracts. We now also evaluated the expressions of OXPHOS in the isolated epithelial cells and the data was provided as **Supplementary 4e**. The result showed that mannose rescues the colitis-induced alterations of oxidative phosphorylation in colonic epithelial cells.

22. *In Figure S5, Aloxistatin does not interfere with the effect of mannose on the cells.*

Response: We deeply apologize for the error in the original manuscript. The three inhibitors pepstatin A, nafamostat mesylate and aloxistatin are mislabeled in previous figure. Aloxistatin does interfere with the effect of mannose on the cells. The data is carefully checked and correctly presented now.

23. *Figure 7 does not show how the compound C accelerated the release of cathepsin B. The figure alludes to amount of enzyme but does not assess time of release.*

Response: We greatly appreciate this suggestion. As suggested, we test the effect of compound C treatment on cathepsin B release at different time point. The result showed that compound C treatment significantly increased cathepsin B release from epithelial cells at 12 hours after DSS exposure.

24. *Line 246, it is not possible to validate the effect of lysosomes in a model of colitis by conducting work on a cell line. The latter may hint at an in vivo mechanism, but it does not test one.*

Response: We thank reviewer for this suggestion. It has been reported that Bafilomycin A1 can inhibit lysosomal acidification in vitro and in vivo¹⁰. In the revised manuscript, in vivo effect of bafilomycin A1 on the progression of DSS-induced colitis was evaluated. The result showed that bafilomycin A1 treatment abrogated the mannose-mediated colonic protection. The body weight loss and colonic shortening were similar in the mice administered DSS plus mannose or DSS alone in the presence of bafilomycin A1, which was also validated by pathological analyses (**Supplementary Fig. 9**).

25. *To more unequivocally link the pieces in this study, experiments with a cathepsin B KO are needed (mouse or cell line), and not just reliance on a single drug at a single dose.*

Response: We appreciate this comment. In the revised manuscript, the CRISPR-Cas9 system was employed to generate cathepsin B knockout cells. Notably, cathepsin B deficiency diminished

mannose-mediated upregulation of tight junctions in colonic epithelial cells (**Fig. 6l**).

26. The functional link between the lysosome and the tight junction is not addressed.

Response: We used bafilomycin A1 to inhibit the function both in vivo and in vitro. The result showed that bafilomycin A1 pre-treatment abrogated the mannose-mediated colonic protection in DSS-administrated mice (**Supplementary Fig. 9**). Additionally, the levels of tight junction proteins were comparable in cells treated with DSS alone or with DSS combined with mannose in the presence of bafilomycin A1 (**Fig. 5h**).

27. The statement on the discussion that mannose blocks IBD by maintaining intestinal permeability is at best correlative and I would soften this conclusion.

Response: We thank the reviewer for this suggestion. In the revised manuscript, we changed the sentence to “We also identified a beneficial effect of mannose in preventing IBD by modulating the tight junctions.”, which might make the statement more accurate (**line 351**).

28. While I like the inclusion of, and focus on mitochondria, it is not entirely clearly how they relate to the colitis and could in fact be a parallel event. Use of mitochondrial targeted antioxidants (e.g. mitoTEMPO) could be used to experimentally to implicate (or refute) the mitochondria in a more tangible way.

Response: We would like to thank the reviewer for this suggestion. As suggested, the effect of mitoTEMPO on DSS-induced epithelial injury was evaluated in vivo and in vitro. The mito-TEMPO treatment aggravated the DSS-induced disruption of tight junctions and abolished the mannose-mediated upregulation of the expression of tight junction proteins in DSS-treated cells (**Fig. 4e**). Moreover, mito-TEMPO pretreatment abrogated the mannose-mediated colonic protection. The body weight loss and colonic shortening were similar in the mice administered DSS plus mannose or DSS alone in the presence of mito-TEMPO, which was confirmed by pathological analyses (**Supplementary Fig. 5**).

29. The finding that mesalazine and mannose together result in less wasting to DSS and longer colons is nice from a therapeutic perspective, but adds little to the main thrust of this paper. I would move figure 8 to the supplementary material and remove the statement on the combined effects from the abstract as it oversells the mechanism, which, as noted above, is not apparent in the blots provided.

Response: We thank and agree the reviewer’s comment. In the revised manuscript, the figure 8 has been moved into the supplementary material and we also removed the statement on the combined effects from the abstract.

Reviewer #4 (Remarks to the Author):

Dong et al describe the effects of treatment with the sugar mannose in two mouse models of IBD: IL10 KO and DSS treatment, and in DSS treated colonic epithelial cells. They postulate that IBD leads to lysosomal disruption, release of CTSB and consequent mitochondrial damage and epithelial tight junction disruption and that mannose can function at the level of lysosomes to attenuate these phenotypes. They also investigate the role of AMPK in mediation of mannose effects.

The topic is very interesting and the proposed mechanism novel. However, the manuscript suffers from lack of rigor in experimental design, including lack of appropriate controls, lack of image and western blot quantification, and inappropriate or not specified statistical approaches. Additionally, the methods used for lysosomal function evaluation are not adequate to ascertain the proposed mechanism involving loss of lysosomal integrity.

Specific comments:

1. Fig 2A: What statistical tests were used for animal experiments? Statistical approaches have to be specified for each experiment, not just in M&M. One-way ANOVA is mentioned in M&M but this is not appropriate here since there are 2 variables (DSS and mannose) and measurements are taken over time. This applies to all in vivo experiments. There is also no discussion of power analysis for animal numbers.

Response: We are sorry for the lack of detailed description of statistical tests in the previous submission. We have provided the detailed information in the method section in the revised manuscript.

Thank you for bringing up this serious question, we can't agree with you more that the statistical approaches we used in above-mentioned experiments are not appropriate. We have revised the statistical test and used the two-way ANOVA to analyze *in vivo* experiments owing to two variables.

The power analysis for animal numbers was calculated by PASS 12 software and the results showed statistically significant. In addition, three independent experiments were performed for all in vivo experiments, and the number of animals in each group was 6-8 in one experiment.

2. Fig 2A: What was the control for mannose treatment? Neither no treatment (as in Fig 2) or water (as in Fig 3) are appropriate controls - another sugar (such as sucrose or glucose) should be included. This applies to all in vivo experiments.

Response: We are very grateful and agree with this comment. Mannose is a C-2 epimer of glucose. As suggested, the colitis mice were received same amount of glucose as a control for the *in vivo* experiments. The result showed that glucose failed to attenuate the colonic inflammation in both chemically-induced colitis and spontaneous colitis model mice when used at the same molar concentration as mannose. The data was included in the revised manuscripts (**Fig. 2, 3, 7 and Supplementary Fig. 2, 4, 5, 12**).

3. Fig2F: All imaging data need to be quantified. This applies for all imaging data in vivo and in vitro.

Response: We thank and agree the reviewer's comment. Quantization of images can intuitively reflect the differences between groups. All the images had been quantitated including the tissue pathological score, the OD value of western blots and index of correlation about mitochondria, etc.

4. Fig 2G-H: Western blot data needs to be quantified. Data from more than one representative animal per group should be shown (ideally at least 3). Please specify # of animals per group for all experiments. This applies to all western blot experiments.

Response: We thank for this comment. In the revised manuscript, all the western blot data was quantified by ImageJ software. Additionally, 3 mice per group were applied for western blot analysis.

5. Fig 3A: Wt controls are missing form all IL10 KO experiments.

Response: As required, WT mice at same age and sex were included as the control for IL-10 KO mice in all in vivo experiments (**Fig. 3 and Supplementary Fig. 4**).

6. Fig 4F: Since the effects of rotenone are overall much stronger than the effects of DSS, it is difficult to tell anything from this experiment beside that mannose can't suppress the effects of rotenone. Untreated/vehicle, rotenone only and rotenone + mannose controls are missing. This applies to many in vitro experiments in the manuscript – untreated and single treatment controls should always be included.

Response: We greatly appreciate this comment. We, therefore, determine the effect of different concentrations of rotenone on epithelial cell apoptosis. Surprisingly, we found that rotenone displayed toxicity to the NCM460 cells at the dose we previously used. In the revised manuscript, we re-adjusted the concentration of rotenone for the inhibition assay. The result showed that rotenone at low concentration could also prevent the restoration of tight junction protein expression in DSS-stimulated cells (Fig. 4d). Two groups, rotenone only and rotenone combined with mannose treatment, have been included in the data. Meanwhile, the untreated and single treatment controls for other inhibitors were either provided in the revised manuscript.

7. Fig 5D: Images are too low quality to appreciate co-localization. Please provide higher magnification images. All images have to be quantified, including co-localization. Do all sugars localize to lysosomes or is this specific to mannose? A control with another sugar that does not have protective effects should be included.

Response: We apologize for this inconvenience. In the revised manuscript, much more sharper images were provided and the co-localization was quantified. In addition, NCM460 cells were incubated with Rhodamine-labeled glucose to identify the localization of glucose in epithelial cells. The result showed that glucose is evenly distributed in the cytoplasm and is not specifically located in the lysosomes (**Fig. 5c**). Notably, free polymannose-type oligosaccharides incubated with liver cells have previously been shown to predominantly accumulate in lysosomes¹¹. In addition, we also

observed that glucose failed to protect against DSS-induced colitis and IL-10^{-/-} mice developed colitis when used at the same concentrations as mannose (**Fig. 2 and Fig. 3**).

8. *Fig 5E: LysoTracker and acridine orange staining is not sufficient to ascertain loss of lysosomal integrity as it can be also caused by inhibition of acidification. Loss of integrity would be expected to lead to leakage of lysosomal contents, which presence and/or activity in the cytosol can be assessed by fractionation or imaging. Leakage of dextran species from lysosomes can also be visualized.*

Response: We thank the reviewer for the valuable comment. PDMPO is selectively localized to lysosomes and exhibits a pH-dependent dual excitation and emission. In the revised manuscript, lysosomal acidification was assessed by PDMPO staining following the previously reported¹². The results showed that DSS treatment prevented the lysosomal acidification in colonic epithelial cells. Strikingly, mannose treatment rescued the acidification of lysosome in DSS-treated NCM460 cells (**Fig. 5e**).

NCM460 cells were loaded with FITC-dextran and stimulated with DSS in the presence or absence of mannose for 24 hours. Increased green haziness in the cytosol is an indicative of dextran leakage following lysosomal membrane permeabilization¹³. As shown in Figure 5G, DSS triggered a strong leakage of dextran in NCM460 cells, while mannose treatment inhibited the dextran leakage induced by DSS in the epithelial cells. The result indicated that mannose prevented lysosomal dysfunction in cells against DSS stimulation (**Fig. 5f**).

9. *Fig 5F: Decrease in LAMP 1/2 is not sufficient to ascertain loss of lysosomal integrity.*

Response: We thank and agree this comment. As a marker of lysosomes, LAMP1/2 expression is down-regulated when lysosomes disintegrated^{14,15}. In addition, in order to further confirm the effect of mannose on the integrity of lysosomes, we examined the colocalization of dextran-FITC with lysosomes and the level of lysosomal acidification. The results demonstrated that mannose restore DSS induced increase of lysosomal pH of NCM460 cells (**Fig. 5e**) and translocation of dextran-FITC to lysosomes (**Fig. 5f**), which indicated that mannose rescues DSS induced damage of lysosomal integrity. By the way, due to the space limit, we have removed the data about LAMP1/2 expression into **Supplementary Fig. 7**.

10. *Fig 5G: Untreated, bafA only and bafA + mannose controls are missing. Therefore, it is not possible to discern what is due to interaction of bafA with DSS and what to baf A alone. bafA and lysosomal inhibition can be toxic depending on cell type; viability should be assessed to make sure lack of response to mannose is not due to bafA toxicity.*

Response: As required, untreated, baf A1 only and baf A1+mannose controls were added in the revised manuscript (**Fig. 5g**). Additionally, we determined the effect of different concentrations of baf A1 on epithelial cell apoptosis. The result showed that baf A1 displayed little toxicity to the NCM460 cells at the dose we used (**Supplementary Fig. 8**).

11. Lines 256-257 “mannose maintains tight junctions by preserving lysosomal integrity”: this is overinterpretation - data indicate that mannose improves tight junctions and some parameters of lysosomal function but does not show loss of lysosomal integrity.

Response: We thank reviewer for this suggestion. In the revised manuscript, the sentence was amended. We assumed that mannose improves tight junction by partially strengthening lysosomal function (**line 249-250**).

12. Fig S5: What controls were used to demonstrate that inhibitors were effectively inhibiting target proteases? The doses required are cell type specific and appropriate controls must be included. Additionally, lysosomal inhibition per se, can be toxic. How long were the cells treated and was viability checked? None of the inhibitors are fully specific, so the only way to prove specificity of CTSB involvement would be to KO/KD it down. Did you check effect of aloxistatin on caspase activity - they are cysteine proteases as well. Aloxistatin seems to work better than mannose, so it's hard to tell whether it eliminated or just masked effects of mannose - a dose response would be helpful.

Response: We appreciate these comments. Pepstatin A is well known to be an inhibitor of aspartic proteinases such as cathepsins D and E ¹⁶. Aloxistatin is an irreversible, membrane-permeable inhibitor of lysosomal and cytosolic cysteine proteases, including cathepsins B, H, and L ¹⁷. We also detected the cathepsin B activity in aloxistatin-treated cells. Nafamostat mesylate, a synthetic serine protease inhibitor, prevents cellular and lysosomal fragility ¹⁸. By the way, we included the references in the revised manuscript.

As suggested, the intervention time and dose of the inhibitors are presented in the section of material and method. In addition, the effects of aloxistatin on cell viability were determined by Annexin V staining and the data were showed in **Supplementary Fig. 10f**.

To validate the role of cathepsin B in mannose-mediated protection against colonic inflammation, cathepsin B-deficient NCM460 cells were generated by CRISPR/Cas9 system according to the manufacturer's instructions. The result showed that the expression levels of tight junction were similar in DSS-treated cathepsin B^{-/-} cells in the presence or absence of mannose (**Fig. 6l**).

We observed that high dose of aloxistatin can upregulate the expression of caspase3 and caspase9 in NCM460 cells. However, 1μM aloxistatin, the concentration we used in the study, has little effect on cell survival rate and caspase activity (**Supplementary Fig. 10 e and f**). In our study, 1μM aloxistatin was used to inhibit the cysteine protease activity in epithelial cells.

13. Fig 6A-B: What is the cell type accumulating CTSB in vivo? How about other lysosomal enzymes? Are they affected as well - ie. general expansion of the lysosomal compartment, or is this unique to CTSB?

Response: Histopathological images show that cathepsin B is predominantly clustered in the epithelial cells of the colon. In the revised manuscript, we isolated the colonic epithelial cells from DSS-induced colitis mice, and evaluated the cathepsin B expression in the isolated cells. The result showed that the level of cathepsin B was increased significantly in the colonic epithelial cells from

the DSS-induced colitis model mice compared to those from control mice (**Supplementary Fig. 11e**). Mannose treatment significantly suppressed the expression of cathepsin B in colonic epithelial cells. Additionally, we evaluated the expression of cathepsin D and cathepsin L in the isolated epithelial cells from DSS-challenged mice treated with or without mannose. The result showed that mannose treatment also slightly downregulated DSS-induced increase of cathepsin D and cathepsin L (**Supplementary Fig. 11g**).

14. Fig 6C: I am not sure if increased CTSB activity really goes with the proposed mechanism being loss of lysosomal integrity. Loss of lysosomal integrity would be expected to decrease activity since activity is dependent on low pH. pH increases when lysosomal membranes are compromised. CTSB processing would also be affected since mature form is dependent on low pH. What are the sizes of the bands on western blot? Where is the mature form? Fractionation to check CTSB activity in lysosomes vs. cytosol could address this. Additionally, if lysosomal integrity is compromised, other enzymes in addition to CTSB should be affected.

Response: We would like to thank for this comment. As suggested, we evaluated the the acidification of lysosomes in colonic epithelial cells. The results showed that DSS treatment prevented the lysosomal acidification in colonic epithelial cells. Strikingly, mannose treatment rescued the acidification of lysosome in DSS-treated NCM460 cells (**Fig. 5e**).

Cathepsin B is produced from a larger precursor form, pro-cathepsin B, which runs at approximately 44 kDa on SDS-PAGE, and is proteolytically processed and glycosylated to form a mature two-chain protein composing of a heavy chain of 25-26 kDa and a light chain of 5 kDa. In the revised manuscript, we detected the pro-cathepsin B and active cathepsin B by western blot analysis. Additionally, the activity of cathepsin B in lysosomes and cytoplasm was detected (**Supplementary Fig. 11f**).

As required, the activity and expression of other lysosome enzymes, such as cathepsin L and D, were also measured. The result showed that mannose can downregulate DSS-induced increase of cathepsin D and cathepsin L (**Supplementary Fig. 11g**). Overall, the data indicated that mannose protects cells from lysosomal damage triggered by DSS.

15. Fig 7A: AMPK is usually activated in response to low energy - for example glucose starvation. Why is it activated in response to addition of mannose (sugar)?

Response: We thank this comment. AMPK plays a key role as a master regulator of cellular energy homeostasis. The kinase is activated in response to stresses that deplete cellular ATP supplies such as low glucose, hypoxia, and ischemia. Under glucose starvation, AMPK plays an essential role in maintaining redox homeostasis and cell survival through a variety of pathways¹⁹. Gonzalez et al., observed that mannose markedly reduced the production of lactate from glucose, indicating that mannose might interfere with glucose metabolism⁷. It is possible that mannose-induced AMPK phosphorylation might be associated with modulated glucose metabolism. We included a short discussion about this in the revised manuscript (**line 407-411**).

16. Fig S7: compC seems more effective than mannose as far as increasing LAMP 1/2 expression. Also, mannose fails to increase LAMP 1/2 expression in this experiment - inconsistent with Fig 5 and with hypothesis.

Response: We sincerely apologize for this error. The mannose and compound C treatment groups were mislabeled in the original submission. We corrected it in the revised manuscript. The results exhibited that compound C treatment could diminish the protective effect of mannose on expression of LAMP1 and LAMP2 (**Supplementary Fig. 12**).

References:

1. Erben, U. *et al.* A guide to histomorphological evaluation of intestinal inflammation in mouse models. *Int J Clin Exp Pathol* **7**, 4557-4576 (2014).
2. Sharma, V., Ichikawa, M. & Freeze, H.H. Mannose metabolism: more than meets the eye. *Biochem Biophys Res Commun* **453**, 220-228 (2014).
3. Spencer, D.M., Veldman, G.M., Banerjee, S., Willis, J. & Levine, A.D. Distinct inflammatory mechanisms mediate early versus late colitis in mice. *Gastroenterology* **122**, 94-105 (2002).
4. Keubler, L.M., Buettner, M., Hager, C. & Bleich, A. A Multihit Model: Colitis Lessons from the Interleukin-10-deficient Mouse. *Inflammatory bowel diseases* **21**, 1967-1975 (2015).
5. Jiang, C. *et al.* Targeted Imaging of Tumor-Associated Macrophages by Cyanine 7-Labeled Mannose in Xenograft Tumors. *Molecular imaging* **16**, 1536012116689499 (2017).

6. Moyer, M.P., Manzano, L.A., Merriman, R.L., Stauffer, J.S. & Tanzer, L.R. NCM460, a normal human colon mucosal epithelial cell line. *In Vitro Cell Dev Biol Anim* **32**, 315-317 (1996).
7. Gonzalez, P.S. *et al.* Mannose impairs tumour growth and enhances chemotherapy. *Nature* **563**, 719-723 (2018).
8. Ni, J., Chen, S.F. & Hollander, D. Effects of dextran sulphate sodium on intestinal epithelial cells and intestinal lymphocytes. *Gut* **39**, 234-241 (1996).
9. Araki, Y., Sugihara, H. & Hattori, T. In vitro effects of dextran sulfate sodium on a Caco-2 cell line and plausible mechanisms for dextran sulfate sodium-induced colitis. *Oncol Rep* **16**, 1357-1362 (2006).
10. Yuan, N. *et al.* Bafilomycin A1 targets both autophagy and apoptosis pathways in pediatric B-cell acute lymphoblastic leukemia. *Haematologica* **100**, 345-356 (2015).
11. Saint-Pol, A., Codogno, P. & Moore, S.E. Cytosol-to-lysosome transport of free polymannose-type oligosaccharides. Kinetic and specificity studies using rat liver lysosomes. *The Journal of biological chemistry* **274**, 13547-13555 (1999).
12. Ding, A.G. & Schwendeman, S.P. Acidic microclimate pH distribution in PLGA microspheres monitored by confocal laser scanning microscopy. *Pharmaceutical research* **25**, 2041-2052 (2008).

13. Circu, M. *et al.* Modulating lysosomal function through lysosome membrane permeabilization or autophagy suppression restores sensitivity to cisplatin in refractory non-small-cell lung cancer cells. *PloS one* **12**, e0184922 (2017).
14. Schwake, M., Schröder, B. & Saftig, P. Lysosomal Membrane Proteins and Their Central Role in Physiology. *Traffic* **14**, 739-748 (2013).
15. Fehrenbacher, N. *et al.* Sensitization to the lysosomal cell death pathway by oncogene-induced down-regulation of lysosome-associated membrane proteins 1 and 2. *Cancer Research* **68**, 6623-6633 (2008).
16. Yoshida, H. *et al.* Pepstatin A, an aspartic proteinase inhibitor, suppresses RANKL-induced osteoclast differentiation. *J Biochem* **139**, 583-590 (2006).
17. Barrett, A.J. *et al.* L-trans-Epoxy succinyl-leucylamido(4-guanidino)butane (E-64) and its analogues as inhibitors of cysteine proteinases including cathepsins B, H and L. *The Biochemical journal* **201**, 189-198 (1982).
18. Manabe, T. *et al.* Protective effect of nafamostat mesilate on cellular and lysosomal fragility of acinar cells in rat cerulein pancreatitis. *Int J Pancreatol* **12**, 167-172 (1992).
19. Ren, Y. & Shen, H.M. Critical role of AMPK in redox regulation under glucose starvation. *Redox Biol* **25**,

101154 (2019).

Reviewers' comments:

Reviewer #1 (Remarks to the Author):

All my previous concerns have been addressed appropriately

Reviewer #2 (Remarks to the Author):

I have focused on the authors' responses to my specific comments and critiques. I leave it to the other reviewers to comment on their reviews.

I found the revision very disappointing and frustrating for many reasons. Many instances showing lack of good preparation from the beginning that continues in this revision. Their response indicating that the data was re-evaluated after careful review means that it was not carefully reviewed for accuracy for the first submission. The references presented, such as #33, have nothing to do with the technique being cited. Again, not a careful review. Other times the line notations were not correct. Overall poor preparation and attention to details not acceptable.

The manuscript or the point by point response do not demonstrate understanding of some basic issues of Glycobiology. In response to my concern about a "mannose antibody", they cite a kit from a company which gives very little information about the antibody, how it was made or what it recognizes, other than "mannose". To my knowledge, no mannose-specific antibody has been published or peer reviewed. The company states that they use a competition assay against biotinylated mannose. Making such a compound would require a primary amino group on mannose. It does not contain an amino group. Mannosamine would, but that is a totally different compound and in no way can be considered a substitute for mannose. The labeling kit is not valid in my opinion. I don't know of any lab that used it previously to measure mannose.

In describing the use of coumarin-labeled mannose, they cite a chemical preparation procedure which specifically states they label mannosamine. The authors seem to believe that mannose and mannosamine are equivalent. They are not at all in physiology, metabolism or chemical properties.

I don't see any way to salvage these inherently flawed data.

The use of various concentrations of mannose from 5-50 mM, the inclusion or exclusion of glucose in different experiments make it difficult to assess and accurately compare these extreme conditions and their metabolic impacts on cells.

Reviewer #3 (Remarks to the Author):

Authors have responded to all comments. As a caution, many responses acknowledge that a mistake was made in the original version - I would suggest greater care prior to submission in future/

Reviewer #4 (Remarks to the Author):

The manuscript by Dong, et al is overall improved following revisions. Inclusion of appropriate in vivo controls and addition of CTSB KO cells in particular strengthen the manuscript. However, some serious issues remain. In particular, this pertains to the assessment of lysosomal integrity, which has not been demonstrated at all. Without further appropriate and rigorous assessment any conclusion about LMP and release of CTSB is invalid. Based on the presented data I would be comfortable with a conclusion that CTSB is involved in the mediation of downstream effects but not with the claim that this is due to its release into the cytosol, which remains unsubstantiated.

Specific major concerns:

- While the authors improved the quality of many IF and histology images some remain inadequate. In particular, 5c, e, f remain too low magnification to appreciate intracellular colocalization.
- 5f: the authors performed dextran leakage experiment as requested, however, the data is misinterpreted. It clearly demonstrates that while lysosomal pH (lysotracker) is affected, dextran remains in puncta that look like intact lysosomes. No leakage at all. This is contrary to their conclusion that DSS treatment induces LMP. The quantification demonstrates decrease in colocalization but the reason for this is that since lysotracker staining is affected by DSS treatment, it cannot be used as lysosomal marker in this experiment. Another marker is necessary.
- Methods used for image quantification including colocalization need to be thoroughly described in M&M.
- S11g: this is the only figure demonstrating potential presence of CTSB in the cytosol (and only in vitro). It has to be moved to primary figures and repeated in vivo +/- DSS and +/- mannose.

Other issues:

- Statistics: the authors claim that 3 independent experiments were performed in vivo and each had 6-8 mice (reply to reviewer letter). If that's the case, why is the total # of animals 6-8 rather than 18-24? Did you mean 3 independent experiments in vitro and 6-8 mice in vivo? Please make sure this is appropriately reflected in M&M
- Please specify sex and age of the mice used

Detailed responses to the reviewer' comments (NCOMMS-19-34782A)

We are grateful for the reviewers' positive comments, and appreciate the valuable suggestions of the expert reviewers. Reviewer 1 and reviewer 3 were satisfied with our responses. The major concern from reviewer 2 is about the experiments with mannose analysis and labeling. In the resubmitted manuscript, we conducted liquid chromatography/tandem mass spectrometry (LC-MS/MS) analysis to determine the circulating mannose levels. Also, we provided the structure diagram of the fluorescein-labeled mannose and briefly discussed the monosaccharide tracking in the discussion section. For the comments from reviewer 4, we conduct additional experiments to determine cathepsin B expression in the cytosol of colon epithelial cells *in vivo*. Here we address the concerns of the reviewers, point by point.

Reviewer #1 (Remarks to the Author):

All my previous concerns have been addressed appropriately.

Response: We highly appreciate the valuable comments by the reviewer, which have significantly helped us improve the manuscript.

Reviewer #2 (Remarks to the Author):

I have focused on the authors' responses to my specific comments and critiques. I leave it to the other reviewers to comment on their reviews.

I found the revision very disappointing and frustrating for many reasons. Many instances showing lack of good preparation from the beginning that continues in this revision. Their response indicating that the data was re-evaluated after careful review means that it was not carefully reviewed for accuracy for the first submission. The references presented, such as #33, have nothing to do with the technique being cited. Again, not a careful review. Other times the line notations were not correct. Overall poor preparation and attention to detail is not acceptable.

The manuscript or the point by point response do not demonstrate understanding of some basic issues of Glycobiology. In response to my concern about a "mannose antibody", they cite a kit from a company which gives very little information about the antibody, how it was made or what it recognizes, other than "mannose". To my knowledge, no mannose-specific antibody has been published or peer reviewed. The company states that they use a competition assay against biotinylated mannose. Making such a compound would require a primary amino group on mannose. It does not contain an amino group. Mannosamine would, but that is a totally different compound and in no way can be considered a substitute for mannose. The labeling kit is not valid in my opinion. I don't know of any lab that used it previously to measure mannose.

In describing the use of coumarin-labeled mannose, they cite a chemical preparation procedure which specifically states they label mannosamine. The authors seem to believe that mannose and mannosamine are equivalent. They are not at all in physiology, metabolism or chemical properties.

I don't see any way to salvage these inherently flawed data.

The use of various concentrations of mannose from 5-50 mM, the inclusion or exclusion of glucose in different experiments make it difficult to assess and accurately compare these extreme conditions and their metabolic impacts on cells.

Responses:

1. The references presented, such as #33, have nothing to do with the technique being cited.

Response: We would like to explain that the compound E-64d, a synthetic analog of E-64 that used in reference #33, is also known as aloxistatin which we used as an irreversible membrane-permeable cysteine protease inhibitor in our experiment. In the resubmitted manuscript, we also included another reference for aloxistatin ¹. (ref 35, line 266)

2. In response to my concern about a “mannose antibody”, they cite a kit from a company which gives very little information about the antibody, how it was made or what it recognizes, other than "mannose". To my knowledge, no mannose-specific antibody has been published or peer reviewed. The company states that they use a competition assay against biotinylated mannose. Making such a compound would require a primary amino group on mannose. It does not contain an amino group. Mannosamine would, but that is a totally different compound and in no way can be considered a substitute for mannose. The labeling kit is not valid in my opinion. I don't know of any lab that used it previously to measure mannose.

Response: In our study, the serum mannose levels were evaluated by a competitive enzyme immunoassay. The mannose ELISA kit was purchased from Aviva Systems Biology company (OKEH02595, Aviva Systems Biology, San Diego, CA, USA). Indeed, another group had used this kit to determine serum mannose concentration (Feng D, *et al.* Front Endocrinol. 2019)². They used the ELISA kit to measure the serum mannose level in healthy control subjects and patients with polycystic ovary syndrome.

The primary concern from the reviewer is about the anti-mannose antibody. After intensive consulting with the technical staff of Aviva Systems Biology company, we got detailed information about the preparation of the anti-mannose antibody. Firstly, the mannose was conjugated with a carrier protein for immunization. Once synthesized, the mannose-carrier conjugate was used to immunize animals for antibody production. The anti-mannose antibody was purified by affinity chromatography with mannose-agarose. The anti-mannose antibody specifically bound with the mannose-agarose with high affinity, while other antibodies did not. Finally, the anti-mannose antibody was eluted. Each batch of the anti-mannose antibody will undergo strict quality control.

To further validate the result about the circulating mannose concentration in humans and mice, we conducted liquid chromatography/tandem mass spectrometry (LC-MS/MS) analysis to determine the circulating mannose levels, as previously described ³. These data were similar to the data from ELISA analysis. In the resubmitted manuscript, the data using LC-MS/MS analysis were presented instead of the ELISA results (Figure 1a, 1b, 1c, 1e, and 1g). The information was included in the method section (lines 595-604).

The detailed method is as follows,

a. Instruments and reagents

A Nexera UHPLC LC-30A (Shimadzu) equipped with a Triple Quad™ 4500 (AB SCIEX), crosslinked HPLC column (3.0mm.I.D.×100 mm, 1.7 μm, waters) was used in this study. D-mannose (≥99 % pure), D-mannose-1-¹³C (99 atom % ¹³C, 99% pure, internal standard, IS), and HPLC grade acetonitrile and water were purchased from Sigma-Aldrich (St. Louis, MO).

b. Methods

b1. Chromatographic conditions

Mobile phase A is the aqueous phase (water containing 0.1% ammonia); mobile phase B is the organic phase (acetonitrile containing 0.1% ammonia). The flow rate is 0.35 ml/min, and the elution gradient is shown in Tab. 1.

Table 1 Chromatographic elution gradient

Time, min	Module	Events	Parameter
0.0	Pumps	Pump B Conc.	85%
9	Pumps	Pump B Conc.	85%
9.3	Pumps	Pump B Conc.	60%
11.0	Pumps	Pump B Conc.	60%
11.1	Pumps	Pump B Conc.	85%
14.0	System Controller	Stop	

b2. Mass spectrometry conditions

The mass spectrometry conditions were showed in Tab. 2.

Table 2 Mass spectrometry conditions of the analyte

Items	Parameter
Scan Type	MRM (MRM)
Polarity	Negative
Ion Source	Turbo Spray
Q1 Mass (amu)/ Q3Mass (amu)	179.1/89; 179.1/59
DP (V)	30
CE (eV)	11
CUR (L/min)	40
IS (V)	-4500
TEM (°C)	350
GS1 (L/min)	50

GS2 (L/min)	40
EP (V)	10
CXP (V)	10

b3. Sample preparation

An aliquot (50 μ l) of serum sample was first mixed with 200 μ l methanol-acetonitrile-water (v:v:v, 2:2:1) mixed solvent containing 5 μ g/ml D-mannose-1-¹³C IS. The mixture was then vortex mixed for 30 seconds and centrifuged for 30 minutes at 20,000 \times g at room temperature. After centrifugation an aliquot (10 μ l) of the supernatant was taken for LC-MS/MS analysis.

b4. Standard curve and sample test results

A series of standard samples of mannose were prepared by diluting the stock solution with water to obtain the following concentrations: 0.2, 1, 5, 10, 25, and 50 μ g/ml. Fig. 1 shows standard curve line of mannose. Fig. 2 shows an overlaid mannose (50 μ g/ml) and glucose (50 μ g/ml) spectrum. The chromatograph of human plasma mannose is shown in Fig 3.

Figure 1. The standard curve line of mannose.

The standard curve line of mannose (0.2, 1, 5, 10, 25, and 50 μ g/ml) using LC-MS/MS.

Figure 2. The chromatogram of mannose and glucose.
The chromatogram of mannose (6.88 min) and glucose (7.95 min).

Figure 3. The chromatogram of mannose in human plasma

3. In describing the use of coumarin-labeled mannose, they cite a chemical preparation procedure, which specifically states they label mannosamine. The authors seem to believe that mannose and mannosamine are equivalent. They are not at all in physiology, metabolism or chemical properties.

Response: Mannose, an isomer of glucose, is a monosaccharide sugar that is hardly labeled. Indeed, the condensation reaction between the amino group and the carboxyl group is easier than the esterification reaction between the hydroxyl group and the carboxyl group. Therefore, an amino group is usually introduced to improve the binding activity of the monosaccharide. For example, 2-NBDG, 2-(N-(7-Nitrobenz-2-oxa-1,3-diazol-4-yl)Amino)-2-Deoxyglucose, is a fluorescent glucose analog that has been used to monitor glucose uptake in live cells (Yamada K, *et al.* Nat Protoc. 2007)
4. It consists of a glucosamine molecule substituted with a 7-nitrobenzofurazan fluorophore at

its amine group. (Fig. 4a). Jiang *et al.*, also used the same method to label mannose (Cy7-labeled D-mannosamine) in their experiments in order to monitor mannose uptake in tumor-associated macrophages (TAMs)⁵. We believe that coumarin-labeled mannosamine could be used as a fluorescent indicator for the monitoring of mannose uptake in living cells.

The compound, coumarin-labeled-mannosamine, was synthesized by Ruixi Biological Technology Co., Ltd (Xi'an, China). Coumarin (10 mM) was conjugated to the terminal amino group of D-mannosamine hydrochloride (20 mM) in a 1:1 mixture (0.5 ml) of acetonitrile and sodium borate buffer (0.1 M, pH 8.5). After it was stirred at 4 °C for 2 hours in the dark, the reaction mixture was placed in room temperature storage for 1 hour. The products were isolated and purified by semipreparative, reverse-phase high-performance liquid chromatography. The mobile phase changed from 60% solvent A (0.1% trifluoroacetic acid in water) and 40% solvent B (0.1% trifluoroacetic acid in 80% aqueous acetonitrile) to 100% solvent B during 30 minutes at a flow rate of 2 ml/min. The ultraviolet detector was set to 215 nm for the experiments, and its structure diagram is shown in Fig. 4b. So far, it is a common procedure to introduce an amino group during the process of labeling monosaccharide with fluorescence.

Figure 4. The structure diagram of fluorescein-labeled monosaccharide.

The structure diagram of 2-NBDG (a) and coumarin-labeled-mannose (b).

In this resubmitted manuscript, we provided the structure of the fluorescein-labeled mannose in the supplementary figure 7a. Meanwhile, we add a short discussion about the procedure for mannose tracking in the discussion section (line 396-402).

4. The use of various concentrations of mannose from 5-50 mM, the inclusion or exclusion of glucose in different experiments make it difficult to assess and accurately compare these extreme conditions and their metabolic impacts on cells.

Response: We are sorry for this confusion. In our experiments, we used different cells for determining the protective effect of mannose on epithelial cell damage. We, therefore, tested various concentrations of mannose and determined the optimal mannose concentration for each cell. We have determined that 5-50 mM mannose is valid for both NCM460 cells and mouse primary colonic epithelial cells. For the mouse primary colonic epithelial cells, 5 mM mannose is the optimal concentration to protect DSS-induced epithelial cell damage. For the NCM460 cells, 25 mM mannose was used through all the experiments.

Reviewer #3 (Remarks to the Author):

Authors have responded to all comments. As a caution, many responses acknowledge that a mistake

was made in the original version - I would suggest greater care prior to submission in future.

Response: We highly appreciate the valuable comments by the reviewer, which have significantly helped us improve the manuscript. Again, we deeply apologize for the error in the first original manuscript.

Reviewer #4 (Remarks to the Author):

The manuscript by Dong, et al is overall improved following revisions. Inclusion of appropriate in vivo controls and addition of CTSB KO cells in particular strengthen the manuscript. However, some serious issues remain. In particular, this pertains to the assessment of lysosomal integrity, which has not been demonstrated at all. Without further appropriate and rigorous assessment any conclusion about LMP and release of CTSB is invalid. Based on the presented data I would be comfortable with a conclusion that CTSB is involved in the mediation of downstream effects but not with the claim that this is due to its release into the cytosol, which remains unsubstantiated.

Specific major concerns:

- While the authors improved the quality of many IF and histology images some remain inadequate. In particular, 5c, e, f remain too low magnification to appreciate intracellular colocalization.
- 5f: the authors performed dextran leakage experiment as requested, however, the data is misinterpreted. It clearly demonstrates that while lysosomal pH (lysotracker) is affected, dextran remains in puncta that look like intact lysosomes. No leakage at all. This is contrary to their conclusion that DSS treatment induces LMP. The quantification demonstrates decrease in colocalization but the reason for this is that since lysotracker staining is affected by DSS treatment, it cannot be used as lysosomal marker in this experiment. Another marker is necessary.
- Methods used for image quantification including colocalization need to be thoroughly described in M&M.
- S11g: this is the only figure demonstrating potential presence of CTSB in the cytosol (and only in vitro). It has to be moved to primary figures and repeated in vivo +/- DSS and +/- mannose.

Other issues:

- Statistics: the authors claim that 3 independent experiments were performed in vivo and each had 6-8 mice (reply to reviewer letter). If that's the case, why is the total # of animals 6-8 rather than 18-24? Did you mean 3 independent experiments in vitro and 6-8 mice in vivo? Please make sure this is appropriately reflected in M&M
- Please specify sex and age of the mice used

Responses:

1. While the authors improved the quality of many IF and histology images some remain inadequate. In particular, 5c, e, f remain too low magnification to appreciate intracellular colocalization.

Response: We thank the reviewer for this suggestion. Much sharper images (Fig. 5c, d, e, f, Supplementary Fig. 7b, Supplementary Fig. 14b) have provided as requested in the resubmitted manuscript.

2. The authors performed dextran leakage experiment as requested, however, the data is misinterpreted. It clearly demonstrates that while lysosomal pH (lysotracker) is affected, dextran remains in puncta that look like intact lysosomes. No leakage at all. This is contrary to their conclusion that DSS treatment induces LMP. The quantification demonstrates decrease in colocalization but the reason for this is that since lysotracker staining is affected by DSS treatment, it cannot be used as lysosomal marker in this experiment. Another marker is necessary.

Response: We would like to thank the reviewer for this insightful comment. We agree with the reviewer that lysotracker is not the best choice for monitoring lysosome to determine the dextran leakage in our research because lysotracker staining is affected by DSS in NCM460 cells. In the resubmitted manuscript, we used anti-LAMP2 antibody instead of lysotracker as a lysosomal marker to evaluate the dextran leakage according to the previous study ⁶. The result showed that mannose treatment increased the localization of FITC-dextran in lysosomes in DSS-treated cells (Fig. 5f).

3. Methods used for image quantification including colocalization need to be thoroughly described in M&M.

Response: We will provide detailed information about the image quantification in the method and material section (lines 681-686).

4. S11g: this is the only figure demonstrating potential presence of CTSB in the cytosol (and only *in vitro*). It has to be moved to primary figures and repeated *in vivo* +/- DSS and +/- mannose.

Response: We would like to thank this comment. As suggested, the S11g was removed to Figure 6h in the resubmitted manuscript. Additionally, we evaluated the levels of cathepsin B in the lysosome and cytoplasm from DSS-stimulated epithelial cells by western blotting (Figure 6g). In our previous submission, the immunohistochemical staining and immunoblotting analysis showed the increased cathepsin B expression in the epithelial cells in the colon tissues from DSS-treated mice, and mannose treatment sharply suppressed the expression of cathepsin B in DSS-treated mice. To further validate the presence of cathepsin B in the cytosol of colon epithelial cells *in vivo*, we determined the cathepsin B expression in the epithelial cells from colitis mice treated with or without mannose. The result showed that mannose reduced the level (Supplementary Fig. 12g) and activity (Supplementary Fig. 12h) of cathepsin B in the cytoplasm of the epithelial cells from colitis mice. Overall, our data suggest that mannose treatment inhibited the level and activity of cytoplasmic cathepsin B in colonic epithelial cells both *in vitro* and *in vivo* upon inflammatory stimulation.

5. Statistics: the authors claim that 3 independent experiments were performed *in vivo* and each had 6-8 mice (reply to reviewer letter). If that's the case, why is the total # of animals 6-8 rather than 18-24? Did you mean 3 independent experiments *in vitro* and 6-8 mice *in vivo*? Please make sure this is appropriately reflected in M&M.

Response: The detailed information will be provided as suggested. In our study, three independent experiments were performed for all *in vivo* experiments, and the number of animals in each group was 6-8 in one experiment. In the result section, data from one representative experiment of three independent experiments are presented. We added the information in all the figure legend.

6. Please specify sex and age of the mice used.

Response: The detailed information was provided in the method section as suggested (line 454-

457).

References:

1. Murray EJ, Grisanti MS, Bentley GV, Murray SS. E64d, a membrane-permeable cysteine protease inhibitor, attenuates the effects of parathyroid hormone on osteoblasts in vitro. *Metabolism* **46**, 1090-1094 (1997).
2. Feng D, *et al.* Elevated Serum Mannose Levels as a Marker of Polycystic Ovary Syndrome. *Frontiers in endocrinology* **10**, 711 (2019).
3. Taguchi T MI, Mizutani T, Nakajima H, Fukumura Y, Kobayashi I, Yabuuchi M, Miwa I. Determination of D-mannose in plasma by HPLC. *Clin Chem* **49**, 181-183 (2003).
4. Yamada K, Saito M, Matsuoka H, Inagaki N. A real-time method of imaging glucose uptake in single, living mammalian cells. *Nature protocols* **2**, 753-762 (2007).
5. Jiang C, Cai H, Peng X, Zhang P, Wu X, Tian R. Targeted Imaging of Tumor-Associated Macrophages by Cyanine 7-Labeled Mannose in Xenograft Tumors. *Mol Imaging* **16**, 1536012116689499 (2017).
6. Lajoie P, Guay G, Dennis JW, Nabi IR. The lipid composition of autophagic vacuoles regulates expression of multilamellar bodies. *J Cell Sci* **118**, 1991-2003 (2005).

Again, we would like to thank the four professional reviewers for your recognition of our work. Due to the tight schedule, we may have some problems with some details, for which we are deeply sorry. We now believe that all the comments are well addressed, and look forward to hearing from you.

Sincerely yours,
Daming Zuo,
Ph. D. Professor
Institute of Molecular Immunology,
School of Laboratory Medicine and Biotechnology,
Southern Medical University,
Guangzhou, Guangdong, 510515, P.R.China.
Phone: 86 20 6164 8552
Fax: 86 20 6164 8221
E-mail: zdaming@smu.edu.cn

Reviewers' comments:

Reviewer #2 (Remarks to the Author):

The authors have tried to thoroughly answer the problems I had with the paper. Clearly, mannose is having some effects on DSS-induced colitis and in cell models meant to mimic the intestine. This seems to be a novel and surprising finding.

The measurement of mannose concentration by LC-MS is solid and acceptable.

Despite the detailed preparation methods and reference to the use of coumarin-labeled mannosamine by Jiang et al, I remain unconvinced that this compound can serve as a measure of either mannose uptake or distribution within the cell. It is not metabolized like mannose, which is quickly used for glycosylation or catabolized. Interpretations using it are not reliable, in my opinion.

MINOR POINTS

in the Introduction, mannose is said to be in many plants and fruits, giving the impression that it is a source of mannose for the diet. Most of those sources have beta-linked mannose and it is degraded by intestinal flora; very little is available for the host.

Some of the bar graphs are very difficult to see, especially the control bars to the extreme left. Albumin loss into the intestinal lumen using the indicated ELISA kit is not appropriate for fecal samples according to the Abcam company literature. Usually alpha-1 antitrypsin is used since it is not degraded like albumin

Reviewer #4 (Remarks to the Author):

The authors appropriately addressed all my questions. The only suggestion I have is to move either Fig S12g or S12h demonstrating in vivo CTSD leakage to the main figures as these data are essential to support the conclusions.

Detailed responses to the reviewer' comments (NCOMMS-19-34782)

We are grateful for the reviewers' positive comments, and appreciate the valuable suggestions of the expert reviewers. The major concern from reviewer 2 is about the experiments with mannose labeling. In the resubmitted manuscript, we conducted liquid chromatography/tandem mass spectrometry (LC-MS/MS) analysis to determine the mannose levels in the lysosome of colon epithelial cells. We also conduct additional experiments to determine the concentration of fecal alpha-1 antitrypsin. Here we address the concerns of the reviewers, point by point.

Reviewer #2 (Remarks to the Author):

The authors have tried to thoroughly answer the problems I had with the paper. Clearly, mannose is having some effects on DSS-induced colitis and in cell models meant to mimic the intestine. This seems to be a novel and surprising finding.

The measurement of mannose concentration by LC-MS is solid and acceptable.

Despite the detailed preparation methods and reference to the use of coumarin-labeled mannosamine by Jiang et al, I remain unconvinced that this compound can serve as a measure of either mannose uptake or distribution within the cell. It is not metabolized like mannose, which is quickly used for glycosylation or catabolized. Interpretations using it are not reliable, in my opinion.

MINOR POINTS

in the Introduction, mannose is said to be in many plants and fruits, giving the impression that it is a source of mannose for the diet. Most of those sources have beta-linked mannose and it is degraded by intestinal flora; very little is available for the host.

Some of the bar graphs are very difficult to see, especially the control bars to the extreme left.

Albumin loss into the intestinal lumen using the indicated ELISA kit is not appropriate for fecal samples according to the Abcam company literature. Usually alpha-1 antitrypsin is used since it is not degraded like albumin

Responses:

1. Despite the detailed preparation methods and reference to the use of coumarin-labeled mannosamine by Jiang et al, I remain unconvinced that this compound can serve as a measure of either mannose uptake or distribution within the cell. It is not metabolized like mannose, which is quickly used for glycosylation or catabolized. Interpretations using it are not reliable, in my opinion.

Response: We appreciate this comment by the reviewer. As previously described, the LC-MS/MS-based metabolomics analysis could be applied to tissue, cells, and subcellular fractions, such as lysosome and mitochondria^{1, 2, 3}. In the resubmitted manuscript, the lysosome and mitochondria fractions of the mannose-treated cells were isolated, and LC-MS/MS analysis was subsequently conducted to determine mannose distribution in the subcellular fractions. D-mannose-1-¹³C acts as an internal standard. The result showed that mannose was mainly located in lysosomes, not mitochondria, in NCM460 cells (**fig. 5d**), whereas the inhibition of mannose transporters blocked

mannose localization to lysosomes (**supplementary fig. 7c**).

2. In the Introduction, mannose is said to be in many plants and fruits, giving the impression that it is a source of mannose for the diet. Most of those sources have beta-linked mannose and it is degraded by intestinal flora; very little is available for the host.

Response: We thank and agree with the reviewer for this comment. A proper description of mannose was provided in the resubmitted manuscript (**line 75**).

3. Some of the bar graphs are very difficult to see, especially the control bars to the extreme left. Albumin loss into the intestinal lumen using the indicated ELISA kit is not appropriate for fecal samples according to the Abcam company literature. Usually alpha-1 antitrypsin is used since it is not degraded like albumin.

Response: We would like to thank the reviewer for these suggestions. We have changed the bars into heavy lines. Moreover, we conducted ELISA analysis (Abcam, ab267809) to determine the α 1-antitrypsin levels in mice feces. In the resubmitted manuscript, the fecal levels of α 1-antitrypsin were presented instead of fecal albumin levels (**fig 2e, 3d, 7i, and supplementary fig. 6d**). The detailed information was included in the method section (**lines 591-598**).

Reviewer #4 (Remarks to the Author):

The authors appropriately addressed all my questions. The only suggestion I have is to move either Fig S12g or S12h demonstrating in vivo CTSD leakage to the main figures as these data are essential to support the conclusions.

Responses: We thank the reviewer for this suggestion. As suggested, figure S12g and S12h have removed to figure 6i and figure 6j in the resubmitted manuscript, respectively (**fig 6i and 6j**).

References:

1. Thelen M, Winter D, Braulke T, Gieselmann V. SILAC-Based Comparative Proteomic Analysis of Lysosomes from Mammalian Cells Using LC-MS/MS. *Methods Mol Biol* **1594**, 1-18 (2017).
2. Borah K, *et al.* A quantitative LC-MS/MS method for analysis of mitochondrial α -specific oxysterol metabolism. *Redox Biol* **36**, 101595 (2020).
3. Ponnaiyan S, Akter F, Singh J, Winter D. Comprehensive draft of the mouse embryonic fibroblast lysosomal proteome by mass spectrometry based proteomics. *Sci Data* **7**, 68 (2020).

Reviewers' comments:

Reviewer #2 (Remarks to the Author):

I remain unconvinced about the validity of using a tagged derivative of mannose amine as a surrogate for mannose.

However, I would accept the positive results of a critical experiment. The authors can label cells with either radioactive or ¹³C-labeled mannose along with their tagged mannose amine. Fractionate cells and present a complete distribution of both markers along. Then show ~10x enrichment of lysosomes using a standard lysosomal marker (via western blot or enzyme assay) and most importantly, a similar distribution of the tagged mannose amine. If those data show co-fraction of the two markers that accounts for 50% of the two labels, I will be satisfied and convinced that their claims of lysosomal involvement are correct. I will have no further objections to this impressive and important study.

Reviewer #2 (Remarks to the Author):

I remain unconvinced about the validity of using a tagged derivative of mannose amine as a surrogate for mannose.

However, I would accept the positive results of a critical experiment. The authors can label cells with either radioactive or ^{13}C -labeled mannose along with their tagged mannose amine. Fractionate cells and present a complete distribution of both markers along. Then show ~10x enrichment of lysosomes using a standard lysosomal marker (via western blot or enzyme assay) and most importantly, a similar distribution of the tagged mannose amine. If those data show co-fraction of the two markers that accounts for 50% of the two labels, I will be satisfied and convinced that their claims of lysosomal involvement are correct. I will have no further objections to this impressive and important study.

Response: We greatly appreciate this comment by the reviewer. In the previous submission, we used fluorescently labeled mannosamine to monitor the distribution of mannose in cells.

Here, we used ^{13}C -labeled mannose to monitor the distribution of mannose in colonic epithelial cells following a previous reference ¹. Moreover, we quantified the level of mannose in the cells and lysosomes via LC-MS/MS analysis using commercial mannose-1- ^{13}C as a standard. The result showed that mannose was mainly located in lysosomes, not mitochondria, in NCM460 cells (**fig. 5c, d**). The inhibition of mannose transporters blocked mannose localization in lysosomes (**supplementary fig. 7a**). Additionally, the standard lysosomal marker and mitochondrial marker were assessed via western blotting, and the results were included.

To clear up the confusion between mannose and mannosamine, we have replaced the data with new data from the radioactive-labeled mannose in the revised manuscript.

The detailed method is as follows,

a. Instruments and reagents

A Nexera UHPLC LC-30A (Shimadzu) equipped with a Triple QuadTM 4500 (AB SCIEX), crosslinked HPLC column (3.0mmI.D.×100 mm, 1.7 μm , waters) was used in this study. D-mannose-1- ^{13}C (99 atom % ^{13}C , 99% pure), D-glucose-1- ^{13}C (99 atom % ^{13}C , 99% pure), and HPLC grade acetonitrile and water were purchased from Sigma-Aldrich (St. Louis, MO).

b. Methods

b1. Chromatographic conditions

Mobile phase A is the aqueous phase (water containing 0.1% ammonia); mobile phase B is the organic phase (acetonitrile containing 0.1% ammonia). The flow rate is 0.35 ml/min, and the elution gradient is shown in Tab. 1.

Table 1 Chromatographic elution gradient

Time, min	Module	Events	Parameter
0.0	Pumps	Pump B Conc.	85%
9	Pumps	Pump B Conc.	85%
9.3	Pumps	Pump B Conc.	60%
11.0	Pumps	Pump B Conc.	60%
11.1	Pumps	Pump B Conc.	85%
14.0	System Controller	Stop	

b2. Mass spectrometry conditions

The mass spectrometry conditions were showed in Tab. 2.

Table 2 Mass spectrometry conditions of the analyte

Items	Parameter
Scan Type	MRM (MRM)
Polarity	Negative
Ion Source	Turbo Spray
Q1 Mass (amu)/ Q3Mass (amu)	180.0/89; 180.0
DP (V)	30
CE (eV)	11
CUR (L/min)	40
IS (V)	-4500
TEM (°C)	350
GS1 (L/min)	50
GS2 (L/min)	40
EP (V)	10
CXP (V)	10

b3. Sample preparation

After incubated with mannose-1-¹³C or glucose-1-¹³C, the lysosome, mitochondria and the whole cells of NCM460 cells were isolated, then the fraction were extracted with ethanol:water (1:1). The mixture was then centrifuged for 30 minutes at 20,000 × g at room temperature. After centrifugation, an aliquot (10 μl) of the supernatant was taken for LC-MS/MS analysis.

b4. Standard curve results

A series of standard samples of mannose-1-¹³C or glucose-1-¹³C were prepared by diluting the stock solution with water to obtain the following concentrations: 10, 50, 250, 500, and 1000 ng/ml. Fig. 1 shows standard curve line of mannose-1-¹³C. Fig. 2 shows standard curve line of glucose-1-¹³C.

Figure 1. The standard curve of mannose-1-¹³C.

The standard curve line of mannose-1-¹³C (10, 50, 250, 500, and 1000 ng/ml) using LC-MS/MS.

Figure 2. The standard curve of glucose-1-¹³C.

The standard curve line of glucose-1-¹³C (10, 50, 250, 500, and 1000 ng/ml) using LC-MS/MS.

Reference:

1. Duran JM, Cano M, Peral MJ, Ilundain AA. D-mannose transport and metabolism in isolated enterocytes. *Glycobiology* **14**, 495-500 (2004).

Reviewers' comments:

Reviewer #2 (Remarks to the Author):

This long saga continues with a response to my original skepticism about the use of tagged Mannoseamine as a surrogate for mannose. I explained that I would accept the authors' contention if they showed that either radioactive ^3H -Mannose or, preferably, ^{13}C -labeled mannose was co-fractionates with tagged mannoseamine and that it accounts for a substantial portion of the total of both labeled molecules. To their credit, the authors partially tried to respond the comment. They did not do a side-by-side comparison of the two molecules as I requested. In fact there was no mention of using tagged mannoseamine. Apparently it did not co-fractionate or they did not do the experiment I suggested. Instead they chose a modified version of this experiment and a rather simplified analysis. However, this experiment raises other significant issues.

First, is the concentration of ^{13}C -mannose used is 1 μM while the cellular experiments use 25mM mannose. A 25,000 fold difference is not an acceptable alteration since 25mM mannose could be handled much differently than 1 μM . The second issue is the time of incubation--only 5 min-- and third is the decision to look at only mitochondria, lysosomes and total signal. Mannose entering the cell is usually phosphorylated to man-6-p almost instantly. The authors do not account for all of the potential metabolism that must surely occur in any cell metabolizing mannose. This has been shown over decades of experiments. These three issues offer little meaningful to support their claim.

However, there is another way to look at this and it lies in the details of experimental protocol. Cellular experiments use 25mM mannose. If the authors repeat this experiment using 25 mM ^{13}C mannose and show the lysosomes substantially ACCUMULATES mannose at multiple times over the 24 hours, that would alleviate some issues. I assume it is not the 5 min time frame that resets the potential benefits of mannose, but rather the long-term presence of mannose (in the lysosomes?) which could potentially disturb lysosomal homeostasis and integrity leading to the release of Cathepsin B. This may only be true in the presence of DSS, a highly negatively charged macromolecule which would be trafficked to the lysosome by endocytosis or macropinocytosis. The combination of DSS+mannose may ameliorate the lysosomal damage induced by DSS. A time course of over 24 hr showing progressive release of Cathepsin B from the lysosomes into the cytoplasm, along with maintenance of high levels of ^{13}C -mannose within lysosomes, and an accounting for ^{13}C -mannose derived macromolecules (mostly glycoproteins) and small metabolites would lend strength to explaining the authors very interesting results.

Assuming an internal intestinal volume of $\sim 1\text{ml}$, a gavage of 500 μg mannose per gm body weight calculates to 12mg in a 25gm mouse, or a concentration of $\sim 60\text{mM}$. So choosing 25 mM mannose for cellular experiments is not unreasonable choice to mimic the mouse experiments.

Again, the dramatic results the authors show for effects on mice are impressive, however using a daily gavage of mannose would not be a welcomed approach by patients.

Minor point: only male mice were used. This is a potential problem and is open to criticism at several levels.

Detailed responses to the reviewer's comments (NCOMMS-19-34782E)

We are grateful for the reviewer's positive comments and appreciate the valuable suggestions of the expert reviewer. The major concern from reviewer 2 is whether tagged mannosamine as a surrogate for mannose or not. In the resubmitted manuscript, the ^{13}C -labeled mannose and ^{13}C -labeled mannosamine was used to track the mannose transport in colonic epithelial cells. We conducted liquid chromatography/tandem mass spectrometry (LC-MS/MS) analysis to determine the presence of mannose-1- ^{13}C and mannosamine-1- ^{13}C in the lysosome of colon epithelial cells. Here we address the concerns of the reviewer.

Reviewer #2 (Remarks to the Author):

1. This long saga continues with a response to my original skepticism about the use of tagged Mannosamine as a surrogate for mannose. I explained that I would accept the authors' contention if they showed that either radioactive 3H-Mannose or, preferably, ^{13}C -labeled mannose was co-fractionates with tagged mannosamine and that it accounts for a substantial portion of the total of both labeled molecules. To their credit, the authors partially tried to respond the comment. They did not do a side-by-side comparison of the two molecules as I requested. In fact there was no mention of using tagged mannosamine. Apparently it did not co-fractionate or they did not do the experiment I suggested. Instead they chose a modified version of this experiment and a rather simplified analysis. However, this experiment raises other significant issues.

Responses:

We thank the reviewer for this valuable suggestion.

We are sorry that we misunderstood the initial reviewer's request. We complied with the protocol in Duran et al. (doi: 10.1093/glycob/cwh059) to monitor mannose transport in the previous submission. Considering that the main conclusion of our study is that mannose attenuates colitis by protecting intestinal epithelial barrier function, we thought it might not be necessary to explore the difference between mannose and mannosamine for this study. Additionally, the ^{13}C -labeled mannose was mainly located in the lysosome of colonic epithelial cells determined by LC-MS/MS analysis. Therefore, we removed the data collected by fluorescent-labeled mannosamine in the previous submission.

We totally agree with the reviewer. As suggested, we monitor the cellular distribution of ^{13}C -labeled mannose (25 mM) over 24 hours after incubation in the presence of DSS. The results showed that mannose was mainly located in lysosomes, not mitochondria, in the colonic epithelial cells.

Additionally, we incubated the NCM460 cells with the same amount (12.5 mM each) of ^{13}C -labeled mannose and ^{13}C -labeled mannosamine. After 24 hours, we measured the two radioactive molecules by LC-MS/MS analysis. As expected, the amount of both labeled molecules in the lysosome was similar, indicating that mannose co-fractionated with mannosamine in the lysosome of colonic epithelial cells (Figure 1). The data was included in Figure 5 and Supplementary Figure 8 in the revised manuscript.

Figure 1. The chromatogram of mannosamine-1-¹³C and mannose-1-¹³C.

NCM460 cells were treated with ¹³C-labeled mannose (12.5 mM) along with same amount of ¹³C-labeled mannosamine for 12 hours and 24 hours. The chromatogram (a) and the amount (b) of ¹³C-labeled mannose along with ¹³C-labeled mannosamine were determined by LC-MS/MS analysis.

2. First, is the concentration of ¹³C-mannose used is 1 μ M while the cellular experiments use 25mM mannose. A 25,000 fold difference is not an acceptable alteration since 25mM mannose could be handled much differently than 1 μ M. The second issue is the time of incubation--only 5 min-- and third is the decision to look at only mitochondria, lysosomes and total signal. Mannose entering the cell is usually phosphorylated to man-6-p almost instantly. The authors do not account for all of the potential metabolism that must surely occur in any cell metabolizing mannose. This has been shown over decades of experiments. These three issues offer little meaningful to support their claim.

However, there is another way to look at this and it lies in the details of experimental protocol. Cellular experiments use 25mM mannose. If the authors repeat this experiment using 25 mM ¹³C mannose and show the lysosomes substantially ACCUMULATES mannose at multiple times over the 24 hours, that would alleviate some issues. I assume it is not the 5 min time frame that resets the potential benefits of mannose, but rather the long-term presence of mannose (in the lysosomes?) which could potentially disturb lysosomal homeostasis and integrity leading to the release of Cathepsin B. This may only be true in the presence of DSS, a highly negatively charged macromolecule which would be trafficked to the lysosome by endocytosis or macropinocytosis. The combination of DSS+mannose may ameliorate the lysosomal damage induced by DSS. A time course of over 24 hr showing progressive release of Cathepsin B from the lysosomes into the cytoplasm, along with maintenance of high levels of ¹³C-mannose within lysosomes, and an accounting for ¹³C-mannose derived macromolecules (mostly glycoproteins) and small metabolites would lend strength to explaining the authors very interesting results.

Responses:

We greatly appreciate the reviewer's thoughtful comments that help improve the manuscript.

As suggested, we have repeated the experiment using 25 mM ¹³C-mannose (approximately 4.5 mg for each 1×10^6 cells) to determine the lysosomes substantially accumulate mannose at 12 hours and 24 hours after mannose incubation. The results showed that mannose was mainly located in lysosomes, not mitochondria, in NCM460 cells (Figure. 5c). Additionally, the mannose was maintained in the lysosome of NCM460 cells over

24 hours. The lysosomal level of mannose is higher in cells treated with mannose for 12 hours than that for 24 hours (Supplementary Figure 15).

Besides, we determined the release of cathepsin B from the lysosomes into the cytoplasm in DSS-stimulated NCM460 cells at 12 and 24 hours after mannose treatment. Considering that western blot is only a semi-quantitative analysis, we compared the activity of cathepsin B in the cellular fractions at different time points by cathepsin B activity analysis. Interestingly, the result showed that the lysosome/cytosol ratio of cathepsin B activity was lower in cells treated with mannose for 24 hours compared to the cells treated with mannose for 12 hours (Supplementary Figure 15). The data indicated that the efficiency of the mannose-mediated inhibitory effect of cathepsin B release from the lysosome to the cytoplasm is associated with the level of mannose in the lysosome.

Mannose enters the cells using GLUT and is phosphorylated into mannose-6-phosphate by hexokinases. Mannose can then be used for glycosylation purposes or isomerized into fructose-6-phosphate. We agree with the reviewer's comment that mannose-derived glycoproteins and small metabolites may be involved in the mannose-mediated protection of lysosomal disorders. It is very interesting and will need a long time for the study. Therefore, we would like to further investigate this issue, and a brief discussion about this is included in the discussion section in the revised manuscript (pages 19-20, lines 403-410).

3. Assuming an internal intestinal volume of ~1ml, a gavage of 500ug mannose per gm body weight calculates to 12mg in a 25gm mouse, or a concentration of ~60mM. So choosing 25 mM mannose for cellular experiments is not unreasonable choice to mimic the mouse experiments.

Response: We would like to thank the reviewer for this suggestion.

After oral administration, mannose can be absorbed and metabolized by other organs besides the colon, such as the stomach and small intestine. On the other hand, the mice will drink freely after treatment with mannose, which might dilute the concentration of mannose in the colon.

In our experiments, we tested various concentrations of mannose and determined the optimal mannose concentration for the DSS-treated NCM460 cells. We have determined that 5-50 mM mannose is valid for NCM460 cells. We observed that 25 mM mannose is the optimal concentration to protect DSS-induced epithelial cell damage in vitro. Therefore, we chose 25 mM mannose for the in vitro studies through all the experiments.

4. Again, the dramatic results the authors show for effects on mice are impressive, however using a daily gavage of mannose would not be a welcomed approach by patients.

Response: We greatly appreciate this comment by the reviewer. Oral administration of substances is a common procedure in scientific experiments using laboratory animals. Our study demonstrated that oral administration of mannose significantly alleviated colonic inflammation in colitis mice.

Indeed, sustained-release allows delivery of a specific drug at a programmed rate that leads to drug delivery for a prolonged period of time. Therefore, preparations of sustained-release mannose may be used safely and effectively in patients. We add a short discussion about this in the revised manuscript (pages 21-22, lines 445-449).

5. Minor point: only male mice were used. This is a potential problem and is open to criticism at several levels.

Response: We appreciate this comment by the reviewer. We have determined the effect of mannose in female

mice with DSS-induced colitis. Our results showed that mannose also ameliorated the severity of DSS-induced colitis in female mice (Supplementary Figure 3), indicating that there is no gender difference in mannose-mediated attenuation of colonic inflammation in DSS-induced colitis mice. The data are presented as Supplementary Figure 3 in the revised manuscript.

Figure 2. Mannose administration ameliorates DSS-induced colitis in female mice.

Female mice (n=6 per group) were treated with 3.0% DSS in the presence or absence of mannose (500 µg/g/d) for 7 consecutive days. (a) The body weight changes during the experiments were monitored. (b) Colon tissues were isolated on the last day of the experiment. A representative photograph of colon tissue from each group is provided, and the colon length was recorded. Scale bar=1 cm. (c) The histological analysis of mouse colon tissue taken on the last day of the experiment was performed by hematoxylin and eosin (H&E), and alcian blue. Scale bar=100 µm. Histological scores of the DSS-induced colitis were evaluated (9 slides/sample). (d-f) Intestinal permeability was determined by the serum FD-4 concentration (d) and albumin level of serum (e), as well as the fecal α1-antitrypsin level (f) on day 7 after DSS challenge. *p<0.05, **p<0.01. Data from one representative experiment of three independent experiments are presented.

Reviewers' comments:

Reviewer #2 (Remarks to the Author):

This long saga continues with a response to my original skepticism about the use of tagged Mannoseamine as a surrogate for mannose. I explained that I would accept the authors' contention if they showed that either radioactive ^3H -Mannose or, preferably, ^{13}C -labeled mannose was co-fractionates with tagged mannoseamine and that it accounts for a substantial portion of the total of both labeled molecules. To their credit, the authors partially tried to respond the comment. They did not do a side-by-side comparison of the two molecules as I requested. In fact there was no mention of using tagged mannoseamine. Apparently it did not co-fractionate or they did not do the experiment I suggested. Instead they chose a modified version of this experiment and a rather simplified analysis. However, this experiment raises other significant issues.

First, is the concentration of ^{13}C -mannose used is 1 μM while the cellular experiments use 25mM mannose. A 25,000 fold difference is not an acceptable alteration since 25mM mannose could be handled much differently than 1 μM . The second issue is the time of incubation--only 5 min-- and third is the decision to look at only mitochondria, lysosomes and total signal. Mannose entering the cell is usually phosphorylated to man-6-p almost instantly. The authors do not account for all of the potential metabolism that must surely occur in any cell metabolizing mannose. This has been shown over decades of experiments. These three issues offer little meaningful to support their claim.

However, there is another way to look at this and it lies in the details of experimental protocol. Cellular experiments use 25mM mannose. If the authors repeat this experiment using 25 mM ^{13}C mannose and show the lysosomes substantially ACCUMULATES mannose at multiple times over the 24 hours, that would alleviate some issues. I assume it is not the 5 min time frame that resets the potential benefits of mannose, but rather the long-term presence of mannose (in the lysosomes?) which could potentially disturb lysosomal homeostasis and integrity leading to the release of Cathepsin B. This may only be true in the presence of DSS, a highly negatively charged macromolecule which would be trafficked to the lysosome by endocytosis or macropinocytosis. The combination of DSS+mannose may ameliorate the lysosomal damage induced by DSS. A time course of over 24 hr showing progressive release of Cathepsin B from the lysosomes into the cytoplasm, along with maintenance of high levels of ^{13}C -mannose within lysosomes, and an accounting for ^{13}C -mannose derived macromolecules (mostly glycoproteins) and small metabolites would lend strength to explaining the authors very interesting results.

Assuming an internal intestinal volume of $\sim 1\text{ml}$, a gavage of 500 μg mannose per gm body weight calculates to 12mg in a 25gm mouse, or a concentration of $\sim 60\text{mM}$. So choosing 25 mM mannose for cellular experiments is not unreasonable choice to mimic the mouse experiments.

Again, the dramatic results the authors show for effects on mice are impressive, however using a daily gavage of mannose would not be a welcomed approach by patients.

Minor point: only male mice were used. This is a potential problem and is open to criticism at several levels.

Detailed responses to the reviewer's comments (NCOMMS-19-34782H-Z)

We greatly appreciate the interest in our paper from the editorial teams. Thank you very much for your continued support and your valuable comments. We would like to emphasize the therapeutic potential of mannose on colitis progression. As required, we have constructed the manuscript. We changed the title of the manuscript to "Mannose attenuates colitis and improves intestinal barrier integrity by preserving lysosomal function". We have moved the data from supplementary figure 18 to figure 7, which demonstrated that mannose combined with mesalazine provides a synergistic therapeutic effect on colitis in mice. The data about the mechanism of action via AMPK-dependent lysosomal integrity was removed in the revision. Additionally, we agree that the action of mannose accumulation and its metabolism are complicated in this study, which warrants further investigations. Especially, the direct mechanism of action for mannose treatment of colitis *in vivo* is not clear. Therefore, we discuss the limitation of this study in the revision as suggested.

The primary concern from reviewer 2 is the subsequent metabolism of mannose after being taken up by these colonic cells. In the resubmitted manuscript, the ¹³C-labeled mannose was used to track the mannose metabolism in colonic epithelial cells. We conducted liquid chromatography/tandem mass spectrometry (LC-MS) analysis to determine the presence of lactate, pyruvate, mannose-6-P, GDP-mannose, and mannose for glycoprotein synthesis of colon epithelial cells. We also modified the description of mannose location in the revised manuscript. Here we address the concerns point by point.

Reviewer #2 (Remarks to the Author):

The authors have tried to respond to my previous review, but they still have not yet done it completely.

1. Response to Question 1. The authors cannot claim that mannose taken up by these colonic cells is primarily in the lysosome since they did not do a more extensive analysis to examine synthesis of glycoproteins or small molecules (lactate, pyruvate, mannose-6-P, GDP-Mannose). If you only analyzed two fractions, what happens to the remaining ¹³C-mannose?

Does it simply remain there?

Responses:

We thank the reviewer for this valuable suggestion.

We agree with the reviewer. Mannose can participate in the synthesis of glycoproteins or small molecules (lactate, pyruvate, mannose-6-P, GDP-Mannose) (**figure 1a**). In order to further explore the distribution and subsequent metabolism of mannose, we detected the content of the mannose derived molecules, unmetabolized mannose and the mannose involved in glycoprotein synthesis as you suggested. We incubated NCM460 cells with 25mM ¹³C-mannose. After 24 hours, the cells were divided into three equal fractions for the detection of mannose derived molecules (e.g., lactate, pyruvate, mannose-6-phosphate, and GDP-mannose) (**figure 1b**), unmetabolized mannose in lysosomes and non-lysosomal fractions and mannose involved in glycoprotein synthesis, respectively (**figure 1c**). The result showed that mannose is most catabolized to mannose-6-phosphate and involved in glycoprotein synthesis, and the minority of mannose was used to synthesize pyruvate, lactate, and GDP-mannose (**figure 1b**). In addition, we found that the intracellular residual mannose is located in lysosomes (**figure 1c**). The data (**Fig. 5e, f, line 227-230**) and the methods (**line 581-604**) are provided in the revised manuscript.

figure 1. The presence of mannose metabolites from mannose-1-¹³C

(a) Mannose (Man) in glycoconjugates can be derived directly from mannose imported through mannose transport. Mannose phosphorylated by hexokinase (HK) to Man-6-P. Phosphomannose isomerase (PMI) interconverts Fru-6-P and Man-6-P in a reaction. Fru-6-P can generate pyruvate through the glycolytic pathway. Pyruvate can be converted to lactate under the catalysis of lactate dehydrogenase. Man-6-P is converted into Man-1-P by phosphomannomutase (PMM) and then to GDP-mannose via GDP-mannose pyrophosphorylase for incorporation into glycoconjugates. NCM460 cells were treated with ¹³C-labeled mannose (25 mM) for 24 hours. (b) The amount of mannose-1-¹³C derived molecules (pyruvate, lactate, mannose-6-phosphate) were detected by LC-MS analysis. (c) The amount of mannose involved in glycoproteins and free mannose were detected by LC-MS/MS analysis.

The detailed method is as follows,

A: Measurement of lactate, pyruvate, mannose-6-P, and GDP-mannose

We quantified the level of ¹³C-lactate, ¹³C-pyruvate, ¹³C-mannose-6-P, and GDP-¹³C-mannose in the cells via LC-MS analysis. For ¹³C-mannose-tracing assays, NCM460 cells were treated in glucose-free RPMI 1640 medium supplemented with 100 U/ml penicillin, 100 µg/ml streptomycin, 1mM glucose, and 10% FBS and 25 mM ¹³C-labeled-mannose. Cells were rapidly extracted with ice-cold PBS for three times and washed with 500 µl extraction solvent (50% methanol, 30% acetonitrile, and 20% water) after 24 hours. Cells were then centrifuged at 16,000×g for 30 minutes at 4 °C, and the supernatants were analyzed by liquid chromatography-mass spectrometry (LC-MS). Exactive Orbitrap mass spectrometer (Thermo Fisher Scientific) was used together with a Thermo Fisher Scientific Accela HPLC system. The HPLC setup consisted of a ZIC-pHILIC column (SeQuant, 150 mm × 2.1 mm, 5 µm, Merck KGaA) with a ZIC-pHILIC guard column (SeQuant, 20 mm × 2.1 mm) and an initial mobile phase of 20% 20 mM ammonium carbonate, pH 9.4 and 80% acetonitrile. Cell extracts (5 µl) were injected, and metabolites were separated over a 15-minute mobile phase gradient, decreasing the acetonitrile content to 20%, at a flow rate of 200 µl/min and a column temperature of 45 °C. All metabolites were detected across a mass range of 75–1000m/z using the Exactive mass spectrometer at a resolution of 25,000 (at 200m/z), with electrospray ionization and polarity switching to enable both positive and negative ions to be determined in the same run over a total analysis time of 23 minutes. Lock masses were used and the mass accuracy obtained for all metabolites was below 5 p.p.m. Data were acquired with Thermo LCQuan 2.7 (Thermo Fisher Scientific) software.

B: Measurement of mannose involved in glycoprotein synthesis

Protein lysate of NCM460 cells was deglycosylated by Protein Deglycosylation Mix 2 (New England Bio-labs), according to the non-denaturing reaction protocol distributed by the manufacturer. Briefly, 100 µg of protein lysate of NCM460 cells, 5 µl of 10×deglycosylation mix buffer, and 5 µl of protein deglycosylation mix 2 were mixed to a total volume of 50 µl and incubated at 25 °C for 30 minutes, followed by overnight incubation at 37 °C. The supernatant was hydrolyzed by 2 M trifluoroacetic acid at 100 °C for 16 hours. An aliquot (10 µl) of the supernatant was taken for LC-MS/MS analysis.

C: Measurement of unmetabolized mannose in lysosomes and non-lysosomal fractions

After incubation with D-mannose-1-¹³C (Sigma-Aldrich, 25 mM) for 24 hours, the lysosomal and non-lysosomal fraction of NCM460 cells were isolated. The fraction was extracted with 1 ml ethanol: water (1:1). The mixture was then centrifuged for 30 minutes at 20,000 × g at room temperature. After centrifugation, an aliquot (10 µl) of the supernatant was taken for LC-MS/MS analysis. In order to calculate the proportion of unmetabolized mannose and various metabolized molecules, we convert the measured concentrations of metabolites to the concentrations derived from mannose (for example: 1mM ¹³C-pyruvate is equivalent to 1mM ¹³C-mannose; 1mM ¹³C-lactate is equivalent to 1mM ¹³C-mannose; 1mM ¹³C-mannose-6-phosphate is equivalent to 1mM ¹³C-mannose; 1mM GDP-¹³C-mannose is equivalent to 1mM ¹³C-mannose), then divide the amount of mannose by the amount of mannose entering the cell (the amount that enters the cell The amount of mannose = the amount of mannose added - the amount in the cell supernatant)

2. The authors also state "we measured the two radioactive molecules by LC-MS/MS analysis." These are not radioactive molecules, and they stable isotopes. Moreover, Figure 1 in their response shows a "level of mannose (mM)" What does that mean? It's a concentration, with no information about the samples. How was a concentration calculated?

Responses:

We agree with the reviewer that ¹³C-labeled mannose and ¹³C-labeled mannosamine are stable isotopes. We modified the description in the revised manuscript (**line 223**).

In Figure 1 of the last response letter, NCM460 cells were incubated with the same amount (12.5 mM each) of mannose-1-¹³C and mannosamine-1-¹³C for 12 hours or 24 hours. NCM460 cells were isolated and counted after incubation with mannose-1-¹³C and mannosamine-1-¹³C. The cellular fraction was extracted with 1ml ethanol: water (1:1). The mixture was then centrifuged for 30 minutes at 20,000 × g at room temperature. After centrifugation, an aliquot (10 µl) of the supernatant was taken for LC-MS/MS analysis. In order to display the content of mannose in cells more rigorously, the mannose concentration was converted into the amount of mannose per 10⁶ cells in the revised manuscript.

3. Response to question 2.

"The results showed that mannose was mainly located in lysosomes, not mitochondria". Again, mannose should be metabolized (glycolysis or glycan synthesis) and requires an accounting for that material to provide information as to whether ¹³C mannose is actually metabolized. In their response, the authors also provide information in the discussion (pages 19-20, lines 403-410). They say that the majority of mannose entering the cells accumulates in lysosomes and is also phosphorylated--but they don't show this. Instead they offer a reference from Trypanosomes. Perhaps ¹³C mannose does accumulate in lysosomes and, for unknown reasons, is excluded from further metabolism. This would be unprecedented, but so is this study and that's why it is important to do this analysis. How much in glycoproteins and how much in catabolized metabolites? It is impossible to know if the amount of mannose in the lysosomal fraction represents a small minority or a

substantial portion, which would imply a block in the exit of mannose from the lysosome and its use for glycoprotein synthesis or catabolism.

Responses:

We agree with the reviewer. We have done relevant experiments according to your suggestion to explain the distribution of mannose and its subsequent metabolism. The experimental results show that most of the mannose does not simply remain here, but participates in catabolism or is used to synthesize glycoproteins or small molecules. In contrast, a small amount of mannose was used to produce lactate and pyruvate. Besides the presence of few mannose in the lysosome, a quantity of mannose was phosphorylated to mannose-6-phosphate, and some mannose was used for the synthesis of glycoproteins and GDP-mannose. In addition, we found that a small fraction of unmetabolized mannose was mainly present in lysosomes by testing (**figure 2**). Finally, I also apologize for the inaccuracy of my previous description. We modified the sentence into “LC-MS/MS analysis showed that intracellular residual mannose could be located in lysosomes in NCM460 cells” (**line 220-221**). In the revised manuscript, we also presented the data showing that mannose was involved in metabolism and glycoprotein synthesis (**fig. 5g**). A brief discussion about this is included in the discussion section in the revised manuscript (**lines 377-386**).

figure 2. The percentage of free mannose and molecules derived from mannose-1-¹³C

NCM460 cells were treated with ¹³C-labeled mannose (25 mM) for 24 hours. The percentage of mannose-1-¹³C derived molecules (pyruvate, lactate, mannose-6-phosphate), mannose involved in glycoproteins and free mannose were calculated.

4. Figure S15 shows mannose as up to 1 mg/10⁶ cells. How can this be?

Responses:

We apologize for this inaccurate description. NCM460 cells were stimulated with 2.0% DSS in the presence of ¹³C-labeled mannose (25mM) for 12 or 24 hours. After incubation, the cells were collected and counted. Then, NCM460 cells were extracted with 1 ml ethanol: water (1:1) and then centrifuged for 30 minutes at 20,000 × g at room temperature. After centrifugation, an aliquot (10 μl) of the supernatant was taken for LC-MS/MS analysis. In Figure S15, the amount of mannose represents the amount of mannose per 10⁶ cells. We standardized the units of mannose and cellular metabolites to concentration as mM per 10⁶ cells in the revised manuscript (**Fig. S16**).

We are very grateful to Reviewer 2 for his/her very valuable suggestions on our work. We paid more attention to the effect of mannose on colon epithelial cells and colitis because our team mainly engaged in immunology and translational medicine, while did not pay much attention to the metabolism of mannose in the original version. However, after the professional, patient and comprehensive guidance of the reviewer 2, our article has been greatly improved because a lot of efforts have been made in the three aspects of mannose determination, tracking and whereabouts. Meanwhile, 7 rounds of revisions by the reviewer 2 also broadened our understanding of sugar chemistry. Reviewer 2 made us thoroughly feel the rigor, integrity and charm of science. Finally, we sincerely thank the reviewer 2 for your support and help to our work in the past two years, thank you for your tireless giving us valuable suggestions, and hope that we can get your help in the future work.

Reviewers' comments:

Reviewer #2 (Remarks to the Author):

The authors have responded well over the years to my comments. This time is no exception and most of the issues have been resolved. This reviewer appreciates their kind words and their commitment to finishing the impressive work they started.

I have a few important comments and suggestions.

First, any referral to "lysosome" to indicate the fraction or fractionation must be used with great care. Usually the term lysosomal fraction is most appropriate. However, based on the data provided in the Abcam lysosome isolation kit (lysosome-isolation-kit-ab234047), the lysosomal purification based on specific activity of acid phosphatase is about 2.5x. Lysosomes comprise less than 5% of the total cell proteins, therefore this lysosomal fraction contains <10% lysosomes, and 90% something else. Must be very careful about this.

Also, the very great majority of lactate formed from glucose or mannose will not be within the cells. To estimate how much lactate or pyruvate is produced would require analysis of the medium. So making much sense of the amount of this within the cells is not very revealing. More likely, much of the glucose or mannose has been converted to CO₂ by OxPhos and would not be seen. I'm not sure the small molecule data says anything about the fate of either sugar.

Finally, on line 383 of the main text, there is a referral to Mannose-6-phosphate and lysosomal enzymes. The relationship of metabolic Mannose-6-phosphate and Mannose-6-phosphate on lysosomal enzymes is only in the name. The Mannose-6-phosphate on lysosomal enzymes is only in the context of an N-linked glycan, not free Mannose-6-phosphate in the metabolic issues discussed here. This reference should be eliminated from the Discussion. Not relevant.

I'm also not sure how relevant the mannoseamine data is to the current paper; might consider dropping it.

Overall, the observations and general flow remain impressive and will stimulate a lot of discussion.

Detailed responses to the reviewer's comments (NCOMMS-19-34782H-Z)

Reviewer #2 (Remarks to the Author):

The authors have responded well over the years to my comments. This time is no exception and most of the issues have been resolved. This reviewer appreciates their kind words and their commitment to finishing the impressive work they started.

I have a few important comments and suggestions.

1. First, any referral to "lysosome" to indicate the fraction or fractionation must be used with great care. Usually the term lysosomal fraction is most appropriate. However, based on the data provided in the Abcam lysosome isolation kit (lysosome-isolation-kit-ab234047), the lysosomal purification based on specific activity of acid phosphatase is about 2.5x. Lysosomes comprise less than 5% of the total cell proteins, therefore this lysosomal fraction contains <10% lysosomes, and 90% something else. Must be very careful about this.

Responses:

We thank the reviewer for this valuable suggestion.

We agree with the reviewer.

Since acid phosphatases are mainly found in lysosomes, the acid phosphatases in lysate, enriched lysosome, and purified lysosome are actually acid phosphatases in lysosomes. The kit purchased from Abcam (ab234047) might have limitation in the purity of isolated lysosomes. We detected the lysosomal marker LAMP2 and the cytoplasmic marker β -actin after isolating lysosomes to verify the purity of lysosomes. Indeed, several literatures have also used this kit for lysosome isolation^{1, 2, 3}. Thus far, it is feasible to use Abcam lysosome isolation kit (lysosome-isolation-kit-ab234047) to isolate lysosomes. To be more accurate, we modified the description of lysosomal fraction in the revised manuscript.

2. Also, the very great majority of lactate formed from glucose or mannose will not be within the cells. To estimate how much lactate or pyruvate is produced would require analysis of the medium. So making much sense of the amount of this within the cells is not very revealing. More likely, much of the glucose or mannose has been converted to CO₂ by OxPhos and would not be seen. I'm not sure the small molecule data says anything about the fate of either sugar.

Responses:

We agree with the reviewer and thank the valuable comments.

As the editor said, much of the glucose or mannose goes to carbon dioxide through anaerobic glycolysis or aerobic respiration eventually. In the last edition we followed the reviewer's suggestion to observe the distribution and fate of mannose after entering the cell, and the results showed that mannose is most catabolized to mannose-6-phosphate and involved in glycoprotein synthesis. We agree with the reviewer that much of the glucose or mannose has been converted to CO₂ by OXPHOS. Also, the great majority of lactate and pyruvate formed from mannose secreted into culture supernatants. Therefore, we modified the description of the result and included a short discussion in the revised manuscript (line 355-360).

3. Finally, on line 383 of the main text, there is a referral to Mannose-6-phosphate and lysosomal enzymes. The relationship of metabolic Mannose-6-phosphate and Mannose-6-phosphate on lysosomal enzymes is only in the name. The Mannose-6-phosphate on lysosomal enzymes is only in the context of an N-linked glycan, not free Mannose-6-phosphate in the metabolic issues discussed here. This reference should be eliminated from the

Discussion. Not relevant.

Responses:

We highly appreciate the valuable comments by the reviewer. We have removed the reference in the revised manuscript.

4. I'm also not sure how relevant the mannoseamine data is to the current paper; might consider dropping it.

Response:

We thank the reviewer for this suggestion. We have removed the results about mannoseamine in the revised manuscript.

Overall, the observations and general flow remain impressive and will stimulate a lot of discussion.

1. Park SH, Park S, Shin I. Synthesis of an Hsp70 inhibitor and its assessment of lysosomal membrane permeabilization. *STAR Protoc* **2**, 100349 (2021).
2. Guo S, *et al.* TRIB2 modulates proteasome function to reduce ubiquitin stability and protect liver cancer cells against oxidative stress. *Cell Death Dis* **12**, 42 (2021).
3. Minghua Yang PC, Jiao Liu, Shan Zhu, Guido Kroemer,, Daniel J. Klionsky MTL, Herbert J. Zeh, Rui Kang, Daolin Tang. clockophagy is a novel selective autophagy process favoring. *Science advances* **5**, (2019).